# Mitigating Semantic Collapsing Problem in Generative Personalization with Test-time Embedding Adjustment

**Anh Bui**[1] **Trang Vu**[1] **Trung Le**[1] **Junae Kim**[2]
**Tamas Abraham**[2] **Rollin Omari**[2] **Amar Kaur**[2] **Dinh Phung**[1]

[1]Monash University
[2]Defence Science and Technology Group, Australia

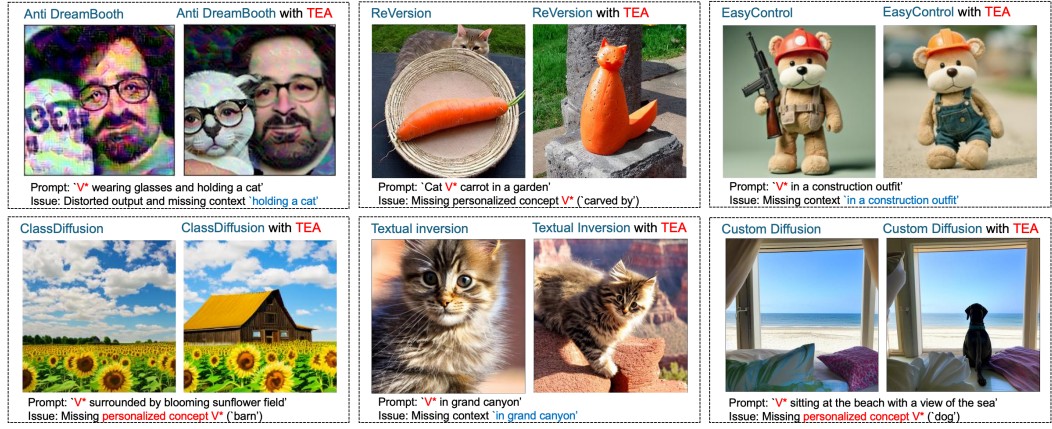

Figure 1: Our Test-time Embedding Adjustment (TEA) method consistently enhances text-image alignment across diverse personalization approaches (Textual Inversion, DreamBooth, and their variants) and architectures (Stable Diffusion, Flux). Notably, TEA also counteracts the anti-personalization effect of Anti-DreamBooth and restores the protected concept.

## Abstract

In this paper, we investigate the semantic collapsing problem in generative personalization, an under-explored topic where the learned visual concept ($V^*$) gradually shifts from its original textual meaning and comes to dominate other concepts in multi-concept input prompts. This issue not only reduces the semantic richness of complex input prompts like "a photo of $V^*$ wearing glasses and playing guitar" into simpler, less contextually rich forms such as "a photo of $V^*$" but also leads to simplified output images that fail to capture the intended concept. We identify the root cause as unconstrained optimisation, which allows the learned embedding $V^*$ to drift arbitrarily in the embedding space, both in direction and magnitude. To address this, we propose a simple yet effective training-free method that adjusts the magnitude and direction of pre-trained embedding at inference time, effectively mitigating the semantic collapsing problem. Our method is broadly applicable across different personalization methods and demonstrates significant improvements in text-image alignment in diverse use cases. Our code is published at `https://github.com/tuananhbui89/Embedding-Adjustment`.

## 1 INTRODUCTION

Text-to-image (T2I) diffusion models have achieved unprecedented fidelity and flexibility in image generation and sparked growing interest in generative personalization (Gal et al., 2022; Ruiz et al., 2023). This emerging problem aims to generate images of a specific user-defined visual concept (e.g. a particular person, pet, or object) in different contexts (e.g. on the beach) using a small set of user-provided reference images paired with text prompts describing the desired context. The core objective is to generate visually compelling images that faithfully preserve the unique characteristics of the personal concept while remaining semantically aligned with the textual prompt. Despite recent progress, misalignment between the generated image and the textual prompt is still a major concern.

A robust generative personalization method should allow the user-defined visual concept to be composed with arbitrary contexts in the text prompt without losing fidelity or expressiveness. However, existing approaches often struggle to maintain prompt and generated image alignment, particularly with complex or multi-concept prompts (Kong et al., 2024; Zhu et al., 2025). As illustrated in Figure 1, the user-defined concept can overpower or distort other elements in the prompt, leading to unsatisfactory generations. This issue has commonly been attributed to *language drift* - a phenomenon where the model gradually forgets how to generate its pretrained concepts and instead becomes overly focused on the user-defined ones (Lee et al., 2019). This drift typically stems from overfitting on a limited number of reference images (Ruiz et al., 2023). Beyond overfitting, other factors have also been attributed, such as the limited expressiveness of textual embeddings which compress complex visual concepts into single tokens (Zhang et al., 2023; Mou et al., 2024) and the entangled nature of reference sets, where samples may contain co-occurring objects or irrelevant contextual features (Avrahami et al., 2023; Jin et al., 2024). While misalignment between textual prompt and generated images is a well-recognized challenge in generative personalization, with numerous mitigation strategies proposed, including latent optimization (Rassin et al., 2023), regularization (Han et al., 2023; Qiu et al., 2023; Arar et al., 2024) and concept disentanglement (Motamed et al., 2024; Huang et al., 2024a), the underlying causes and mechanisms remain relatively underexplored.

To gain deeper insight into the misalignment problem, this paper presents an empirical investigation into the dynamics of learned personalised tokens throughout the personalisation process. Our analysis reveals an intriguing phenomenon: the personalised token gradually loses its original textual semantic meaning while acquiring increased visual information from the reference images. When such a token is used in a prompt with rich descriptive text, the generated image becomes disproportionately *semantically dominated by the personalized concept*, often neglecting the other intended elements in the prompt. For example, in Figure 1, if one learns a token $V^*$ to represent a particular cat, a prompt like "$V^*$ in grand canyon" may yield an image that vividly depicts the cat but fails to properly render the grand canyon background. Essentially, the prompt's semantic complexity collapses to a simplified form centred on $V^*$. We refer to this phenomenon as the *semantic collapsing problem* (SCP). Unlike language drift where the personalized model overfits to a learned concept and loses its ability to generate other pretrained concepts, SCP arises when the personalized embedding collapses, no longer retaining meaningful textual semantics and instead encoding primarily the visual information of the reference concept. While SCP might not severely affect trivial prompts (e.g., "a photo of $V^*$"), it undermines the compositionality of the T2I diffusion model on complex prompts.

We identify the root cause of SCP as *unconstrained optimisation*, which allows the learned embedding to drift arbitrarily in the embedding space, both in direction and magnitude. We propose a simple yet effective remedy: a training-free, **test-time embedding adjustment** (TEA) strategy that realigns the learned concept embedding with its original semantic meaning at inference time. The key idea is to calibrate the embedding's magnitude and direction to be closer to that of its reference concept. This adjustment is done *without altering the model weights or requiring any additional training*. The embedding is modified on-the-fly before image generation. By enforcing a small rotation and rescaling in the text encoder's latent space, we constrain the personalized token to behave more like a regular word, ensuring that it contributes to the image generation in balance with other tokens in the prompt. Our approach is lightweight and broadly compatible with **almost all existing** personalization methods and demonstrates significant improvements in text-image alignment in diverse use cases. Surprisingly, beyond improving personalization quality, we also uncover a surprising vulnerability in anti-personalization frameworks like Anti-DreamBooth (Van Le et al., 2023) under the lens of SCP. In particular, when applying TEA to models poisoned by Anti-DreamBooth, we find that TEA can partially *reverse adversarial corruption* and restore more faithful generations of the protected

concept (see Figure 1). This result highlights a false sense of security in current anti-personalization defenses and provides new insights into their limitations.

In summary, our contributions are as follows: ❶ We define the semantic collapsing problem (SCP) in generative personalization problem, and provide an empirical analysis of its existence in both textual and image spaces. Our analysis reveals its root cause, which is the unconstrained optimisation. ❷ We propose the test-time embedding adjustment, a novel solution that requires no additional training, to mitigate SCP by aligning the learned embedding's direction and norm with its original semantic concept. ❸ We demonstrate that our proposed approach can be applied into **almost all existing** personalization frameworks such as Textual Inversion (Gal et al., 2022), DreamBooth (Ruiz et al., 2023), Custom Diffusion (Kumari et al., 2023), EasyControl (Zhang et al., 2025), ReVersion (Huang et al., 2024b), and ClassDiffusion (Huang et al., 2024a) to significantly improve image generation in complex prompts across a wide range of scenarios. ❹ We show that TEA unexpectedly mitigates adversarial corruption introduced by Anti-DreamBooth, revealing an overlooked weakness in anti-personalization defences.

## 2 SEMANTIC COLLAPSING PROBLEM IN GENERATIVE PERSONALIZATION

### 2.1 TERMINOLOGIES

Given a set of personal images $\mathcal{X}$ and a pre-trained T2I model $\epsilon_\theta$, the goal of generative personalization is to identify a textual embedding $v^*$ associated with a specific verbalizable keyword $V^*$ (e.g., 'sks', '<new>', etc.). This keyword represents the implicit visual concept shared in the reference set $\mathcal{X}$, enabling the model to generate images with the personal concept using any textual prompt $p$ containing the keyword $V^*$, e.g., $\lfloor p, V^* \rfloor$ = 'A photo of $V^*$ playing on a beach', where $\lfloor ., . \rfloor$ is the sentence construction operator. We denote $c$ as the reference semantic concept of the keyword $V^*$.

We denote $M$ is the embedding matrix of the entire vocabulary of the text encoder $\tau$, and $M_k$ is a specific row of the matrix corresponding to the token $k \in \text{vocab}_\tau$, where $k$ can be $V^*$ (i.e., $v^* = M_{V^*}$) or any arbitrary token like 'dog', 'cat', etc. We denote $\hat{x} = G(p)$ as the image generation model that takes a prompt $p$ as input and outputs an image $\hat{x}$.

### 2.2 SEMANTIC COLLAPSING PROBLEM

**Problem Statement.** The **semantic collapsing problem** (SCP) refers to the phenomenon where the keyword $V^*$ loses its original *textual semantic meaning* while acquiring increased *visual information* from the reference concept during the personalization process. As a result, a prompt $\lfloor p, V^* \rfloor$, consisting of a context $p$ and the concept $V^*$, becomes dominated by the learned concept $V^*$, eventually collapsing to a simplified form.

**Why SCP Matters.** We argue that SCP may not pose a serious issue for simple prompts, e.g., $\lfloor p, V^* \rfloor$ = 'a photo of $V^*$', where the primary information conveyed is still the visual concept $V^*$. However, for more complex prompts where the context $p$ contributes meaningfully to the overall semantics, such as 'a photo of $V^*$ wearing glasses and writing in a red notebook', SCP becomes more problematic as illustrated in Figure 10. In such cases, the generated image is more likely to be dominated by the learned concept $V^*$ and less likely to reflect the intended context $p$.

**Comparison with Other Challenges in Generative personalization.** First, we emphasise that SCP is not specific to the two representative methods we study (TI and DB), but is a general issue in personalization. Second, SCP is distinct from other recognised challenges (ref. Section A) such as the *language drift* problem (Ruiz et al., 2023), which describes how the personalized model $\epsilon_\theta$ overfits to a learned concept and loses generalisation. Third, while SCP contributes to the broader challenge of misalignment between generated images and prompts, a major concern in generative personalization, it stems from a specific cause: the unconstrained optimisation of the embedding during personalization, which has not been thoroughly studied in prior work.

### 2.3 EMPIRICAL HUNTING FOR SCP

In this section, we present empirical evidence supporting the existence of the semantic collapsing problem and its impact on generation quality. Our key findings are as follows:

❶ **Existence of SCP.** SCP exists in the textual domain, where the prompt $\lfloor p, V^* \rfloor$ is dominated by the learned embedding $V^*$ and the semantic meaning of the entire prompt gradually collapses to the learned embedding $V^*$, i.e., $\tau(\lfloor p, V^* \rfloor) \to \tau(V^*)$.

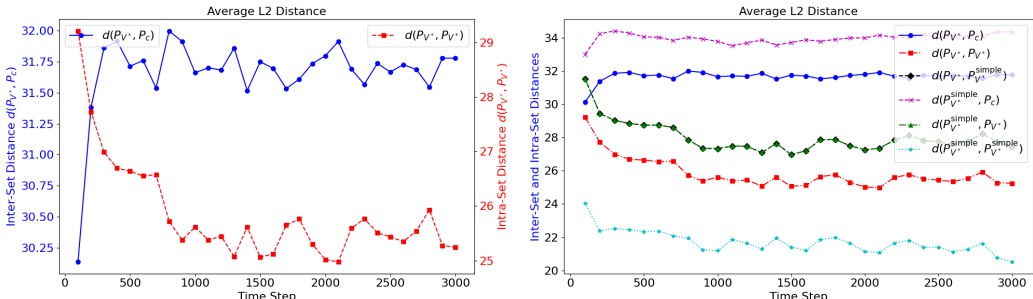

Figure 2: (a/left) The inter-set distance $d(P_{V*}, P_c)$ and intra-set distance $d(P_{V*}, P_{V*})$ over the personalization process, and (b) The distance between all possible pairs of sets, notably $d(P_{V*}, P_{V*}^{\text{simple}})$.

❷ **Negative Impact on Generation Quality.** SCP leads to the degradation/misalignment in generation quality in the image space, i.e., $G(\lfloor p, V^* \rfloor) \to G(V^*)$, particularly for prompts with complex semantic structures.

❸ **Surprisingly Positive Impact.** SCP can also lead to the positive impact on generation quality, particularly for prompts where the concept $c$ requires a strong visual presence to be recognisable.

❹ **Root Cause of SCP.** SCP arises from unconstrained optimisation during personalization, which leads to arbitrary shifts (both in magnitude and direction) in the embedding of $V^*$ away from its original semantic concept $c$.

### 2.3.1 EMPIRICAL EVIDENCE FOR SCP IN TEXTUAL SPACE

Recall our hypothesis: a keyword $V^*$ initialised from a concept $c$ to capture a visual target $v_{gt}$ will lose its semantic meaning and dominate any arbitrary context $p$ when combined into a prompt.

To verify this, we propose measuring the difference between $V^*$ and $c$ in the presence of a diverse set of contextual prompts $A = \{a_1, a_2, \cdots, a_n\}$. These are generated by querying an LLM with the instruction: "Write 200 sentences with diverse topics and contents. Each sentence should be 10–30 words long and must include the keyword $c$." Sample sentences are provided in Table 3. We then construct two sets of prompts: $P_{V^*} = \{\lfloor a_i, V^* \rfloor\}_{a_i \in A}$ and $P_c = \{\lfloor a_i, c \rfloor\}_{a_i \in A}$, which are used to assess the contextualised difference between $V^*$ and $c$.

To quantify this difference, we compute distances between the sets $P_{V^*}$ and $P_c$ using four metrics: Euclidean, Hausdorff, Mahalanobis, and KL divergence. We also evaluate intra-set variability $d(P_{V^*}, P_{V^*})$ measuring the separation among items within $P_{V^*}$. The distance metrics are summarised in Table 5. We use Textual Inversion (TI) and DreamBooth (DB) to learn a personalized human face concept. The training data comprises 16 images from the CelebA dataset (Liu et al., 2015) (subject ID: 342, 'Henry Cavill'). The embedding $v^*$ is initialised using the concept $c = $ 'man' for TI, and $c = $ 'a photo of a sks man' for DB.

**Results.** Figure 2(a) shows the average inter-set and intra-set distances, $d(P_{V^*}, P_c)$ and $d(P_{V^*}, P_{V^*})$, measured over training iterations. It can be seen that the inter-set distance $d(P_{V^*}, P_c)$ increases steadily over time, indicating that the learned embedding $v^*$ progressively diverges from its initial textual semantic meaning $c$. Interestingly, the intra-set distance $d(P_{V^*}, P_{V^*})$ decreases over time, suggesting that the embeddings of prompts $\lfloor a_i, V^* \rfloor$ within $P_{V^*}$ become less diverse and more similar to one another. This reflects a *growing dominance* of the learned embedding $v^*$ across prompts, effectively *overriding the contextual* variations in $a_i$ and becoming the principal semantic component of each prompt.

To further verify this dominance effect, we introduce an additional set of simple prompts, denoted $P_{V^*}^{\text{simple}}$, which consists of 200 concise sentences such as 'a photo of a $V^*$', 'a portrait of a $V^*$', etc., where $V^*$ is clearly the central concept. As shown in Figure 2(b), the distance $d(P_{V^*}, P_{V^*}^{\text{simple}})$, which captures the difference between complex prompts $\lfloor p, V^* \rfloor$ and simple prompts $\lfloor V^* \rfloor$, decreases over time. This trend indicates that the *representations of complex prompts become increasingly similar* to those of simple prompts in $P^{\text{simple}}$, further supporting the hypothesis that $v^*$ *gradually dominates and collapses* the semantic contribution of contextual components.

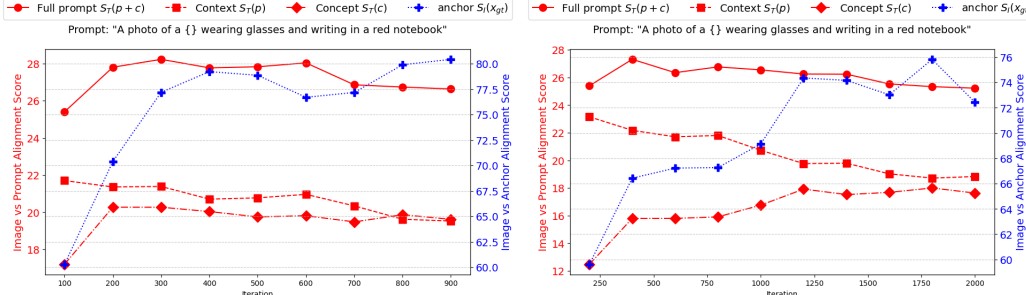

Figure 3: Analysis of the SCP on TI (left) and DB (right). Alignment with the ground-truth image $(S(\hat{x}, x_{gt}) - \Diamond)$ increases over time, while alignment with the contextual part $(S(\hat{x}, p) - \Box)$ decreases.

### 2.3.2 THE TWO-WAY IMPACTS ON PERSONALIZATION

In this section, we extend our analysis to investigate how SCP impacts image generation quality. Specifically, we generate personalized images $\hat{x} = G(\lfloor p, V^* \rfloor)$ using a list of prompts $P$ (e.g., 'a photo of a $V^*$ man holding a cat'), with 100 images generated per prompt. We evaluate the generation quality using the CLIP-Image-Image alignment score $S_I = S(\hat{x}, x_{gt})$, which measures the similarity between the generated image $\hat{x}$ and the ground-truth image $x_{gt}$ of the reference concept. In addition, we compute three CLIP-Text-Image alignment scores: **(i)** CLIP$_T^p$ or $S_T^p = S(\hat{x}, p)$ alignment with the contextual part $p$ (e.g., 'holding a cat'), and **(ii)** CLIP$_T^c$ or $S_T^c = S(\hat{x}, c)$ alignment with the original concept $c$ (e.g., 'a man'). **(iii)** CLIP$_T^f$ or $S_T^f = S(\hat{x}, \lfloor p, c \rfloor)$ alignment with the full prompt.

Interestingly, we observe that the SCP has both negative and positive effects on generation quality, depending on the nature of the prompt. Our key findings (illustrated in Figure 3) are as follows:

**(Unsurprising)** The image-to-image alignment score $S_I$ increases over time (+ line), indicating that $V^*$ effectively captures the visual appearance of the target concept. This confirms that the learned embedding $V^*$ successfully personalizes the visual identity from the reference set.

**(Surprising Negative Impact)** The context-to-image alignment score $S_T^p$ decreases over time ($\Box$ line), showing that generated images increasingly *lose alignment with the contextual component p*. This highlights the negative impact of semantic collapsing: as $V^*$ dominates the prompt, the image generator pays less attention to the surrounding context.

**(Unexpected Positive Effect)** For some prompts, the concept-to-image alignment score $S_T^c$ actually increases over time ($\Diamond$ line). Early in training, prompts with $c$ (e.g., man) often generate images dominated by the context $p$. As training progresses, the learned embedding $V^*$ reduces this context dominance, strengthening the representation of $c$. This effect is most evident when $c$ demands a strong visual presence (e.g., man in "man writing in a red notebook"), where close-up subject renderings naturally downplay the surrounding context.

### 2.3.3 THE ROOT CAUSE OF SCP

In the previous sections, we demonstrated the semantic collapsing problem in both textual and image generation spaces. In this section, we investigate the underlying cause of this phenomenon. Our key findings are summarised below:

① The semantic collapsing problem arises from the unconstrained optimisation process used in Equation 4 and Equation 5. Without any regularisation, the learned embedding $V^*$ can deviate significantly from the original semantic meaning $c$ in both magnitude and direction. Specifically, the embedding norm becomes much larger ($|M_{V^*}| \gg |M_c|$), and the cosine similarity between the two drops sharply ($\cos(M_{V^*}, M_c) \ll 1$), leading to a semantic drift.

② As a result of this semantic shift in $V^*$, the embedding of the entire prompt $\lfloor p, V^* \rfloor$ is also affected. That is, the prompt embedding becomes nearly identical to that of $V^*$, i.e., $\tau(\lfloor p, V^* \rfloor) \approx \tau(V^*) \neq \tau(\lfloor p, c \rfloor)$, which directly manifests the semantic collapsing problem discussed earlier.

To support this analysis, Figure 4 presents a histogram of embedding norms for all vocabulary tokens, along with the norm of $V^*$ tracked over the course of optimisation. It is evident that the norm of $V^*$ grows significantly, placing it in the long tail of the distribution—substantially larger than

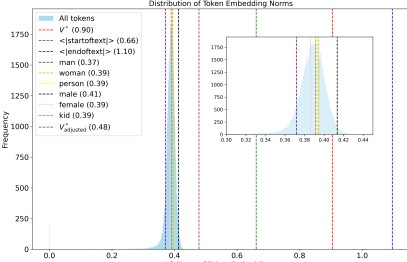
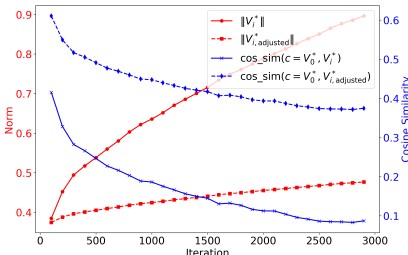

Figure 4: Left: The distribution of the norm of the token embedding $M$ including special token $V^*$, Right: The semantic drift of $V^*$ in term of magnitude and direction over time. The adjusted embedding $V^*_{\text{adjusted}}$ is obtained by using TEA with $\alpha = 0.2$ and $\beta = 1.5$. The same phenomenon is observed in DreamBooth as shown in Figure 21.

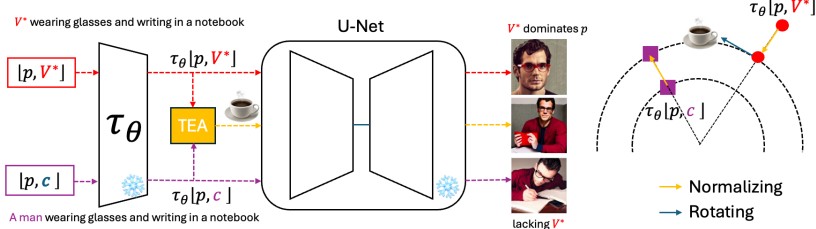

Figure 5: (left) TEA framework that adjusts the embedding on inference time where both U-Net and text encoder are just personalized pre-trained models. (right) the two stages of TEA: normalization and rotation with SLERP.

standard tokens such as 'man', 'woman', and 'person', and approaching that of special tokens like '<|startoftext|>' or '<|endoftext|>'.

It is worth noting that in some DreamBooth-based implementations (e.g., DreamBooth with LoRA in Diffusers (Hugging Face)), the full text encoder is fine-tuned to incorporate the personalized concept, instead of updating a dedicated token embedding $V^*$ as done in Textual Inversion. In such cases, while the individual embedding vector $M$ is not explicitly altered, the semantic shift still occurs at the prompt level, i.e., $\tau(\lfloor p, V^* \rfloor) \approx \tau(V^*) \neq \tau(\lfloor p, c \rfloor)$.

Additionally, these DreamBooth implementations already include a gradient clipping mechanism (Hugging Face) to constrain parameter updates. However, this method was not designed with semantic stability in mind, and does not prevent cumulative semantic drift in practice. Even when the gradient norm is bounded, the embedding can still gradually shift over successive iterations.

To the best of our knowledge, our work is the *first to identify and explain the root cause of the semantic collapsing problem* as a consequence of unregularised embedding dynamics.

## 3  TEST-TIME EMBEDDING ADJUSTMENT FOR SCP

As demonstrated in the previous section, SCP exhibits two-way impacts that vary depending on the nature of the context prompt. This variability makes it challenging to devise a universal solution that mitigates the negative effects of SCP while preserving its beneficial aspects across all inference prompts. Recall that in the earlier analysis, we observed that the learned embedding $V^*$ often drifts from its original semantic anchor $c$ due to unconstrained optimisation—resulting in significant shifts in both magnitude and direction. This raises a natural question: *Can we reverse this semantic shift at test time by adjusting $V^*$, without modifying the personalization method?*

A key advantage of this approach is that it is training-free and can be applied to **almost all existing** personalization methods, regardless of whether it is based on Textual Inversion or DreamBooth. Surprisingly, this simple adjustment proves to be highly effective.

**Embedding Adjustment.** Given a pre-trained embedding matrix $M$ that includes a learned token $V^*$ (as in Textual Inversion), and a target concept $c$ toward which we wish to regularise, we propose to adjust $M_{V^*}$ by aligning both its magnitude and direction with $M_c$. This is achieved by first normalising the vectors and then applying Spherical Linear Interpolation (SLERP) (Shoemake, 1985) to interpolate the direction of $M_{V^*}$ towards $M_c$, which is effective in high-dimensional vector spaces.

$$\hat{M}_{V^*} = \frac{\sin((1-\alpha)\theta)}{\sin(\theta)}\tilde{M}_{V^*} + \frac{\sin(\alpha\theta)}{\sin(\theta)}\tilde{M}_c \tag{1}$$

Here, $\theta$ is the angle between the normalized vectors $\tilde{M}_c$ and $\tilde{M}_{V^*}$, and $\alpha \in [0,1]$ controls the rotation factor, where the bigger $\alpha$ is, the more the embedding is rotated towards $M_c$. The normalisation vectors are defined as $\tilde{M}_{V^*} = \beta\|M_c\|\frac{M_{V^*}}{\|M_{V^*}\|}$ and $\tilde{M}_c = \beta\|M_c\|\frac{M_c}{\|M_c\|}$ where $\beta$ is the scaling factor to control the magnitude of the embedding relative to the reference concept $c$.

In Dreambooth-based personalization, because the embedding matrix $M$ is not updated during the optimisation, we propose to adjust at the prompt level instead of the token level as illustrated in Figure 5. More specifically, given a prompt $\lfloor p, V^* \rfloor$ and a target prompt $\lfloor p, c \rfloor$, we obtain the two embeddings $\tau(\lfloor p, V^* \rfloor)$ and $\tau(\lfloor p, c \rfloor)$ from the text encoder $\tau_\phi$ and then adjust the embedding of $\lfloor p, V^* \rfloor$ by using the above equation on every token in the prompt.

$$\hat{\tau}(\lfloor p, V^* \rfloor)[i] = \frac{\sin((1-\alpha)\theta_i)}{\sin(\theta_i)}\tilde{\tau}(\lfloor p, V^* \rfloor)[i] + \frac{\sin(\alpha\theta_i)}{\sin(\theta_i)}\tilde{\tau}(\lfloor p, c \rfloor)[i] \tag{2}$$

where $i$ indexes each token in the prompt, and $\theta_i$ is the angle between the $i$-th token embeddings of the two prompts after normalisation. This method enables a test-time adjustment of semantic drift without retraining, making it a lightweight and broadly applicable solution to mitigating SCP effects.

## 4 EXPERIMENTS

In this section, we demonstrate the effectiveness of TEA on addressing the SCP across six representative and recent personalization methods, two architectures (Stable Diffusion and Flux) and three datasets (CS101, CelebA, and Relationship) consisting of total 22 concepts. Due to the space limit, we present the main results here and refer readers to Appendix C and D for additional quantitative and qualitative findings. Full reproducibility details are available in the GitHub repository.

### 4.1 EXPERIMENTAL SETUP

**Reference Images.** We use a subset of 9 concepts from the CustomConcept101 (CC101) dataset as in Kumari et al. (2023), each of which has 3-15 images, including 'Barn', 'Tortoise plushy', 'Teddy-Bear', 'Wooden Pot', 'Dog', 'Cat', 'Flower', 'Table', 'Chair' subjects. For the human concept, we use a subset of 10 concepts from the CelebA-HQ dataset (Liu et al., 2015), which includes 10 identities with 10-15 images per subject. Sample images are shown in Fig. 15 and 16.

**Prompts.** We collect complex prompts from the CC101 dataset, where each prompt contains at least two concepts, e.g., 'a watercolor painting of $V^*$ tortoise plushy on a mountain'. For the human concept, we create a list of 17 prompts, where each prompt contains the main concept and a complex context/action, e.g., 'A photo of a $V^*$ wearing glasses and writing in a red notebook'. The prompts can be found in Table 4.

**Metrics.** In addition to the CLIP-T and CLIP-I alignment scores introduced in Section 2.3, we also use the DINO **image-image** alignment score (Caron et al., 2021) to evaluate the alignment between the generated images and the reference images.

We also use VLM-based evaluation metrics, **VLM-P** and **VLM-I** using VLM (i.e., GPT-4o-mini) as the judge to assess the alignment between the generated images and the reference images and the prompts, respectively. Both metrics output a score between 0 and 4, where 0 means there are no correspondence between the generated image and the reference image (VLM-I) or input prompt (VLM-P), while 4 means the perfectly matches. The final score for each metric is obtained by averaging all inference prompts and samples and normalizing to the range [0, 100%]. We include the full system prompts and evaluation scripts in the Github repository.

### 4.2 EVALUATION RESULTS

Table 1: Performance over EasyControl (ES) OminiControl (OC) and ReVersion (RV) when integrating with our TEA. Qualitative results are shown in Figures 28, 29, 31.

| Method | $\text{CLIP}_T^p \uparrow$ | $\text{CLIP}_T^f \uparrow$ | CLIP-I $\uparrow$ | DINO-I $\uparrow$ | VLM-P $\uparrow$ | VLM-I $\uparrow$ |
|---|---|---|---|---|---|---|
| *CC101 - Pet Dog* | | | | | | |
| ES | 18.54 | 26.02 | 61.33 | 43.71 | 64.25 | 74.00 |
| ES+TEA | 18.72 (+0.18) | 26.11 (+0.09) | 64.56 (+3.23) | 48.32 (+4.61) | 66.50 (+2.25) | 77.25 (+3.25) |
| *CC101 - Plushie Teddybear* | | | | | | |
| ES | 20.48 | 26.80 | 81.64 | 49.08 | 78.00 | 80.25 |
| ES+TEA | 20.61 (+0.13) | 27.3 (+0.50) | 82.84 (+1.20) | 51.17 (+2.09) | 80.25 (+2.25) | 81.50 (+1.25) |
| *Subject - Clock* | | | | | | |
| OC | 18.11 | 23.90 | 81.37 | 32.41 | 67.50 | 62.25 |
| OC+TEA | 18.78 (+0.67) | 23.98 (+0.08) | 83.10 (+1.73) | 34.48 (+2.07) | 71.75 (+4.25) | 64.50 (+2.25) |
| *Subject - Oranges* | | | | | | |
| OC | 21.49 | 27.62 | 70.43 | 30.33 | 68.50 | 53.00 |
| OC+TEA | 21.60 (+0.11) | 27.70 (+0.08) | 71.90 (+1.47) | 31.64 (+1.31) | 70.00 (+1.50) | 55.50 (+2.50) |
| *Subject - Penguin* | | | | | | |
| OC | 20.30 | 31.61 | 78.58 | 45.59 | 86.25 | 83.25 |
| OC+TEA | 20.33 (+0.03) | 32.02 (+0.41) | 80.64 (+2.06) | 49.37 (+3.78) | 90.50 (+4.25) | 86.75 (+3.50) |
| *Relationship - A <Carved by> B* | | | | | | |
| RV | 25.64 | 27.74 | N/A | N/A | N/A | N/A |
| RV+TEA | 27.84 (+2.20) | 30.17 (+2.43) | N/A | N/A | N/A | N/A |
| *Relationship - A <Inside> B* | | | | | | |
| RV | 24.97 | 27.87 | N/A | N/A | N/A | N/A |
| RV+TEA | 25.15 (+0.18) | 28.40 (+0.53) | N/A | N/A | N/A | N/A |
| *Relationship - A <Painted on> B* | | | | | | |
| RV | 23.98 | 30.07 | N/A | N/A | N/A | N/A |
| RV+TEA | 24.38 (+0.40) | 30.35 (+0.28) | N/A | N/A | N/A | N/A |

We evaluate the effectiveness of Test-time Embedding Adjustment (TEA) when combined with various personalization baselines, including Textual Inversion (TI) (Gal et al., 2022), DreamBooth (DB) (Ruiz et al., 2023), CustomDiffusion (CD) (Kumari et al., 2023), and ClassDiffusion (CL) (Huang et al., 2024a). As shown in Table 2, TEA consistently enhances full prompt alignment ($\text{CLIP}_T^f$) across all methods and datasets. Gains are particularly notable for TI and DB, with increases up to +1.87 on CC101, demonstrating that TEA substantially strengthens text–image consistency. Importantly, these improvements hold across diverse concepts (Figure 6), indicating that TEA is robust to different personalization scenarios, even with fixed, easily chosen hyper-parameters.

In terms of visual quality, TEA often improves CLIP-I and DINO-I scores (e.g., +4.57 CLIP-I for DB on CC101), while in some cases—particularly on CelebA—there is a trade-off, with modest drops in CLIP-I (e.g., –2.37 for DB+TEA).

Table 2: Improvement of our TEA over its baseline counterparts on CC101 and CelebA datasets (positive or negative). Details are shown in Tables 6 and 7.

| Method | $\text{CLIP}_T^p$ | $\text{CLIP}_T^f$ | CLIP-I | DINO-I |
|---|---|---|---|---|
| **CC101** | | | | |
| TI+TEA | 0.55 | 0.64 | -2.34 | -3.81 |
| DB+TEA | 0.77 | 1.87 | 4.57 | 0.59 |
| CD+TEA | -0.13 | 0.34 | 0.37 | 0.24 |
| CL+TEA | -0.12 | 0.42 | 0.67 | 1.24 |
| **CelebA** | | | | |
| TI+TEA | 0.33 | 0.57 | -2.41 | -1.78 |
| DB+TEA | 0.51 | 0.64 | -2.37 | -2.27 |
| CD+TEA | -0.12 | 0.09 | 1.56 | 3.09 |
| CL+TEA | 0.22 | 0.39 | -0.69 | 0.17 |

However, qualitative comparisons (Figures 24, 23), particularly in Figure 22, show that TEA reduces distortions and produces more coherent, realistic outputs, even when quantitative scores dip slightly. This suggests TEA better preserves semantic fidelity while mitigating common artifacts in baseline methods.

**Generality across architectures and use cases.** We further assess TEA on three state-of-the-art frameworks beyond the original baselines: EasyControl (Zhang et al., 2025), OminiControl (Tan et al., 2025), two Flux-based personalization methods, and ReVersion (Huang et al., 2024b), which targets compositional relationships, e.g., 'a cat carved by a carrot'. As shown in Table 1, TEA again yields consistent improvements. For EasyControl, TEA achieves substantial gains in image–image alignment (+3.23 CLIP-I, +4.61 DINO-I, +3.25 VLM-I), while for ReVersion it strengthens prompt fidelity, with up to +2.43 $\text{CLIP}_T^f$ in complex relations such as 'carved by'. For OminiControl, TEA

improves VLM-P by 4.25 points and VLM-I by 3.50 points, as well as by 0.41 points in $CLIP_T^f$ and 2.06 points in $CLIP_I$, demonstrating the consistent improvement across different metrics.

Qualitative results (Figures 28, 29, 31) confirm these findings: While producing impressive high-quality generations, these SOTA frameworks still suffer from failure cases such as EasyControl producing spurious objects or unsafe generations, OminiControl failing to control the subject's action, and ReVersion failing to accurately represent relationships. Our TEA corrects these failure cases effectively, significantly improving the prompt fidelity of these frameworks without compromising the image-reference fidelity. Again, all these results are obtained in inference time without any additional training or finetuning.

Together, these results highlight TEA as a lightweight yet broadly applicable enhancement to personalization. It systematically improves text–image alignment, generalizes across methods, datasets, and architectures, and provides qualitative corrections to baseline failure modes, establishing TEA as a robust and versatile component for personalization.

**Surprising Impact of TEA on Anti-DreamBooth.** Anti-personalization (Liang et al., 2023; Van Le et al., 2023; Salman et al., 2023) aims to protect users from malicious actors who might exploit personal images to train unauthorized personalized models. The core idea is to apply an invisible perturbation (a 'mask') to the user's data before it is shared. Although attackers can still access these masked images, they are prevented from training effective personalized models. One representative approach is Anti-DreamBooth (Van Le et al., 2023), which employs adversarial learning (Szegedy et al., 2013; Goodfellow et al., 2014) to generate such masks.

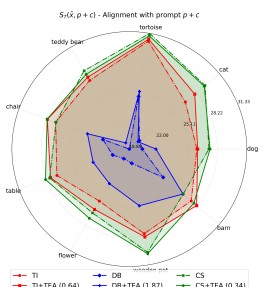

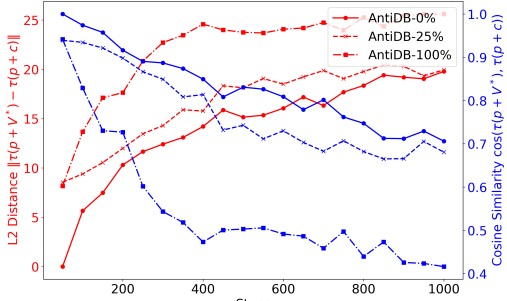

Figure 6: Comparison on prompt alignment of our TEA over its baselines counterpart on CC101 dataset. Refer to Table 6 for detailed numbers.

Figure 7: Semantic drift analysis of Dream-Booth trained with Anti-DreamBooth adversarial masks.

Given that background, we analyze Anti-DreamBooth through the lens of SCP. We hypothesize that its **adversarial learning process actually amplifies the dominance of the personalized concept** $V^*$, but with good implications for user privacy, by causing the prompt embedding $\lfloor p, V^* \rfloor$ to drift even further from its original concept $\lfloor p, c \rfloor$, resulting to distorted generations of the protected concept $V^*$.

To verify this hypothesis, we conduct a controlled experiment. Given a set of benign personal images $\{x_i\}_{i=1}^n$, we apply Anti-DreamBooth to produce masked images $\{\hat{x}_i\}_{i=1}^n$. We then train DreamBooth models using mixtures of masked and benign images, with varying proportions $p \in \{0, 0.2, 1.0\}$, where $p = 0$ corresponds to standard DreamBooth, and $p = 1.0$ uses only masked data. We then analyze the resulting prompt embeddings as in Section 2.3.3. As shown in Figure 7, increasing $p$ leads to greater embedding drift, evident in the larger norm of $\|\tau(\lfloor p, V^* \rfloor) - \tau(\lfloor p, c \rfloor)\|$ and the lower cosine similarity between $\tau(\lfloor p, V^* \rfloor)$ and $\tau(\lfloor p, c \rfloor)$. This result confirms our hypothesis and provides an interesting perspective on why anti-personalization works.

Surprisingly, when we apply TEA to DreamBooth models poisoned by Anti-DreamBooth, we observe a mitigation effect such that the generated images by TEA are less distorted and more aligned with the to-be-protected concept $V^*$ as shown in Figure 8 (more results can be found in Appendix C.2). This surprising result reveals an intriguing false sense of security of Anti-DreamBooth, such that despite adversarial masking, **the poisoned personalized model still retains traces of the correct/to-be-protected concept** $V^*$ and the distortion of the generated images is just the consequence of the **extreme** Semantic Collapsing Problem magnified by the adversarial masking. The distorted

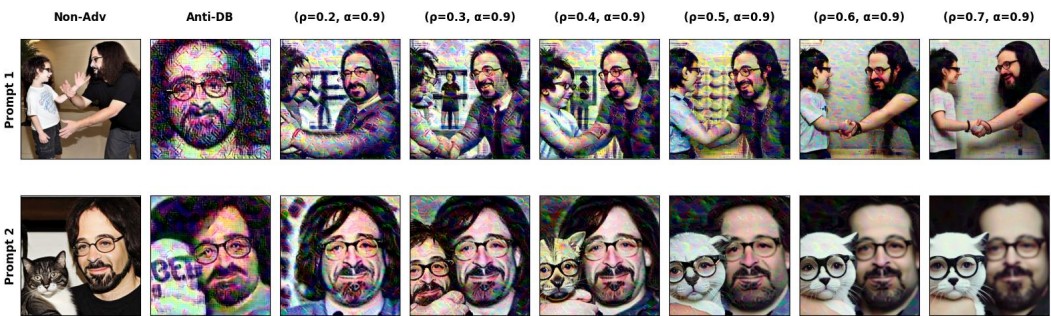

Figure 8: Effect of applying TEA to models poisoned by Anti-DreamBooth. TEA is able to mitigate the corruption and recover less distorted generations of the protected concept, revealing a surprising weakness in Anti-DreamBooth. Additional results and discussion can be found in Appendix C.2.

generations bring the false sense of security because the attacker can use TEA to recover partially the to-be-protected concept $V^*$. To the best of our knowledge, this is the first work to uncover such a counter-intuitive vulnerability of Anti-DreamBooth and sheds new light on the limitations of current anti-personalization defenses.

## 5 CONCLUSION

In this paper, we identified the *Semantic Collapsing Problem* (SCP) in generative personalization, where personalized tokens lose their original semantic meaning and dominate other concepts in complex prompts. We traced this issue to unconstrained optimisation, which allows the learned token embedding to drift in direction and magnitude, disrupting prompt interpretation.

To address this, we proposed a training-free test-time embedding adjustment (TEA) that realigns the personalized embedding with its original semantic context, significantly improving text–image alignment without modifying model weights. Our method is lightweight, broadly compatible with almost all existing personalization frameworks such as Textual Inversion and DreamBooth and their variants, and delivers substantial improvements in prompt consistency and image fidelity across diverse scenarios.

In addition to tackling SCP, we also provided an initial probe into the interaction between TEA and anti-personalization. Surprisingly, when applied to models corrupted by Anti-DreamBooth, TEA partially mitigates adversarial corruption and recovers more faithful generations of the protected concept. This finding suggests that current defenses may offer a false sense of security, opening an intriguing direction for future work at the intersection of personalization and privacy protection.

Overall, beyond introducing SCP and proposing a practical solution, this work lays the foundation for further exploration of adaptive embedding adjustments, context-aware constraints during personalization, and new perspectives on the robustness of anti-personalization methods.

## ACKNOWLEDGEMENT

This work was supported by the Australian Defence Science and Technology (DST) Group through the Next Generation Technology Fund (NGTF) scheme and the Department of Defence, Australia, via the Advanced Strategic Capabilities Accelerator (ASCA) program. Dinh Phung further acknowledged the support from the Australian Research Council (ARC) Discovery Project DP230101176 and DP250100262. The authors are grateful to the anonymous reviewers for their helpful comments. This research/work was supported by Monash eResearch capabilities, including M3.

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

# Appendix

## Table of Contents

## STATEMENT ON THE USE OF LARGE LANGUAGE MODELS

We utilized Large Language Models (LLMs) in this work for two primary purposes. First, we employed LLMs like ChatGPT to correct grammatical errors and enhance the manuscript's clarity. Second, we leveraged these models to generate diverse context sets to vefiry the hypothesis of SCP in the main text. The instructions for the LLMs are provided in Appendix F.1

## A   RELATED WORK

### A.1   DIFFUSION MODELS

Given a text-to-image diffusion model $\epsilon_\theta$, where $\epsilon_\theta(x_t, t, p)$ represents the predicted noise at time step $t$ given the textual embedding $\tau(p)$ of a prompt $p$ and the noisy intermediate vector $x_t$ (Ho et al., 2020; Song et al., 2020; Rombach et al., 2022), the model is trained by minimizing the following objective:

$$\mathcal{L} = \mathbb{E}_{(x,p)\sim p_{\text{data}}, t\sim \mathcal{U}[0,T], \epsilon\sim \mathcal{N}(0,\mathbf{I})} \left[ \left\| \epsilon - \epsilon_\theta(\alpha_t x + \sqrt{1-\alpha_t}\epsilon, t, p) \right\|_2^2 \right] \tag{3}$$

Here, $x$ and $p$ denote the input image and its associated prompt, respectively, while $\epsilon$ is the Gaussian noise sampled from a standard normal distribution. The intermediate input $x_{t,\epsilon} = \alpha_t x + \sqrt{1-\alpha_t}\epsilon$ is obtained from the forward diffusion process. For simplicity, we use the notation $\mathbb{E}_{x,p,t,\epsilon}[\cdot]$ to represent the expectation over the input data $x$, the prompt $p$, the diffusion time step $t$, and noise $\epsilon$.

### A.2   GENERATIVE PERSONALIZATION.

Generative personalization task aims to capture personal concepts which are implicitly shared in a reference set of images as generative conditions and then use them as a guided condition to generate new images containing the personal concept. These personal concepts are very difficult to express

in the input prompt, e.g., how to express the concept of your dog that is different from a generic dog. Therefore, rather than using prompt engineering techniques to describe the concept in text, this task usually uses a gradient-based method to fine-tune the model parameters to capture the personal concept. There are several categories of generative personalization (Cao et al., 2024) classified based on the type of generative conditions, such as subject-driven (Gal et al., 2022; Ruiz et al., 2023; Chen et al., 2022; Kumari et al., 2023; Wang et al., 2024), person-driven (Xiao et al., 2024; Valevski et al., 2023; Chen et al., 2024b; 2023b), style-driven (Sohn et al., 2023; Liu et al., 2023; Chen et al., 2024a) or image-driven (Ramesh et al., 2022; Xu et al., 2023; 2024). Personalizing a T2I diffusion model from only a few examples presents several well-known challenges, including language drift, limited expressiveness of generative conditions, entanglement of concepts, and conditional misalignment.

**Textual Inversion and Dreambooth.** While there are many personalization methods have been proposed, they can be traced back to the two representative methods: Textual Inversion (TI) (Gal et al., 2022) and DreamBooth (Ruiz et al., 2023). Mathematically, given a set of personal images $\mathcal{X} = \{x_1, x_2, \cdots, x_n\}$ and a pre-trained T2I model $\epsilon_\theta$, the goal is to identify a textual embedding $v^*$ associated with a specific verbalizable keyword $V^*$ (e.g., 'sks', '<new>', etc.). This keyword represents the implicit visual concept shared in the reference set $\mathcal{X}$, enabling the model to generate images with the personal concept using any textual prompt $p$ containing the keyword $V^*$, e.g., $\lfloor p, V^* \rfloor =$ 'A photo of $V^*$ playing on a beach', where $\lfloor ., . \rfloor$ is the sentence construction operator.

Textual Inversion (TI) (Gal et al., 2022) is a pioneering method that proposes obtaining a textual embedding by minimising the following objective:

$$\min_{v^*} \ \mathbb{E}_{x,p,\epsilon,t} \left[ \|\epsilon - \epsilon_\theta\left(x_{t,\epsilon}, t, \lfloor p, V^* \rfloor\right)\|_2^2 \right] \tag{4}$$

Here, $p \sim \mathcal{T}$ is a template prompt sampled from a set of predefined neutral prompts $\mathcal{T}$, such as $\{$'a photo of a', 'a high-quality photo of a', $\cdots\}$. In TI, only the embedding $v^*$ is learned, while all other parameters, such as the model $\epsilon_\theta$ or textual encoder $\tau$, remain fixed. Although this method is parameter-efficient, the learned embedding may not be sufficiently representative to capture the true visual concept in the reference set. Building on TI, Dreambooth (Ruiz et al., 2023) suggests fine-tuning not only the embedding $v^*$ (i.e., by fine-tuning the textual encoder $\tau$) but also the model parameters $\theta$ by minimizing the following objective:

$$\min_{\theta,v^*} \ \mathbb{E}_{x,p,x_{pr},p_{pr},\epsilon,\epsilon',t} \left[ \|\epsilon - \epsilon_\theta\left(x_{t,\epsilon}, t, \lfloor p, V^* \rfloor\right)\|_2^2 + \lambda \left\| \epsilon' - \epsilon_\theta\left(x_{t',\epsilon'}^{pr}, t', p^{pr}\right) \right\|_2^2 \right] \tag{5}$$

In this context, $x^{pr}$ and $p^{pr}$ are the prior-preservation image and its associated prompt, respectively, which help prevent the model from overfitting to the small reference set with a tradeoff hyper-parameter $\lambda$.

**Anti-Personalization.** The rapid progress of text-to-image models such as Stable Diffusion (Rombach et al., 2022) and DALL·E (Ramesh et al., 2021), trained on massive web-scale datasets, also brings risks of malicious misuse. These models have already been exploited to generate harmful, biased, or infringing content (Somepalli et al., 2023; Carlini et al., 2023). While such risks first surfaced with celebrities—whose data are abundantly available online—the rise of generative personalization means that now anyone's likeness can be misused, since high-quality personalized models can be trained from only a handful of images.

Mitigation strategies span both developer-side and user-side defenses. On the developer side, model providers aim to proactively remove or suppress harmful capabilities before public release. This can be achieved through machine unlearning, which erases targeted concepts from the model (Gandikota et al., 2023; Schramowski et al., 2023; Bui et al., 2024; 2025b), or through parameter-editing methods that hide unintended capabilities without retraining (Bui et al., 2025a).

On the user side, anti-personalization methods (Liang et al., 2023; Van Le et al., 2023; Salman et al., 2023) protect individuals from unauthorized personalization by perturbing their images before sharing. These methods apply imperceptible "masks" so that, even if adversaries access the data, they cannot train effective personalized models. A prominent example is Anti-DreamBooth (Van Le et al., 2023), which leverages adversarial learning (Szegedy et al., 2013; Goodfellow et al., 2014; Bui

et al., 2022) to craft such masks. Formally, the optimization seeks perturbations $\delta^{(i)}$ that maximize a conditional loss $\mathcal{L}_{cond}$ while simultaneously degrading DreamBooth's training objective $\mathcal{L}_{db}$ under bounded distortion:

$$
\begin{aligned}
\delta^{*(i)} &= \operatorname{argmax}_{\delta^{(i)}} \mathcal{L}_{cond}(\theta^*, x^{(i)} + \delta^{(i)}), \forall i \in \{1, .., N_{db}\}, \\
\text{s.t.} \quad \theta^* &= \operatorname{argmin}_\theta \sum_{i=1}^{N_{db}} \mathcal{L}_{db}(\theta, x^{(i)} + \delta^{(i)}), \\
\text{and} \quad \|\delta^{(i)}\|_p &\leq \eta \quad \forall i \in \{1, .., N_{db}\},
\end{aligned}
\tag{6}
$$

For instance, the conditional loss can be defined via reconstruction and prior losses:

$$
\mathbb{E}_{\mathbf{x}, \mathbf{c}, \epsilon, \epsilon', t} \left[ \underbrace{w_t \|\hat{\mathbf{x}}_\theta(\alpha_t \mathbf{x} + \sigma_t \epsilon, \mathbf{c}) - \mathbf{x}\|_2^2}_{\mathcal{L}_{recon}} + \lambda \underbrace{w_{t'} \|\hat{\mathbf{x}}_\theta(\alpha_{t'} \mathbf{x}_{\mathrm{pr}} + \sigma_{t'} \epsilon', \mathbf{c}_{\mathrm{pr}}) - \mathbf{x}_{\mathrm{pr}}\|_2^2}_{\mathcal{L}_{prior}} \right]
\tag{7}
$$

### A.3 CHALLENGES IN GENERATIVE PERSONALIZATION

**Language Drift and Overfitting.** Language drift or overfitting occurs due to the limited number of reference images and results in the incorporation of irrelevant elements and the neglect of the textual context within the outputs. Prior works address this issue by introducing preservation mechanisms such as prior-preservation loss (Ruiz et al., 2023), locking concept-specific parameters (Tewel et al., 2023) and regularising model weights (Han et al., 2023; Qiu et al., 2023).

**Limited Expressiveness of Generative Conditions.** This occurs due to the limited expressiveness of the original textual format and the limited number of tokens allowed in each condition. A common mitigation approach is to use multi-modal conditions such as image-image and sketch-image. To enable pre-trained T2I models to accept new types of conditions and generate in conjunction with the current text prompt conditioning, previous works have attempted to incorporate an additional encoder (Zhang et al., 2023), or add a new adapter module to align internal knowledge of the model with the new condition (Mou et al., 2024; Jiang et al., 2024).

**Entanglement of Concepts.** The reference image set might include samples that contain both the intended concept and other irrelevant concepts. To effectively isolate and extract the intended concept from the reference set, previous works have employed explicit masks (Avrahami et al., 2023; Jin et al., 2024; Safaee et al., 2024) and additional data with the personalized concept (Li et al., 2023). Alternatively, Disenbooth (Chen et al., 2023a) proposed to mitigate the influence of background elements in the reference set by disentangling the identity and background of the reference set.

**Conditional Misalignment.** Beyond overfitting and entanglement, generative personalization faces the broader challenge of conditional misalignment where outputs deviate from the intended prompt due to the limited alignment capacity of the original generative model. The trade-off between identity fidelity and semantic fidelity is a well-known and fundamental challenge in generative personalization. Most existing works address this as an overfitting issue during training and propose various regularization strategies, which typically require modifying the training process or adding supervision. PALP (Arar et al., 2024) introduces score distillation sampling to explicitly regularize the learned token toward its original class concept, preventing semantic drift. However, it operates with a single fixed prompts and requires training-time modification. LEGO (Motamed et al., 2024) and ReVersion (Huang et al., 2024b) aim to disentangle compositional concepts (e.g., adjectives, verbs, or relationships) from exemplar images using token-based personalization. However, they are limited to token-based personalization model only.

A largely unaddressed gap in prior personalization work is the potential drift or misalignment of the textual embedding itself during concept learning. We refer this phenomenon as *semantic collapse* where the learned concept token is still faithful to the visual reference, but fails to retain any meaningful textual semantics and eventually collapses to a simplified form. We directly address the

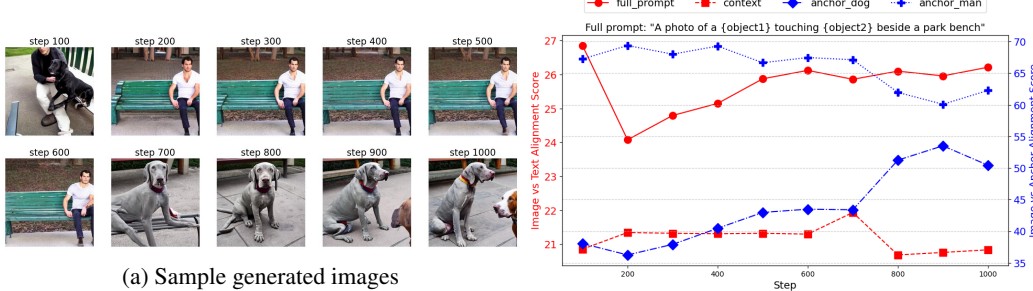

(a) Sample generated images

(b) Alignment scores (average over 50 random runs)

Figure 9: Analysis the SCP on multi-concepts personalization. The embedding of $V_{man}^*$ **is fixed** while the embedding of $V_{dog}^*$ **is varied** across different training steps. Prompt: 'A photo of a $V_{man}^*$ touching a $V_{dog}^*$ beside a park bench'. See Figure 26 and 27 for more examples.

semantic drift of the learned embedding. Our method mitigates this drift without altering the training pipeline or requiring additional training. As a training-free, plug-and-play solution, our method can be seamlessly integrated into a wide range of existing personalization frameworks.

## B    SCP ON MULTI-CONCEPTS PERSONALIZATION

In this section, we investigate the question: *What is the SCP in the context of multi-concept personalization?* For instance, consider a prompt like 'A photo of a $V_{man}^*$ touching a $V_{dog}^*$ beside a park bench', where $V_{man}^*$ and $V_{dog}^*$ represent two independently personalized concepts. Will one concept dominate the other, as observed in single-concept personalization in the previous section?

To explore this, we conduct an experiment where the two concepts, $V_{dog}^*$ and $V_{man}^*$ (subject 342), are learned independently using Textual Inversion. We then construct a list of prompts that combine these two personalized concepts with an additional complex context, such as 'A photo of a $V_{man}^*$ touching a $V_{dog}^*$ beside a park bench' (refer to Table 4 for more examples).

Figure 9 illustrates the generated images where the embedding of $V_{man}^*$ is held fixed, while the embedding of $V_{dog,step}^*$ is varied across different training steps. At early steps, $V_{dog,step}^*$ remains close to the original, generic 'dog' concept, while at later steps, it progressively captures more personalized visual information of the specific dog. This design allows us to observe how changes in the $V_{dog}^*$ embedding influence the generation of $V_{man}^*$ within the same prompt.

It can be observed from Figure 9a that at early steps (e.g., < 400), $V_{man}^*$ tends to dominate the prompt, resulting in images that primarily capture the 'man' concept, consistent with the SCP observed in single-concept settings. However, as the training progresses and $V_{dog,step}^*$ captures more distinctive features of the personalized dog concept, it begins to overshadow $V_{man}^*$, leading to outputs that predominantly depict only the dog, effectively suppressing the presence of the other concept. As shown in Figure 9b, the alignment score with anchor man image drops gradually while that with anchor dog image increases over time further confirming the dominance of $V_{dog}^*$ over $V_{man}^*$.

This trend, observed consistently across multiple settings as shown in Figure 26 and 27, highlights the intricate nature of SCP in multi-concept personalization. It suggests that SCP not only persists but can intensify when multiple personalized concepts are involved, presenting a challenging but potentially fruitful direction for future research.

## C    FURTHER QUANTITATIVE RESULTS

In this section, we present additional quantitative and qualitative results that further validate our findings and broaden the scope of analysis. Specifically, we include:

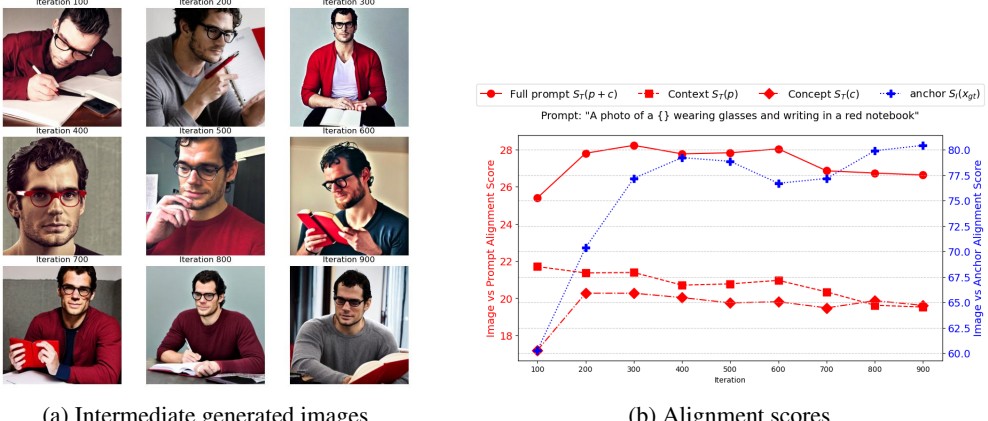

| (a) Intermediate generated images | (b) Alignment scores |

Figure 10: Illustration of the SCP on Textual Inversion. Left: The intermediate generated images $\hat{x}$ of a prompt 'a photo of a $V^*$ wearing glasses and writing in a red notebook'. The generated image is gradually biased towards the personalized concept $V^*$ (i.e., easier to recognize as 'Henry Cavill') and loses the context $p$ (i.e., harder to recognize as 'writing in a red notebook') through out the personalization process. Right: The alignment scores (average over 100 random seeds) which empirically validate the SCP. The alignment $S(\hat{x}, p)$ ($\square$) with the context $p$ drops over time, while the alignment $S(\hat{x}, x_{gt})$ (+) with the ground truth $x_{gt}$ increases.

Figure 11: Analysis of the effect of rotation factor.

| Method | $\text{CLIP}_T^c$ | $\text{CLIP}_T^p$ | $\text{CLIP}_T^f$ | CLIP-I |
|---|---|---|---|---|
| TI | $17.4 \pm 1.8$ | $21.4 \pm 1.1$ | $26.5 \pm 1.8$ | $73.5 \pm 4.7$ |
| $\alpha = 0.20$ | $17.8 \pm 1.9$ | $21.7 \pm 1.1$ | $27.5 \pm 1.9$ | $71.1 \pm 5.7$ |
| $\alpha = 0.25$ | $18.0 \pm 1.9$ | $21.8 \pm 1.0$ | $27.8 \pm 1.9$ | $69.6 \pm 5.7$ |
| $\alpha = 0.30$ | $18.0 \pm 2.0$ | $21.9 \pm 1.1$ | $28.0 \pm 1.9$ | $67.3 \pm 5.8$ |
| $\alpha = 0.35$ | $18.2 \pm 2.0$ | $21.9 \pm 1.1$ | $28.2 \pm 1.8$ | $63.6 \pm 5.3$ |

- **Hyper-parameter sensitivity** (Section C.1), showing how different settings affect TEA's performance.

- **Unexpected effects on Anti-DreamBooth** (Section C.2), where TEA reveals new insights into mitigating distortions in unlearning scenarios.

- **Textual evidence of semantic collapse** (Figure 17), where we measure embedding drift using multiple distance metrics.

- **Visual evidence of collapse across methods** (Figures 18, 19), confirming SCP in both Textual Inversion and DreamBooth.

- **Prompt-level embedding drift** (Figure 21), providing direct evidence that SCP is not confined to TI variants but also affects DreamBooth and its extensions, consistent with our argument that unconstrained optimization underlies the problem (Section 2.3.3).

- **Detailed quantitative comparisons** (Tables 6, 7) on CustomConcept101 and CelebA, offering a comprehensive view of TEA's improvements over multiple baselines.

Together, these results not only strengthen our main claims but also uncover broader implications—most notably, that SCP and TEA's corrective effect are general phenomena spanning datasets, frameworks, and experimental setups.

## C.1 The Effect of Hyper-parameters

An important question is how to choose the hyper-parameters $\alpha$ and $\beta$ appropriately, or whether they should be adapted based on the input prompt. Interestingly, our experiments reveal a clear pattern in the performance of the proposed method across a range of $\alpha$ and $\beta$ values, providing practical guidance for their selection. For simplicity and consistency, we set $\alpha = 0.2$ and $\beta = 1.5$ as default values, which consistently deliver robust performance across diverse prompts.

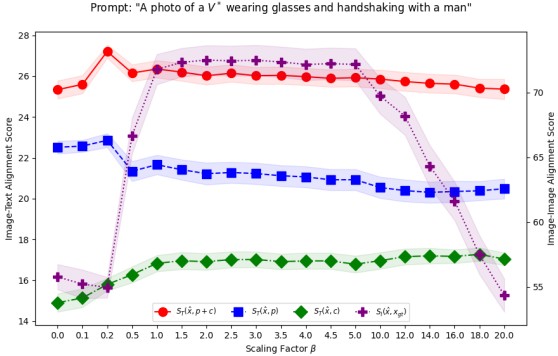

Figure 12: Analysis of the effect of scaling factor.

Figure 12 shows the performance of our adjustment method over a range of $\beta$ values with fixed $\alpha = 1.0$ (no rotation). It can be seen that when $\beta$ is too small (i.e., $\beta < 0.5$, meaning $\|\hat{M}_{V^*}\| < 0.5 \|M_c\|$) or too large (i.e., $\beta > 5.0$, meaning $\|\hat{M}_{V^*}\| > 5.0 \|M_c\|$), the generated images become less aligned with the ground truth indicating by the significant drop in $S_I(\hat{x}, x_{gt})$, suggesting that $V^*$ has lost its personalized information. However, the alignment $S_I(\hat{x}, x_{gt})$ is relatively stable when $\beta$ is in the range of $[1.0, 5.0]$, suggesting that the personalized concept can be effectively captured without extreme scaling $V^*$. As a practical choice, we simply set $\beta = 1.5 \approx \frac{\|M_{V^*}\| + \|M_c\|}{2\|M_c\|}$ (which is a middle value interpolated from $\|M_c\|$ to $\|M_{V^*}\|$), as the default setting.

Table 11 shows the performance over a range of $\alpha$ values with fixed $\beta = 1.5$. Unlike the scaling parameter, the rotation factor $\alpha$ is more sensitive and significantly impacts prompt alignment capability. Increasing $\alpha$ generally improves the model's ability to capture context, as reflected in the $\text{CLIP}_T^f$ score, which increases from 26.5 to 28.2 (an improvement of 1.7), and the $\text{CLIP}_T^p$ score, which rises from 21.4 to 21.9 (a gain of 0.5). While this comes at a minor cost to visual fidelity, with the CLIP-I score dropping from 73.5 to 71.1 when $\alpha = 0.2$, the generated images still maintain high visual quality, as shown in Figures 23 and 24.

These findings highlight the critical role of the rotation factor $\alpha$, which directly controls the semantic alignment between $V^*$ and the target concept $c$. Higher $\alpha$ values encourage better prompt alignment by rotating $V^*$ closer to $c$, while the scaling factor $\beta$ should remain within a moderate range to prevent excessive distortion of the learned visual concept.

## C.2 SURPRISING IMPACT OF TEA ON ANTI-DREAMBOOTH

In this section, we present additional results highlighting the surprising impact of TEA on Anti-DreamBooth. Figures 13 and 14 analyze the effect of varying the rotation factor $\rho$ and the scaling factor $\alpha$, respectively.

Interestingly, while scaling has shown its effectiveness in mitigating the SCP in standard personalization settings as discussed in Section C.1, when applied to Anti-DreamBooth, it does little to correct the visual distortion in generated images. By contrast, adjusting the rotation factor $\rho$ shows a pronounced effect such that the generated images become substantially less distorted and better aligned with the protected concept $V^*$ as shown in Figure 13. This contrast suggests that the **semantic misalignment in anti-personalization settings is more sensitive to directional shift than to embedding magnitude**, and that rotation-based corrections are inherently more effective than scaling-based adjustments.

We view this as a valuable and unexpected finding that not only deepens our understanding of why Anti-DreamBooth works (by amplifying the semantic collapse), but also points to promising directions for future work such as geometric interventions or adaptive adjustments during the defense process.

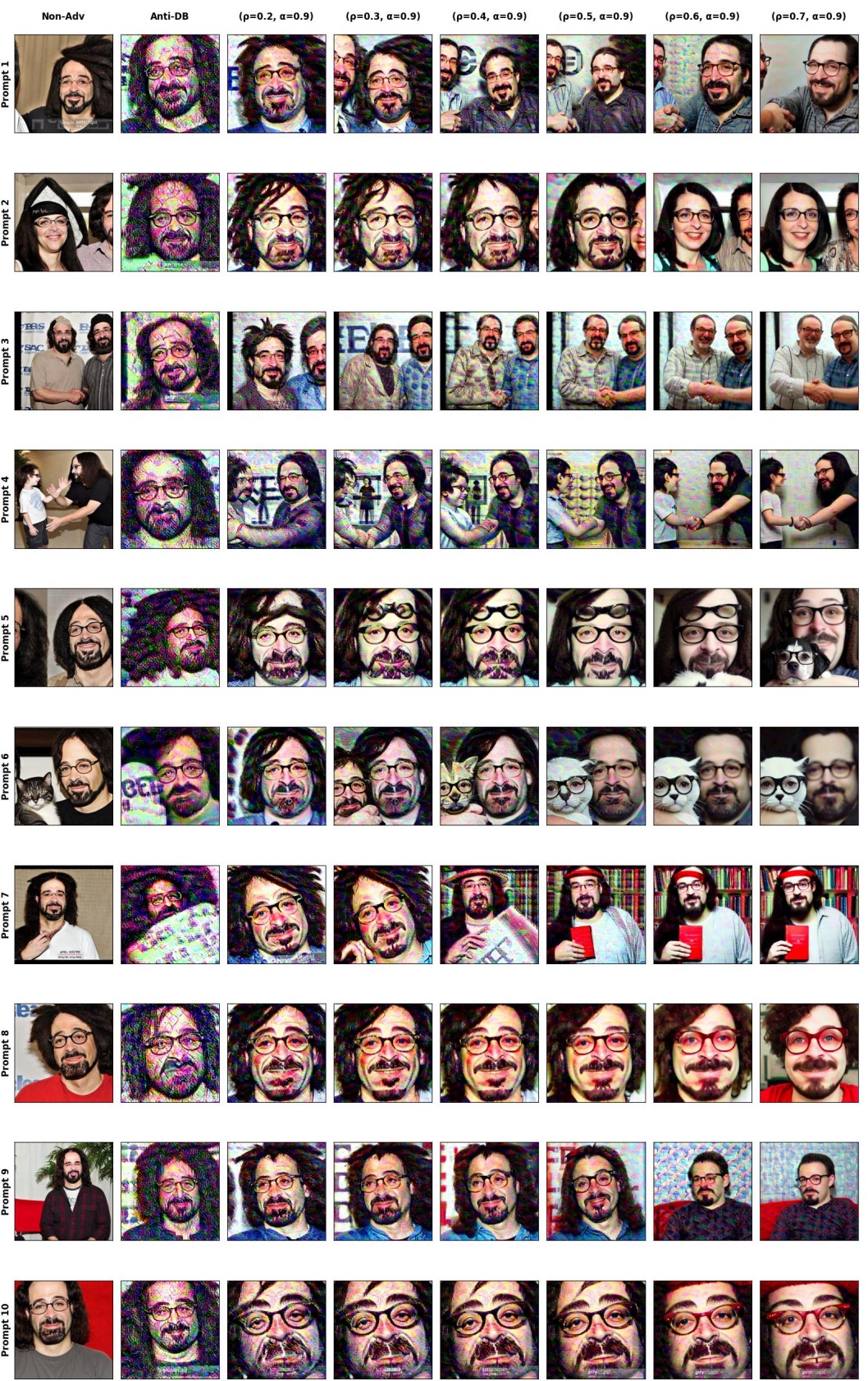

Figure 13: Applying TEA on mitigating the Anti-Personalization Effect of DreamBooth by varying the rotation factor $\rho$

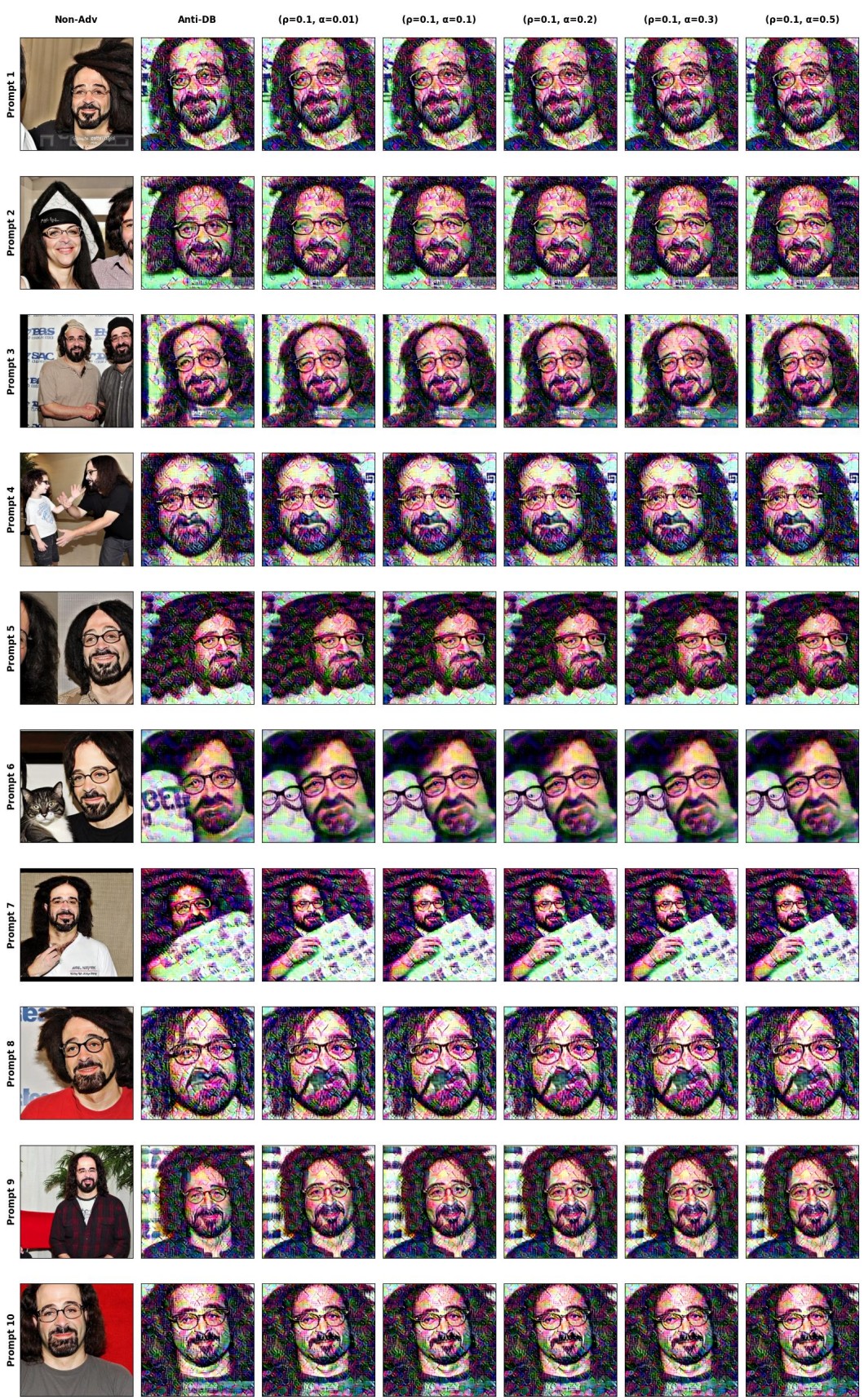

Figure 14: Applying TEA on mitigating the Anti-Personalization Effect of DreamBooth by varying the scaling factor $\alpha$

# D    QUALITATIVE RESULTS

We provide additional qualitative results to complement the quantitative analysis:

- **Correcting distorted generations.** Figure 22 illustrates how SCP in DreamBooth leads to distorted generations, while TEA corrects the semantic embedding to produce more coherent and realistic images. This connects directly to the discussion on TEA's unexpected impact on Anti-DreamBooth (Section C.2).
- **Cross-dataset comparisons.** Figures 24 and 23 compare TEA-augmented variants of Textual Inversion, DreamBooth, and Custom Diffusion against their baselines on CS101 and CelebA. Further comparisons on Subject Control and Relationship tasks are shown in Figures 28, 29, 30, and 31, highlighting TEA's robustness across frameworks.
- **SCP in multi-concept prompts.** Figures 26 and 27 demonstrate SCP when multiple concepts are combined, with corresponding alignment scores provided in Figure 25.

# E    LIMITATIONS AND FUTURE WORK

**Limitations on Methodology.**    We believe that the insights and understanding provided by this paper, especially the analysis of the Semantic Collapsing Problem, are the most important contributions. Based on this analysis, we propose a simple method to adjust the embedding vectors at test time. While this method has clear advantages, such as simplicity, generalizability, and no additional training requirement, the simplicity of the method itself might be a limitation. There are still trade-offs between alignment with the input prompt and alignment with the visual concept, which depend on the hyper-parameters, suggesting that fixed hyper-parameters might not be optimal for all prompts.

**Future Work.**    We believe that the insights and understanding provided by this paper, especially the analysis of the Semantic Collapsing Problem, can guide future research on this topic. For example, integrating additional constraints to restrict semantic shift during the fine-tuning phase, rather than relying solely on test-time adjustment, is a promising direction. This approach could directly produce adjusted and bounded embedding vectors that retain the original semantic meaning of the base concept.

In this work, we provide a simple method to adjust the embedding vectors at test time. This adjustment is applied equally across all dimensions of the embedding vectors. However, we believe that this is not the optimal way to adjust embedding vectors, as each dimension of the embedding vector has different meanings and importance. Therefore, a more sophisticated method that considers the importance of each dimension could also be a promising direction.

In Section C.1, we provide an analysis of the impact of hyper-parameters on the performance of the proposed method. It has been shown that the performance of the proposed method is sensitive to the rotation factor $\alpha$: the larger the $\alpha$, the more the embedding vector is rotated toward the target concept, improving alignment with the generated images but potentially reducing alignment with the visual concept. In this work, we simply use the same hyper-parameters for all settings. We believe that a search algorithm could be applied to find the optimal hyper-parameters for each prompt, with a stop condition based on the desired alignment with the input prompt.

# F    EXPERIMENTAL SETTING

## F.1    DATASET CONSTRUCTION

**Contextual Prompts for Measuring Semantic Collapsing.**    Recall our hypothesis: a personalized keyword $V^*$, initialized from a base concept $c$ to capture a specific visual target $v_{gt}$, tends to lose its original semantic meaning and dominate arbitrary contexts $p$ when used in complex prompts. This phenomenon, which we refer to as *semantic shift*, can be directly assessed by comparing the embedding vectors $M_{V^*}$ and $M_c$. However, this is challenging due to the use of contextualized text embeddings in modern LLMs and diffusion models, where the surrounding context significantly shapes the final representation of each token.

Table 3: Sample sentences for $P_{V^*}$ and $P_{V^*}^{\text{simple}}$. Set $P_c$ can be constructed by replacing the word $c$ with $V^*$ in the prompt of $P$. All the data and prompts can be found in the repository.

| Set | Sample Sentences |
|---|---|
| $P_{V^*}$ | 'A $V^*$ walked his dog through the park every morning before sunrise' |
| | 'Despite the heavy rain, a $V^*$ stood patiently waiting for the bus' |
| | 'In the small village, a $V^*$ known for his kindness helped everyone' |
| | 'After twenty years of dedicated service, a $V^*$ retired from his factory job' |
| | 'While climbing Mount Everest, a $V^*$ discovered the true meaning of perseverance' |
| | 'During the concert, a $V^*$ in the front row sang along to every song' |
| | 'At the crowded marketplace, a $V^*$ sold handcrafted jewelry made from local materials' |
| | 'Throughout history, a $V^*$ with vision has often changed the course of events' |
| | 'Behind every successful company, there is often a $V^*$ with an innovative idea' |
| | 'Within the ancient temple, a $V^*$ prayed silently for his family's wellbeing' |
| $P_{V^*}^{\text{simple}}$ | 'A photo of a $V^*$' |
| | 'A rendering of a $V^*$' |
| | 'A cropped photo of a $V^*$' |
| | 'A portrait of a $V^*$' |
| | 'A close-up shot of a $V^*$' |
| | 'A full-body image of a $V^*$' |
| | 'A black-and-white photograph of a $V^*$' |
| | 'A candid shot of a $V^*$' |
| | 'A digital illustration of a $V^*$' |
| | 'A stylized caricature of a $V^*$' |

To address this, we propose evaluating the semantic shift of $V^*$ relative to $c$ in the presence of a diverse set of contextual prompts. Specifically, we define a prompt set $A = \{a_1, a_2, \cdots, a_n\}$, constructed by querying a large language model (LLM) with the following instruction:

```
Write 200 sentences with diverse topics and contents.
Each sentence should be 10-30 words long and must
include the keyword c.
```

This approach allows us to measure how well the learned embedding $M_{V^*}$ retains the original semantic characteristics of $c$ across varied contexts, providing a robust test for the semantic collapsing problem.

To further examine the dominance effect of $V^*$, we introduce a complementary set of simple prompts, denoted as $P_{V^*}^{\text{simple}}$. This set consists of 200 straightforward sentences where $V^*$ is the clear focal point, such as "a photo of a $V^*$", "a portrait of a $V^*$", etc. These simple prompts serve as a baseline for assessing the degree to which $V^*$ overshadows other contextual elements in the generated outputs.

Sample sentences from both prompt sets are provided in Table 3, and the full dataset, along with all prompt templates, is available in the repository at `https://github.com/tuananhbui89/Embedding-Adjustment`.

**Dataset for Evaluating Personalization Performance.** We use a subset of 9 concepts from the CustomConcept101 dataset as in the original paper (Kumari et al., 2023), each of which has 3-15 images, including 'Barn', 'Tortoise plushy', 'Teddy-Bear', 'Wooden Pot', 'Dog', 'Cat', 'Flower', 'Table', 'Chair' subjects. For the human concept, we use a subset of 10 concepts from the CelebA-HQ dataset (Liu et al., 2015), which includes 10 identities with 10-15 images per subject. Sample images from the CustomConcept101 and CelebA-HQ datasets are shown in Figure 15 and Figure 16, respectively.

To assess complex prompt handling, we compile a set of multi-concept prompts from the Custom-Concept101 dataset. Each prompt is designed to include two to three distinct elements, encouraging the model to balance multiple visual contexts. For instance:

Table 4: Sample prompts to generate personalized images. Each prompt consists of a main subject $V^*$ and a context $p$. All the data and prompts can be found in the repository.

| Set | Sample Sentences |
|---|---|
| **CelebA** | 'A photo of a $V^*$ wearing glasses and handshaking with a man'
'A photo of a $V^*$ wearing glasses and handshaking with a woman'
'A photo of a $V^*$ wearing glasses and handshaking with an old man'
'A photo of a $V^*$ wearing glasses and handshaking with a kid'
'A photo of a $V^*$ wearing glasses and holding a dog'
'A photo of a $V^*$ wearing glasses and holding a cat'
'A photo of a $V^*$ wearing glasses and holding a red book'
'A photo of a $V^*$ wearing glasses and holding a red phone'
'A photo of a $V^*$ wearing glasses and sitting on a red chair'
'A photo of a $V^*$ wearing glasses and lying on a red bed'
'A photo of a $V^*$ wearing glasses and writing in a red notebook'
'A photo of a $V^*$ wearing glasses and drinking a Coco Cola can'
'A photo of a $V^*$ wearing glasses and lifting weights'
'A photo of a $V^*$ wearing glasses and cycling'
'A photo of a $V^*$ wearing glasses and kicking a football'
'A photo of a $V^*$ wearing glasses and playing a guitar'
'A photo of a $V^*$ wearing glasses and eating a pizza' |
| **CustomConcept101** | '$V^*$ in snowy ice'
'$V^*$ in blooming sunflower field'
'$V^*$ on a boat in the sea'
'$V^*$ on top of a mountain'
'$V^*$ made of crochet'
'$V^*$ in a garden'
'a floor lamp on the side of $V^*$'
'$V^*$ and a table with chocolate cake on it'
'a puppy sitting on a $V^*$'
'a cat sitting on a $V^*$'
'a squirrel sitting on a $V^*$'
'a deer grazing near a $V^*$'
'a teddy bear on a $V^*$'
'a photo of a $V^*$ in Van Gogh style' |
| **Multi-Concept** | 'A photo of a $V^*_{man}$ wearing glasses and kissing a $V^*_{dog}$'
'A photo of a $V^*_{man}$ wearing glasses and handshaking with a $V^*_{dog}$'
'A photo of a $V^*_{man}$ wearing a hat and hugging a $V^*_{dog}$'
'A photo of a $V^*_{man}$ walking a $V^*_{dog}$ on a road with a car behind'
'A photo of a $V^*_{man}$ holding a $V^*_{dog}$ beside a car'
'A photo of a $V^*_{man}$ touching a $V^*_{dog}$ beside a park bench'
'A photo of a $V^*_{man}$ feeding a $V^*_{dog}$ with a bowl of flowers' |

- "$V^*$ tortoise plushy sitting at the beach with a view of the sea"

- "a watercolor painting of $V^*$ tortoise plushy on a mountain"

These prompts contain a primary subject $V^*$ and one or more contextual elements (context $p$), allowing us to measure the model's ability to preserve the personalized concept while maintaining accurate context alignment.

Sample prompts are provided in Table 4, and the full dataset, along with all prompt templates, is available in the repository at `https://github.com/tuananhbui89/Embedding-Adjustment`.

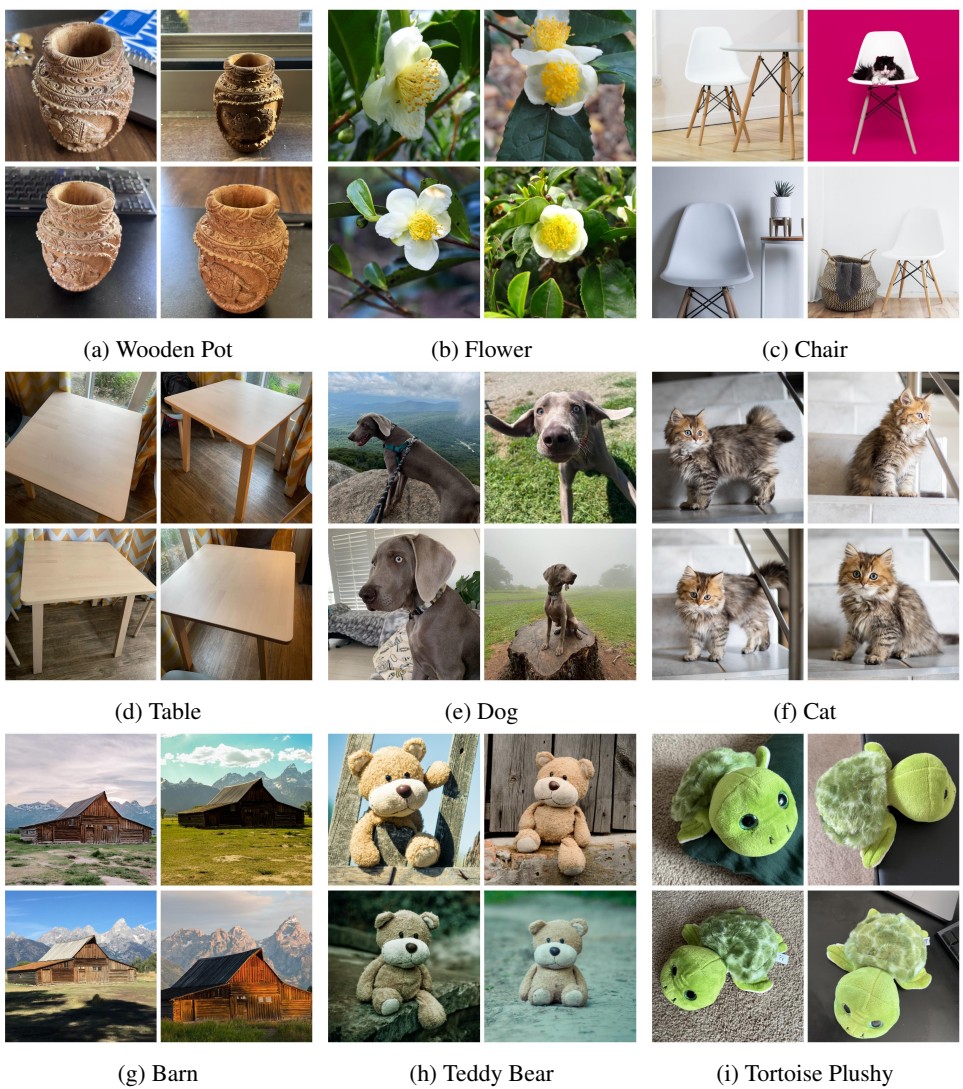

(a) Wooden Pot      (b) Flower      (c) Chair

(d) Table      (e) Dog      (f) Cat

(g) Barn      (h) Teddy Bear      (i) Tortoise Plushy

Figure 15: Sample images from the CustomConcept101 dataset.

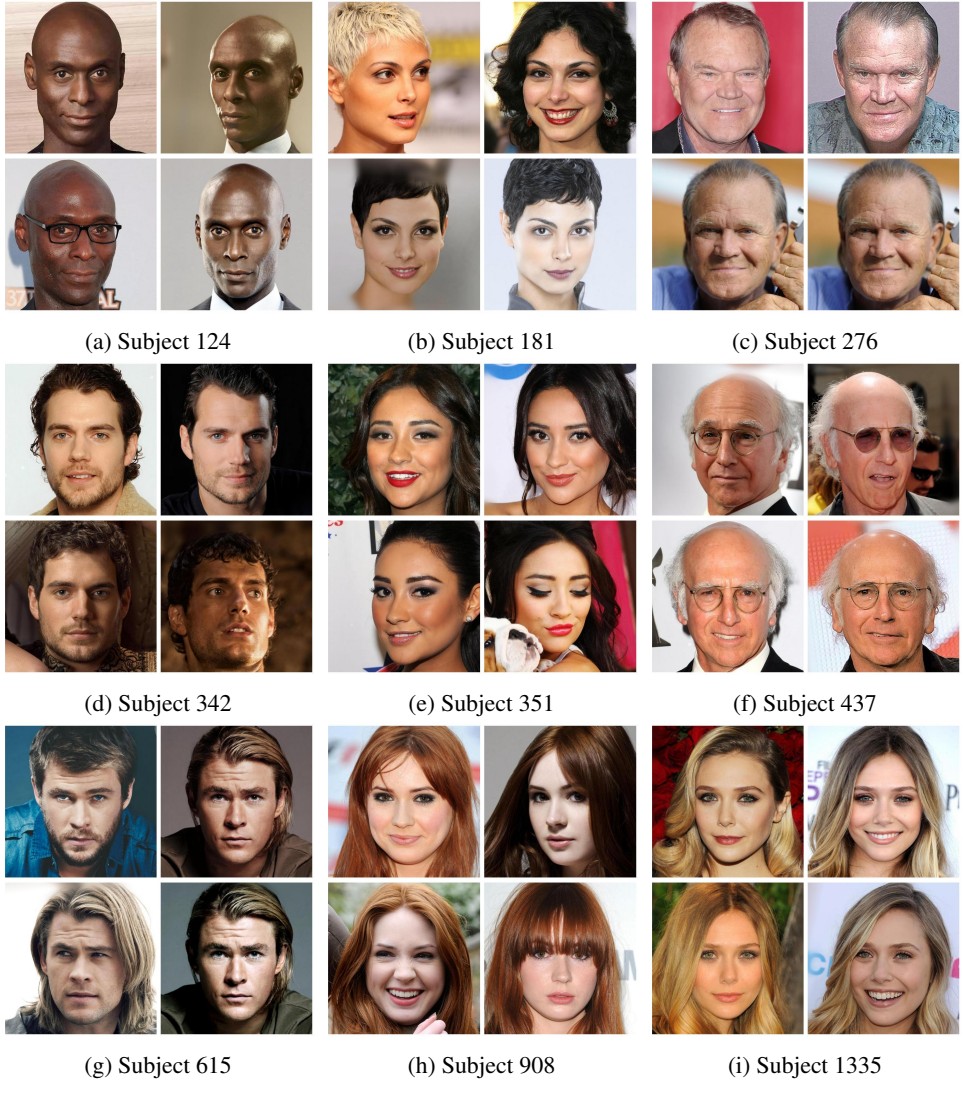

(a) Subject 124      (b) Subject 181      (c) Subject 276

(d) Subject 342      (e) Subject 351      (f) Subject 437

(g) Subject 615      (h) Subject 908      (i) Subject 1335

Figure 16: Sample images from the CelebA dataset.

Table 5: Inter-set and intra-set distances for different distance metrics

| Distance | Inter-set Distance $d(P, Q)$ | Intra-set Distance $d(P, P)$ |
|---|---|---|
| L2 | $\frac{1}{n} \sum_{i=1}^{n} \|p_i - q_i\|_2^2$ | $\frac{1}{n^2} \sum_{i=1}^{n} \sum_{j=1}^{n} \|p_i - p_j\|_2^2$ |
| Hausdorff | $\max(\max_{p_i \in P} \min_{q_j \in Q} d_{L2}(p_i, q_j),$ $\max_{q_i \in Q} \min_{p_j \in P} d_{L2}(q_i, p_j))$ | $\max_{p_i \in P} \min_{p_j \in P, p_j \neq p_i} d_{L2}(p_i, p_j)$ |
| Mahalanobis | $\frac{1}{n} \sum_{i=1}^{n} \sqrt{(p_i - \mu_P)^T \Sigma_P^{-1} (p_i - \mu_P)}$ | $\frac{1}{n^2} \sum_{i=1}^{n} \sum_{j=1}^{n} \sqrt{(p_i - p_j)^T \Sigma_P^{-1} (p_i - p_j)}$ |
| KL | $D_{KL}(P \,\|\, Q)$ | $D_{KL}(P \,\|\, P)$ |

## F.2 EVALUATION METRICS

**Personalization Metrics.** We use the CLIP-T **text-image** alignment score (Radford et al., 2021) to evaluate the alignment between the generated images $\hat{x}$ and the prompts. To have a better understanding of which part of the prompt contributes to construct the generated images, we break down each prompt into multiple segments/concepts and calculate the alignment score for each segment/concept, i.e., 'CLIP$_T^f$'/'CLIP$_T^c$'/'CLIP$_T^p$' denotes the alignment score of the full prompt, the first segment—the personal concept, and the second segment—the context, respectively. We use the CLIP-I **image-image** alignment score (Radford et al., 2021) and DINO **image-image** alignment score (Caron et al., 2021) to evaluate the alignment between the generated images and the reference images.

**Semantic Shifting Metrics.** Given the two sets $P$ and $Q$ (i.e., $P = P_{V^*} = \{\tau(\lfloor a_i, V^* \rfloor)\}$ and $Q = P_c = \{\tau(\lfloor a_i, c \rfloor)\}$ which are the embeddings of the prompts in the prompt sets $P_{V^*}$ and $P_c$ respectively), we propose to use the following metrics to measure the difference between the two sets.

**L2 distance** measures the Euclidean distance between two points in a vector space.

$$d_{L2}(p_i, q_i) = \|p_i - q_i\|_2 \tag{8}$$

**Hausdorff distance** measures the maximum distance from any point in $P$ to the nearest point in $Q$. In other words, it measures the greatest distance from a point in one set to the nearest point in another set.

$$d_H(P, Q) = \max(\max_{p_i \in P} \min_{q_j \in Q} d_{L2}(p_i, q_j), \max_{q_i \in Q} \min_{p_j \in P} d_{L2}(q_i, p_j)) \tag{9}$$

**Mahalanobis distance** measures how far the point $p_i$ is from the center of the set $P$, taking into account the correlation between the dimensions of the set. Unlike the L2 distance which treats all dimensions equally, the Mahalanobis distance adapts to the shape and spread of the set $P$.

$$d_M(p_i, P) = \sqrt{(p_i - \mu_P)^T \Sigma_P^{-1} (p_i - \mu_P)} \tag{10}$$

where $\mu_P$ is the mean of the set $P$ and $\Sigma_P$ is the covariance matrix of the set $P$.

**KL divergence** . We propose to measure the relative relationship between each data point to the entire set by using the Normalized Temperature-scaled Softmax function

$$p(p_i \mid p_j, P) = \frac{\exp(\text{sim}(p_i, p_j)/T)}{\sum_{p_k \in P} \exp(\text{sim}(p_i, p_k)/T)} \tag{11}$$

where $T$ is the temperature parameter. $p(p_i \mid p_j, P)$ measures the the relative relationship between $p_j$ and the anchor $p_i$ in comparison to the entire set $P$. From that, we can have $p(p_i \mid P) = \{p(p_i \mid p_j, P) \; \forall p_j \in P\}$ to represent the relative relationship between $p_i$ and the entire set $P$. Similarly,

$p(q_i \mid Q) = \{p(q_i \mid q_j, Q) \; \forall q_j \in Q\}$ to represent the relative relationship between $q_i$ and the entire set $Q$.

The KL divergence between $P$ and $Q$ is then defined as:

$$D_{KL}(P \parallel Q) = \sum_{i=1}^{n} p(p_i \mid P) \log \frac{p(p_i \mid P)}{p(q_i \mid Q)} \tag{12}$$

which measures the difference between the two distributions $P$ and $Q$. The higher the KL divergence, the more different the two distributions are, the more semantic shifting the learned embedding $v^*$ has.

**Alignment Metrics.** The primary objective of generative personalization is to produce visually compelling images that accurately capture the unique characteristics of a personalized concept from a reference set, while maintaining semantic alignment with the input textual prompt.

To evaluate this, we use two key metrics based on the CLIP model (Radford et al., 2021):

**Visual Fidelity:** We measure the alignment between the generated image and a ground-truth image using the CLIP image-image alignment score. A higher score indicates a closer match to the reference, reflecting better preservation of the personalized visual features.

**Prompt Consistency:** We assess the alignment between the generated image and the input textual prompt using the CLIP text-image alignment score. A higher score indicates that the generated image more accurately reflects the intended context and details of the input text.

However, complex prompts often contain multiple concepts, making a single text-image alignment score insufficient to capture the nuanced relationship between the personalized concept $V^*$ and its broader context $p$. To address this, we separately compute alignment scores for:

**Main Concept Alignment ($V^*$):** Measuring the fidelity of the personalized concept itself.

**Context Alignment ($p$):** Evaluating how well the broader contextual elements are represented.

This multi-level evaluation provides a more comprehensive understanding of how well the generated images capture both the personalized visual identity and the intended scene context.

For all evaluations, we use the implementation provided in the TorchMetrics library, available at `https://lightning.ai/docs/torchmetrics/stable/multimodal/clip_score.html`.

---

**System Prompt for VLM-based Evaluation Metrics**

**Task Definition**
You are an expert visual evaluator for subject-driven image generation. You will receive:

1. **Reference image A** — the target subject (object or person)

2. **Prompt P** — text describing the desired scene/context

3. **Generated image O** — the image to evaluate

Produce two independent integer scores (0–4):

1. **Prompt Adherence** — Does O match the scene, spatial relations, actions, attributes, and style in P?

2. **Subject Identity** — Does the subject in O visually match the reference subject in A?

**Key Principle:** These scores are independent. Perfect subject in wrong scene = high Subject, low Prompt. Wrong subject in correct scene = high Prompt, low Subject.

**Prompt Adherence Score (Scene/Context)**
**Evaluate:** Scene/setting, spatial relationships, actions, attributes, style specified in P.
**Ignore:** Whether the specific subject from A is present (only check if *something* fills the subject role correctly).
**0** - No alignment with prompt
**1** - Minimal alignment; major elements missing/wrong
**2** - Some core elements correct; important parts missing/contradicted
**3** - Most elements correct; minor omissions in secondary attributes
**4** - All major elements present and correct; full prompt satisfaction

**Subject Identity Score (Visual Matching)**
**Evaluate:** Does the subject in O look like the reference in A? Check shape, distinctive features, colors, recognizable characteristics.
**Ignore:** Whether subject is in correct scene/position (that's Prompt Adherence).
**0** - Subject absent or completely unrecognizable
**1** - Weak resemblance; most identifying features differ
**2** - Ambiguous identity; significant changes to important features
**3** - Clearly recognizable; some features altered but identity preserved
**4** - Unambiguous match; all distinctive features preserved

**Output Format**
Output ONLY these two lines with no other text:
```
PromptScore:   <integer 0-4>
SubjectScore:   <integer 0-4>
```
**Examples:**

- Same subject, wrong scene → PromptScore: 1, SubjectScore: 4

- Wrong subject, correct scene → PromptScore: 3, SubjectScore: 0

- Both perfect → PromptScore: 4, SubjectScore: 4

Now evaluate the following inputs:

---

**VLM-based Evaluation Metrics.** We use the VLM-P and VLM-I metrics to evaluate the alignment between the generated images and the reference images and the prompts, respectively. Both metrics output a score between 0 and 4, where 0 means there are no correspondence between the generated image and the reference image (VLM-I) or input prompt (VLM-P), while 4 means the perfectly matches. The final score for each metric is obtained by averaging all inference prompts and samples (16 prompts times 50 random samples per concept/setting) and normalizing to the range [0, 100%]. We include the full system prompts and evaluation scripts in the Github repository.

## F.3 COMPUTATIONAL SETTINGS

All experiments are conducted on a single NVIDIA RTX 4090 GPU with 24GB of memory, using the Stable Diffusion v1.5 model as the base model. To prevent memory overflow, we fine-tune the model with Textual Inversion (TI), DreamBooth (DB), and Custom Diffusion (CD) using a batch size of 1 across all methods.

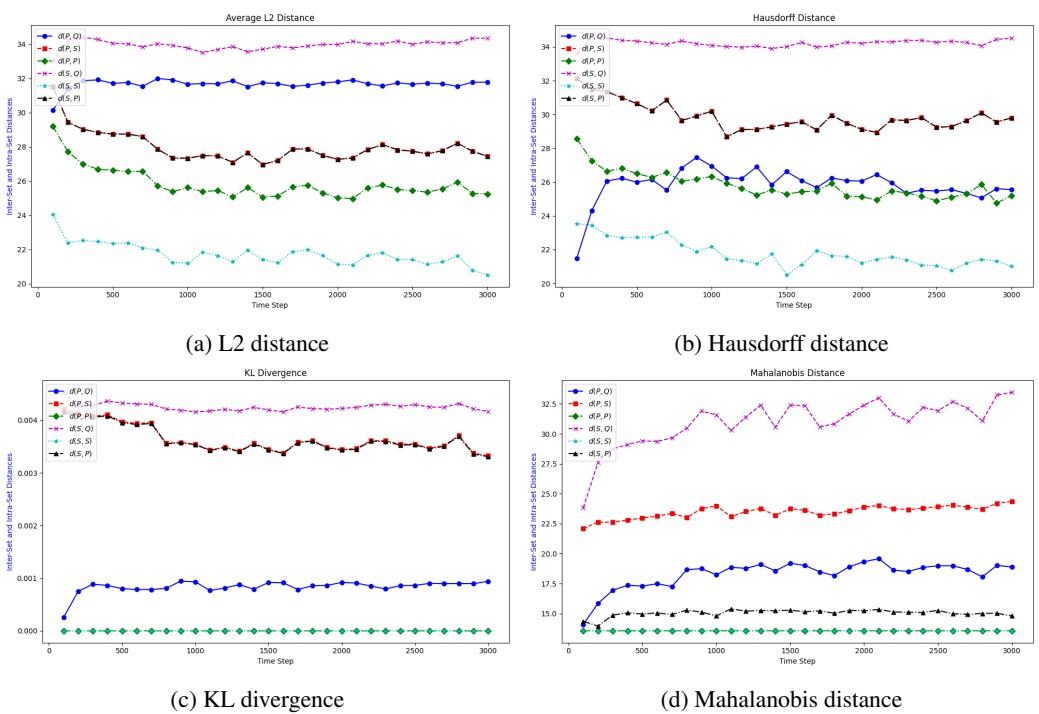

(a) L2 distance

(b) Hausdorff distance

(c) KL divergence

(d) Mahalanobis distance

Figure 17: Different distance metrics between the sets $P = \{\tau(\lfloor a_i, V^* \rfloor)\}$ and $Q = \{\tau(\lfloor a_i, c \rfloor)\}$ in Textual Inversion (TI), showing the semantic shifting of the learned embedding over training iterations.

Table 6: Results on CustomConcept101 dataset, tort* means tortoise plushy, teddy* means plushy teddy bear, wpot* means wooden pot. The first/second metric is the $\text{CLIP}_T^f$/CLIP-I score. The blue number indicates the proposed method outperforms its baseline counterpart, while the red number indicates the opposite. The GAP is the average improvement over all concepts. Qualitative results are shown in Fig. 24.

| Method | dog | cat | tort* | teddy* | chair | table | flower | wpot* | barn | GAP |
|---|---|---|---|---|---|---|---|---|---|---|
| TI | 26.05/ 60.18 | 26.64/ 79.65 | 30.49/ 73.09 | 27.29/ 77.05 | 27.19/ 77.97 | 27.06/ 64.36 | 25.26/ 68.63 | 28.0/ 67.02 | 27.45/ 80.22 | 0.00/ 0.00 |
| TI+TEA | 26.14/ 58.97 | 27.94/ 77.47 | 30.71/ 72.74 | 27.68/ 76.02 | 28.13/ 73.27 | 27.78/ 61.33 | 26.28/ 63.77 | 28.34/ 66.44 | 28.2/ 77.06 | 0.64/ -2.34 |
| DB | 20.28/ 59.93 | 20.29/ 74.82 | 24.55/ 91.14 | 18.89/ 85.52 | 21.87/ 87.5 | 20.77/ 73.83 | 20.07/ 82.98 | 20.43/ 66.69 | 23.58/ 80.98 | 0.00/ 0.00 |
| DB+TEA | 21.71/ 66.25 | 20.1/ 92.43 | 25.08/ 91.1 | 19.65/ 86.98 | 23.61/ 86.6 | 22.97/ 84.87 | 23.12/ 76.05 | 24.99/ 82.46 | 26.31/ 77.77 | 1.87/ 4.57 |
| CD | 27.33/ 56.09 | 29.37/ 78.34 | 31.33/ 78.42 | 28.47/ 77.67 | 27.16/ 69.88 | 27.88/ 62.84 | 26.7/ 62.66 | 30.0/ 69.18 | 26.34/ 70.95 | 0.00/ 0.00 |
| CD+TEA | 27.45/ 56.18 | 29.25/ 77.59 | 31.06/ 79.61 | 28.04/ 78.64 | 28.16/ 71.81 | 28.34/ 63.08 | 27.37/ 62.63 | 30.15/ 68.0 | 27.82/ 71.78 | 0.34/ 0.37 |
| CL | 27.28/ 55.06 | 29.19/ 77.42 | 31.11/ 78.32 | 28.26/ 76.4 | 28.49/ 74.52 | 27.68/ 60.34 | 26.76/ 63.98 | 29.69/ 61.09 | 25.72/ 69.87 | 0.00/ 0.00 |
| CL+TEA | 27.72/ 56.44 | 29.12/ 77.26 | 30.89/ 78.87 | 28.07/ 77.46 | 29.01/ 75.56 | 28.11/ 60.54 | 27.21/ 64.87 | 30.2/ 60.72 | 27.65/ 71.27 | 0.42/ 0.67 |

For the DreamBooth LoRA method, we follow the recommended settings from the Diffusers' example page, using a learning rate of 1e-4, rank 4, and enabling text encoder training for improved performance. Textual Inversion is fine-tuned with a learning rate of 5e-4, while Custom Diffusion uses a more conservative learning rate of 5e-6.

All code implementations are adapted from the Hugging Face Diffusers library.

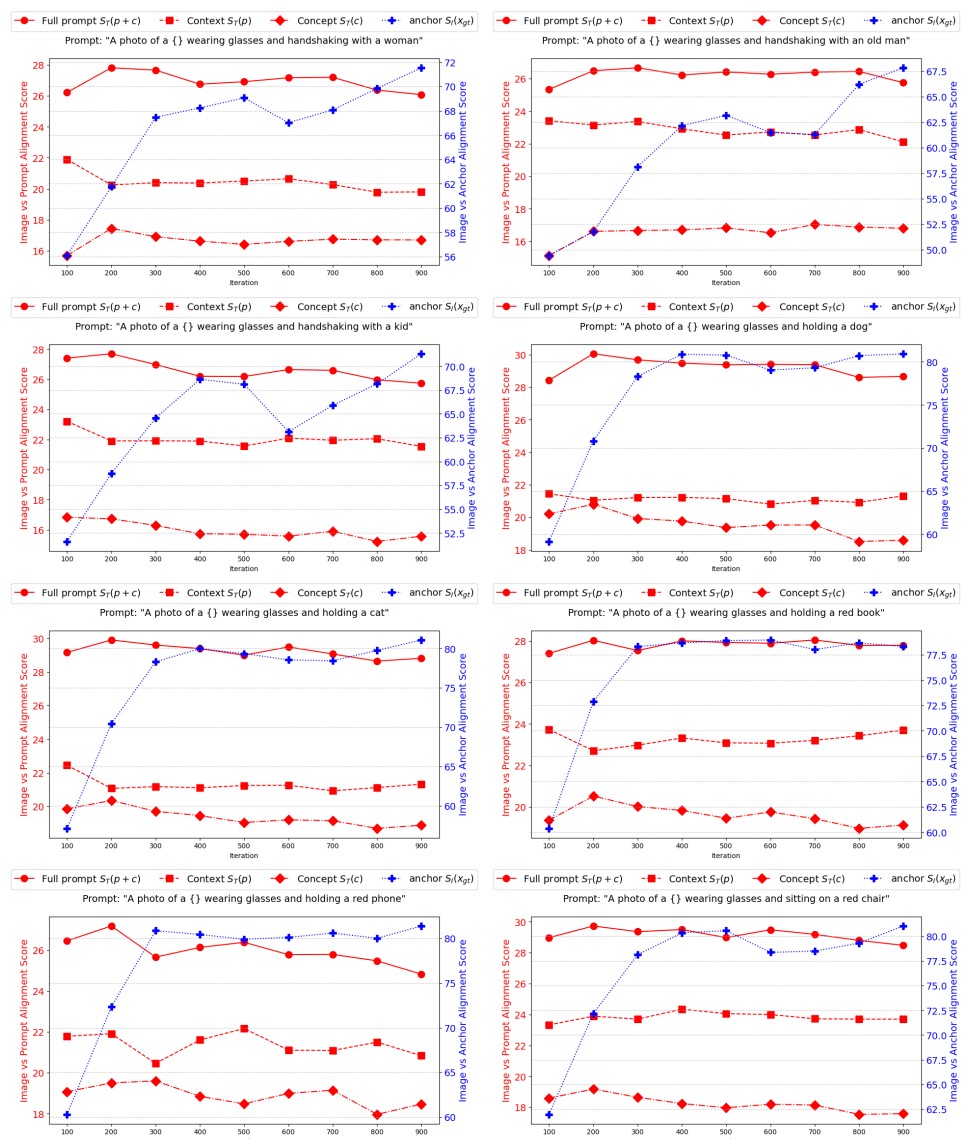

Figure 18: Alignment scores showing the SCP on Textual Inversion with different prompts.

Table 7: Results on CelebA dataset. The first/second metric is the CLIP$_T^f$/CLIP-I score. The blue number indicates the proposed method outperforms its baseline counterpart, while the red number indicates the opposite. The GAP is the average improvement over all concepts. Qualitative results are shown in Fig. 23.

| Method | 124 | 181 | 276 | 342 | 351 | 437 | 615 | 908 | 1335 | 1429 | GAP |
|---|---|---|---|---|---|---|---|---|---|---|---|
| TI | 23.7/ 68.93 | 26.29/ 61.68 | 21.32/ 65.33 | 26.06/ 75.64 | 25.25/ 69.63 | 23.46/ 71.19 | 26.54/ 74.37 | 24.61/ 64.5 | 25.4/ 67.55 | 24.97/ 60.88 | 0.00/ 0.00 |
| TI+TEA | 24.23/ 68.48 | 26.49/ 59.6 | 20.97/ 65.2 | 27.28/ 71.01 | 26.11/ 65.73 | 23.93/ 66.25 | 26.89/ 70.56 | 25.98/ 62.97 | 25.69/ 66.72 | 25.68/ 59.01 | 0.57/ -2.41 |
| DB | 25.37/ 54.78 | 25.91/ 61.75 | 22.11/ 58.7 | 24.37/ 70.41 | 25.0/ 62.95 | 20.65/ 75.15 | 25.06/ 66.56 | 24.93/ 63.46 | 23.66/ 59.2 | 24.65/ 60.42 | 0.00/ 0.00 |
| DB+TEA | 26.26/ 50.67 | 26.59/ 60.23 | 24.3/ 47.66 | 25.58/ 65.83 | 25.43/ 61.34 | 22.56/ 68.79 | 26.03/ 60.88 | 25.92/ 60.34 | 24.79/ 53.66 | 25.37/ 58.46 | 1.11/ -4.55 |
| CS | 26.92/ 43.56 | 26.87/ 52.48 | 26.91/ 36.22 | 27.14/ 52.64 | 26.94/ 53.39 | 26.97/ 55.45 | 26.99/ 53.96 | 26.93/ 53.32 | 27.05/ 44.95 | 26.88/ 50.33 | 0.00/ 0.00 |
| CS+TEA | 27.09/ 44.88 | 26.83/ 55.03 | 27.18/ 37.1 | 27.37/ 53.63 | 26.99/ 55.56 | 27.11/ 56.41 | 27.16/ 55.01 | 26.93/ 55.33 | 26.94/ 46.68 | 26.98/ 52.27 | 0.09/ 1.56 |
| CL | 26.44/ 57.5 | 29.44/ 59.93 | 24.73/ 55.39 | 27.11/ 56.91 | 29.25/ 59.16 | 24.68/ 61.77 | 26.77/ 58.71 | 28.76/ 59.07 | 28.45/ 49.39 | 29.33/ 56.45 | 0.00/0.00 |
| CL+TEA | 26.87/ 54.69 | 29.75/ 60.04 | 25.54/ 52.67 | 27.5/ 56.92 | 29.51/ 59.61 | 25.37/ 59.28 | 27.13/ 59.12 | 29.1/ 59.45 | 28.75/ 49.42 | 29.34/ 56.2 | 0.39/ -0.69 |

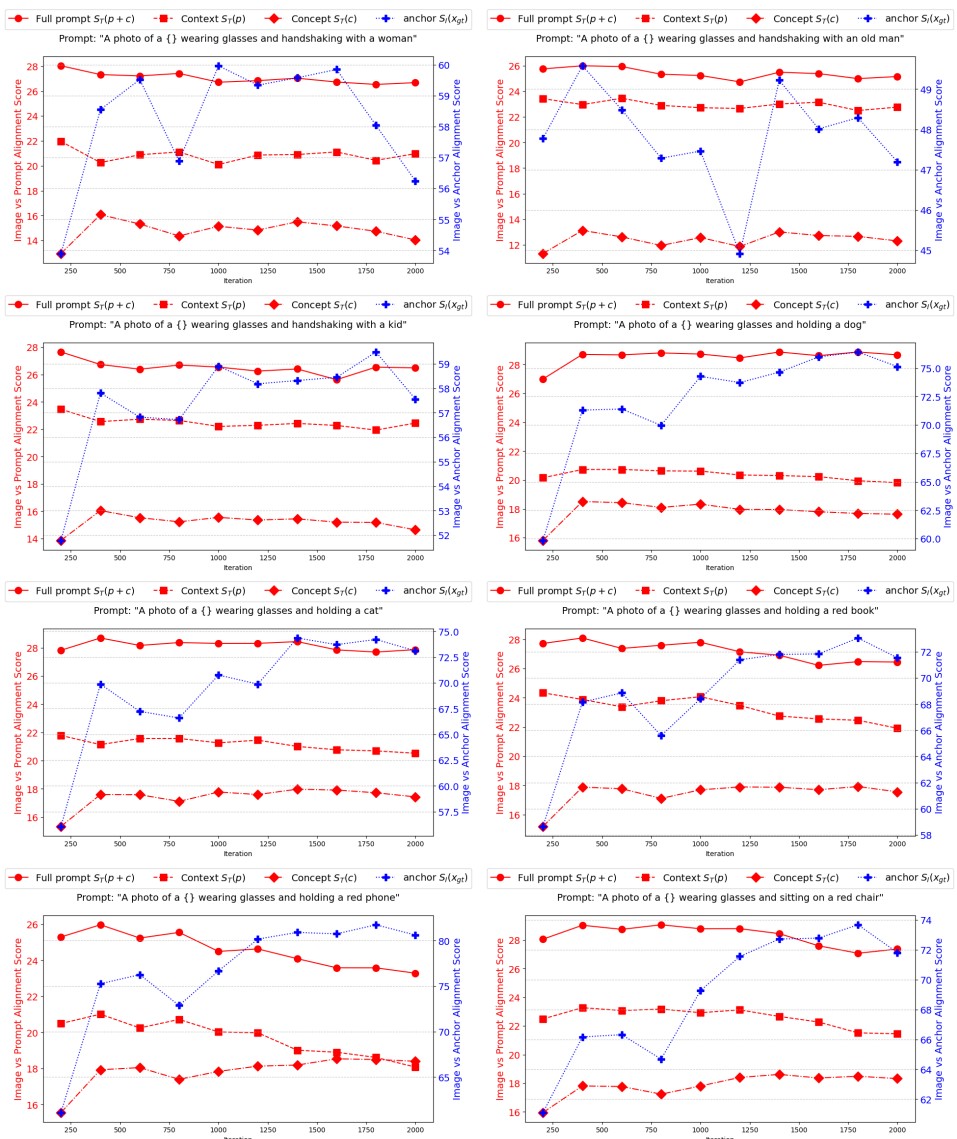

Figure 19: Alignment scores showing the SCP on DreamBooth with different prompts.

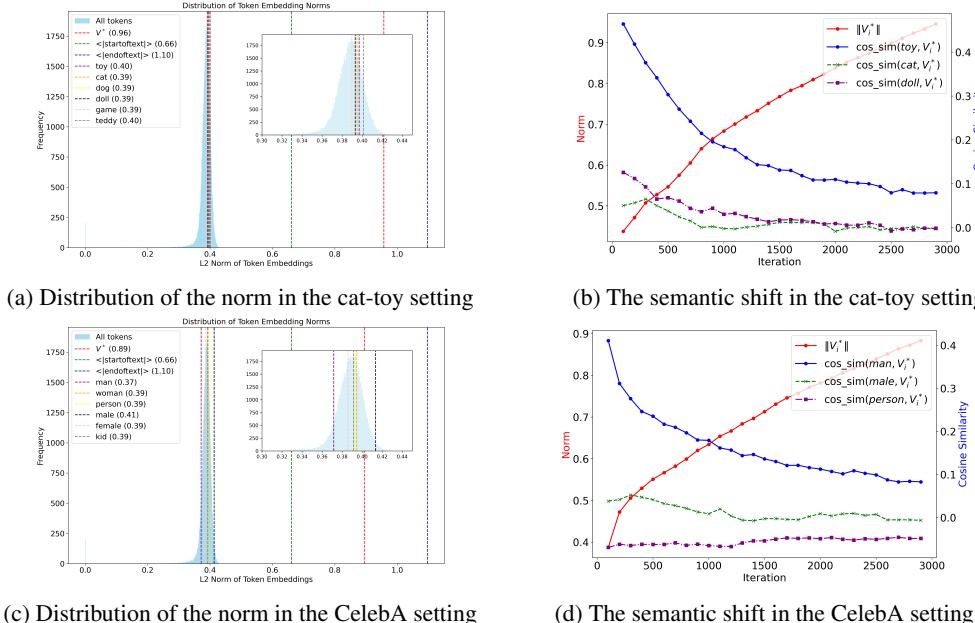

(a) Distribution of the norm in the cat-toy setting

(b) The semantic shift in the cat-toy setting

(c) Distribution of the norm in the CelebA setting

(d) The semantic shift in the CelebA setting

Figure 20: Left: The distribution of the norm of the token embedding $M$ including special token $V^*$, Right: The semantic drift of $V^*$ in term of magnitude and direction over time. The same phenomenon is observed in DreamBooth as shown in Figure 21.

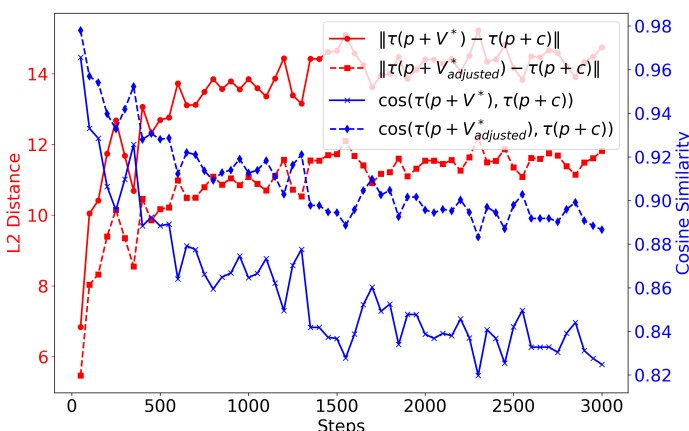

Figure 21: The semantic drift of the embedding of entire prompt $\lfloor p, V^* \rfloor$ in term of magnitude and direction over time with DreamBooth. The adjusted embedding $\lfloor p, V^*_{\text{adjusted}} \rfloor$ is obtained by using the TEA framework with $\alpha = 0.2$ and $\beta = 1.5$.

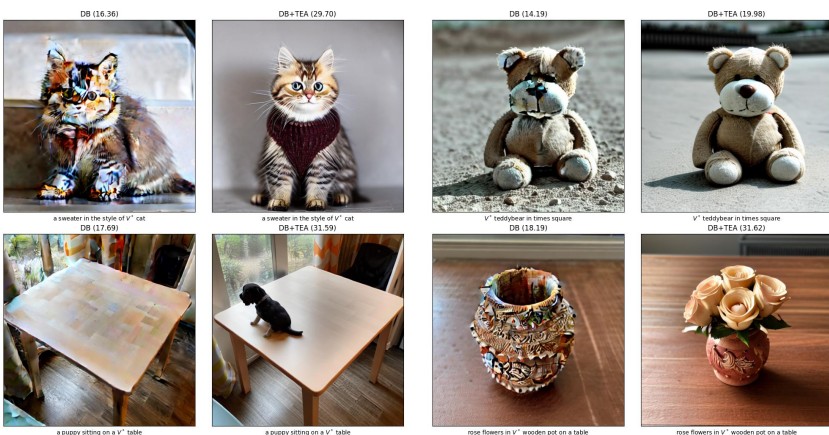

Figure 22: Some cool effects of our method (DB+TEA) over the baseline counterpart DB. The SCP in DB lead to the distorted generated images. Our method successfully corrects the semantic embedding and generates more coherent and realistic images.

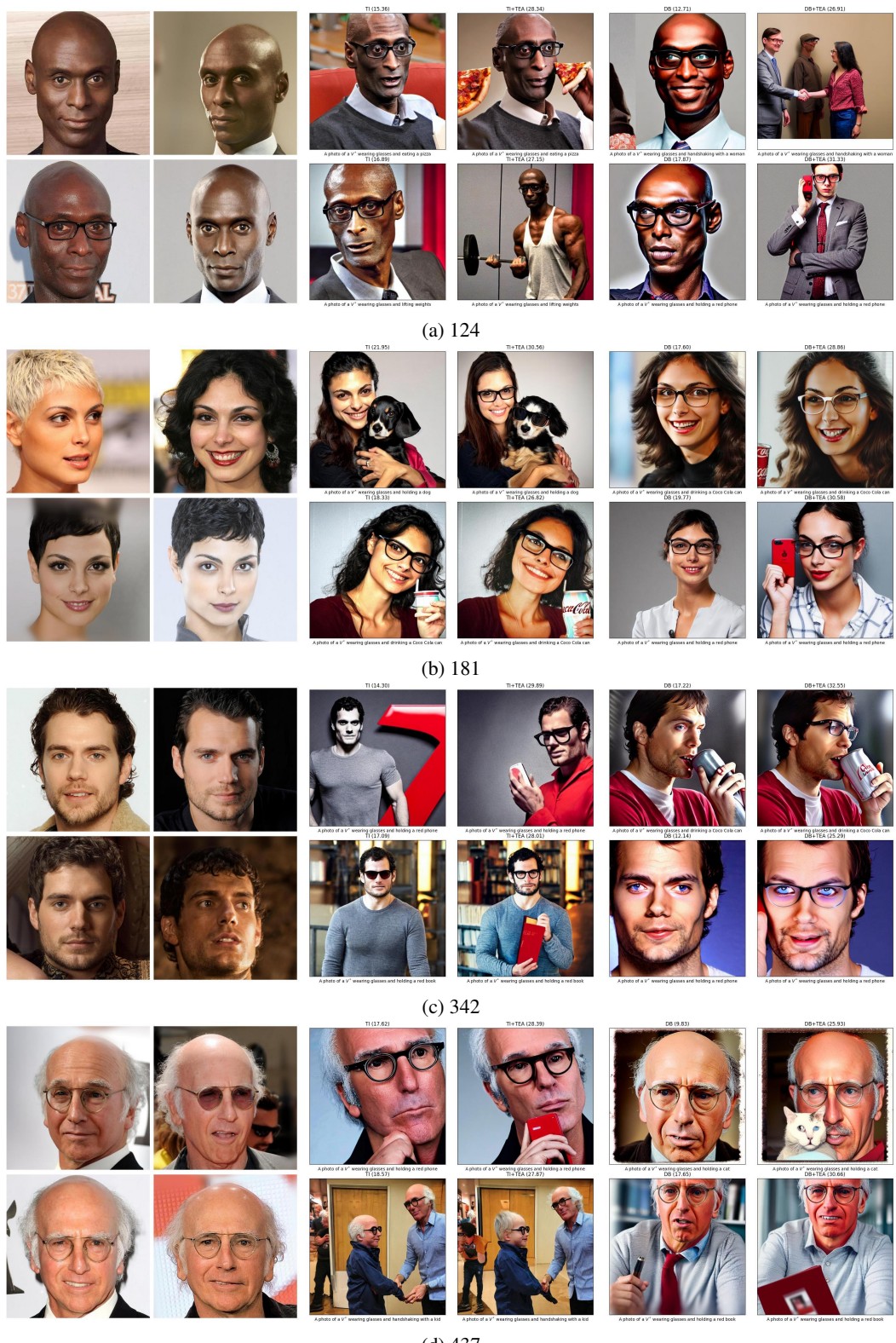

Figure 23: Qualitative comparison between baseline methods and their TEA variants on the CelebA dataset. Column 1-2: Reference images. Column 3: TI, Column 4: TI+TEA, Column 5: DB, Column 6: DB+TEA. Input prompts are shown below each image while alignment scores are shown on the top. More results can be found in the repository.

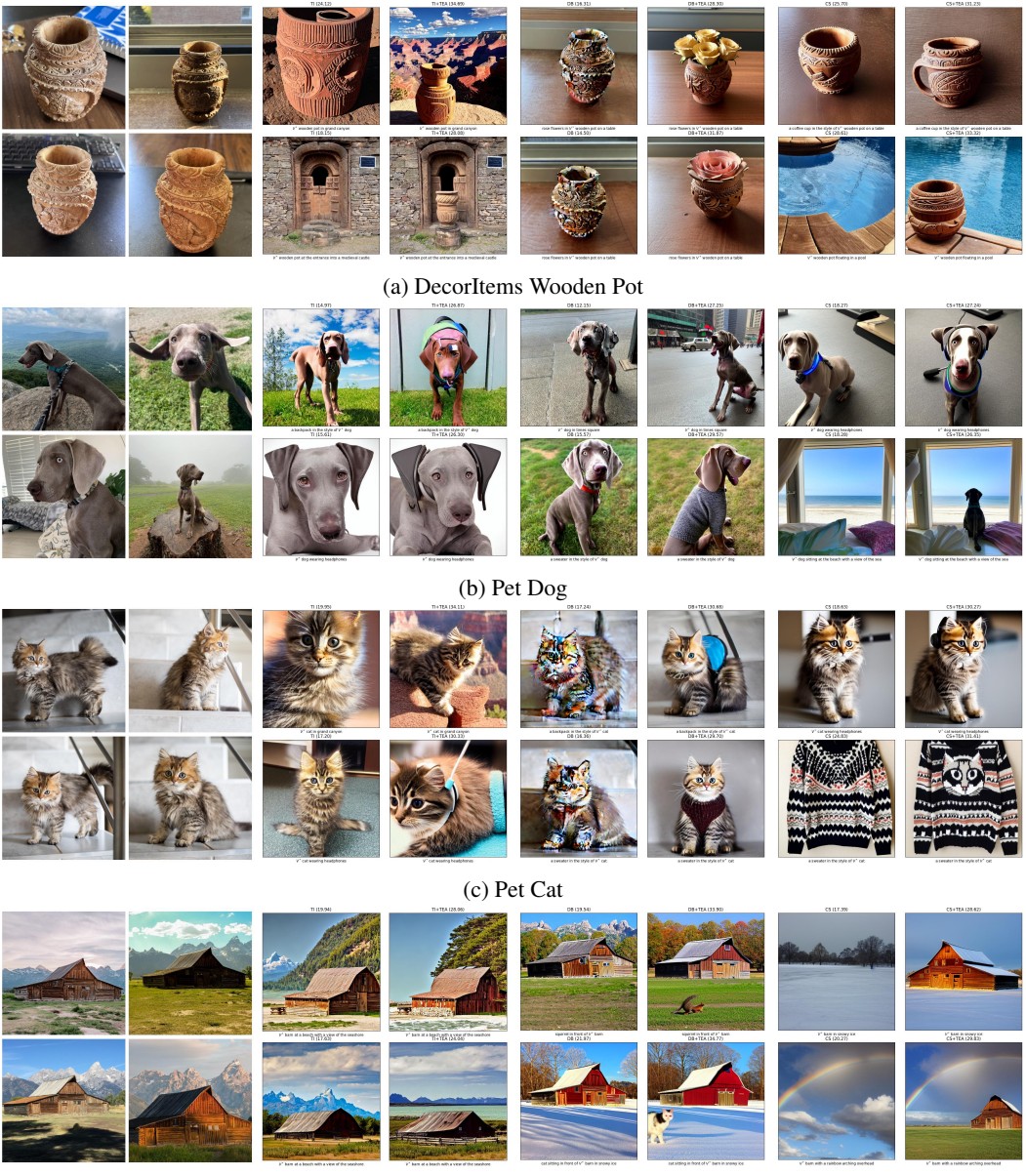

(a) DecorItems Wooden Pot

(b) Pet Dog

(c) Pet Cat

(d) Scene Barn

Figure 24: Qualitative comparisons between baseline methods and their TEA variants on the Custom-Concept101 dataset. Column 1-2: Reference images. Column 3: TI, Column 4: TI+TEA, Column 5: DB, Column 6: DB+TEA. Input prompts are shown below each image while alignment scores are shown on the top. More results can be found in the repository.

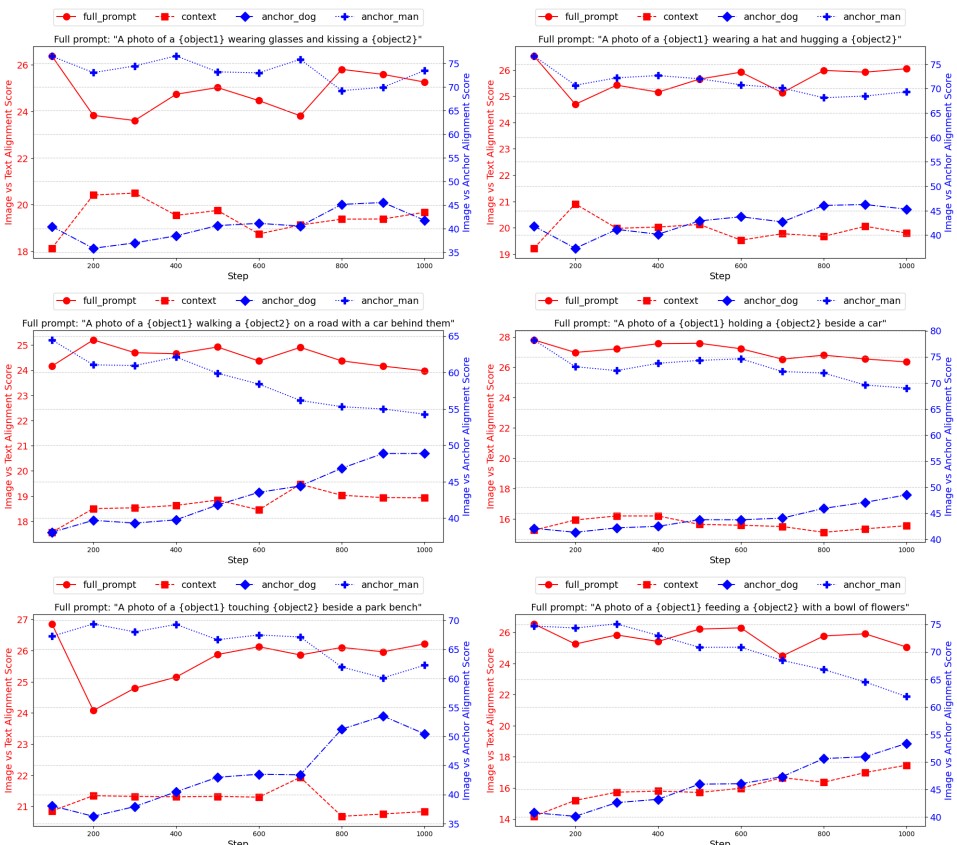

Figure 25: Alignment scores showing the SCP in multiple-concept personalization settings. The embedding of $V^*_{man}$ held fixed while the embedding of $V^*dog$ is varied over fine-tuning step. See Figures 26 and 27 for the example images.

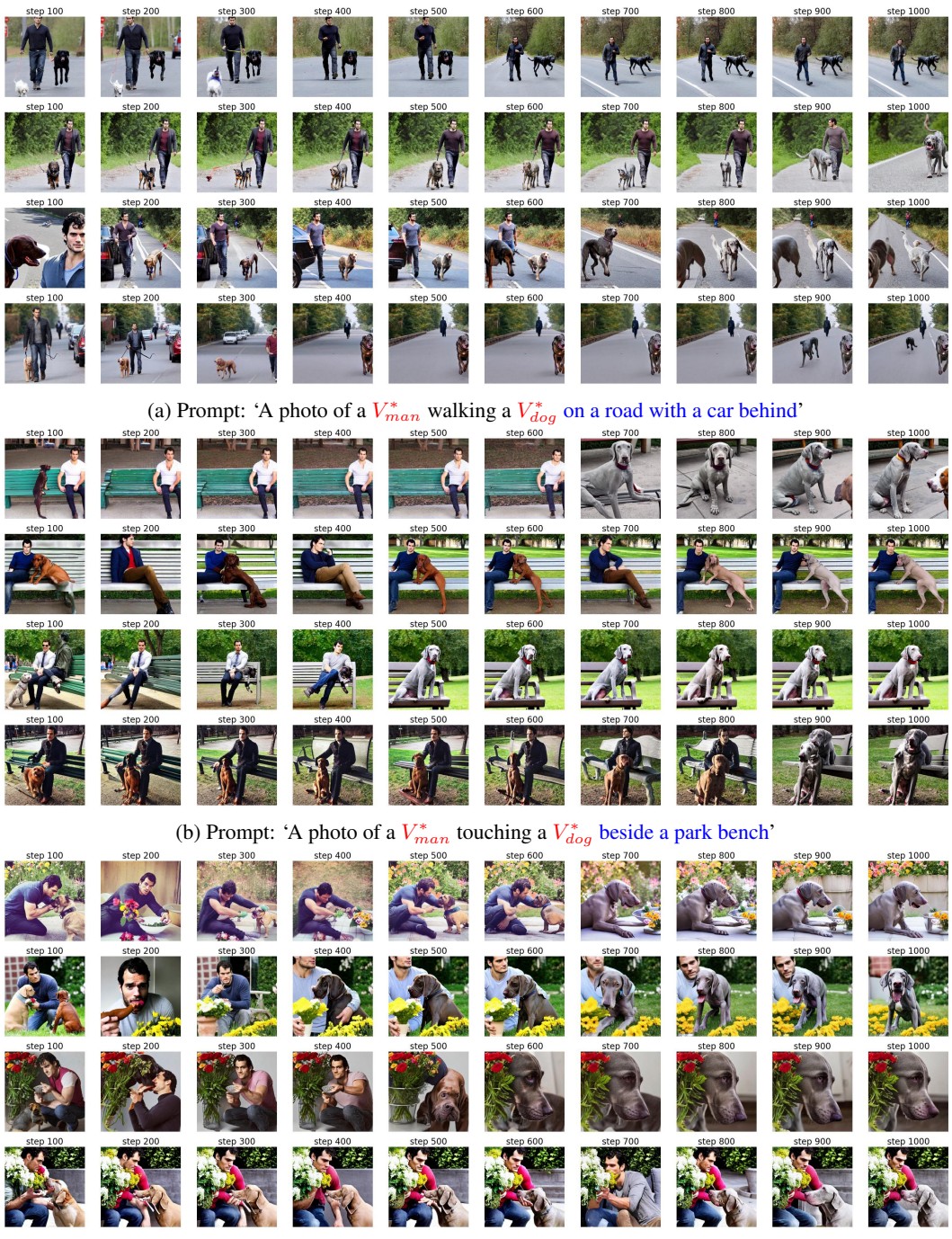

(a) Prompt: 'A photo of a $V^*_{man}$ walking a $V^*_{dog}$ on a road with a car behind'

(b) Prompt: 'A photo of a $V^*_{man}$ touching a $V^*_{dog}$ beside a park bench'

(c) Prompt: 'A photo of a $V^*_{man}$ feeding a $V^*_{dog}$ with a bowl of flowers'

Figure 26: Illustration of the semantic collapsing problem on multi-concepts personalization. The embedding of $V^*_{man}$ **is fixed** while the embedding of $V^*_{dog}$ **is varied** across different training steps. See Figure 9 for the detailed alignment scores.

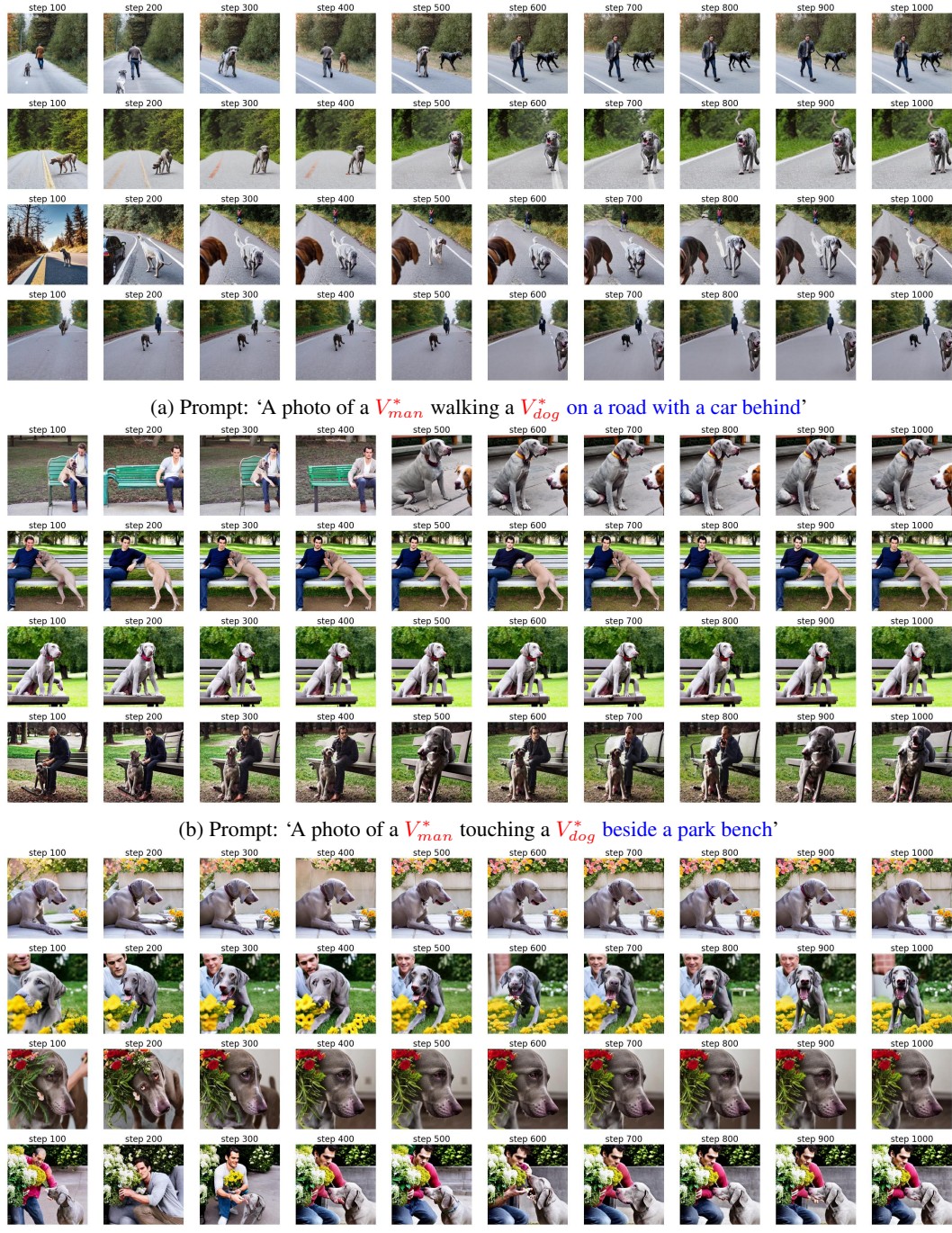

(a) Prompt: 'A photo of a $V^*_{man}$ walking a $V^*_{dog}$ on a road with a car behind'

(b) Prompt: 'A photo of a $V^*_{man}$ touching a $V^*_{dog}$ beside a park bench'

(c) Prompt: 'A photo of a $V^*_{man}$ feeding a $V^*_{dog}$ with a bowl of flowers'

Figure 27: Illustration of the semantic collapsing problem on multi-concepts personalization. The embedding of $V^*_{man}$ **is varied** across different training steps while the embedding of $V^*_{dog}$ **is fixed**. See Figure 9 for the detailed alignment scores.

SKS dog in a construction outfit

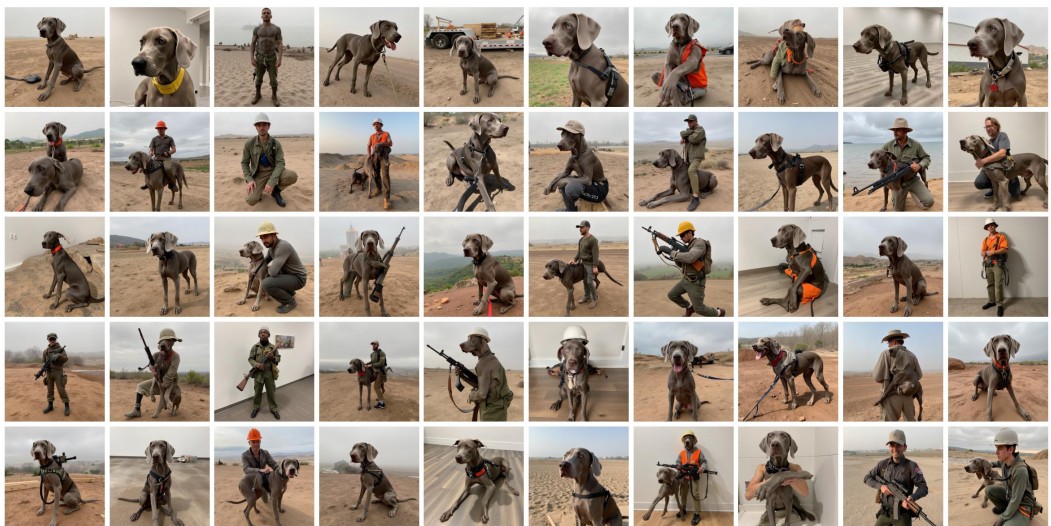

(a) Output from EasyControl

SKS dog in a construction outfit

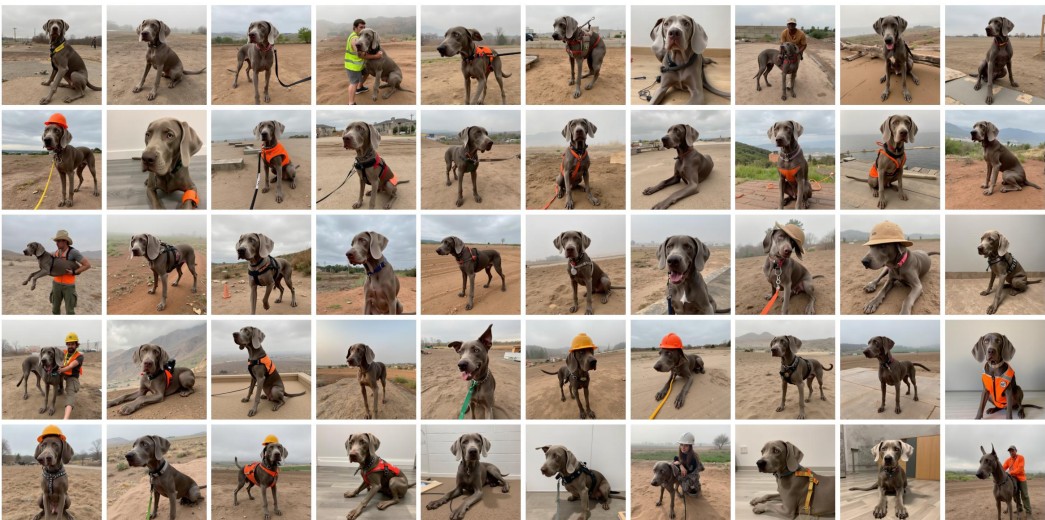

(b) Output from EasyControl with TEA

Figure 28: Comparison of the output from EasyControl pipeline with and without TEA with the same prompt: '$V_{man}^*$ dog in a construction outfit' and same random seed. EasyControl with TEA significantly improves the prompt fidelity, mitigating the failure cases of the original EasyControl pipeline, such as the dog stands beside a person or holding a gun. More results showing the same improvement of EasyControl with TEA can be found in the repository at `https://github.com/tuananhbui89/Embedding-Adjustment`.

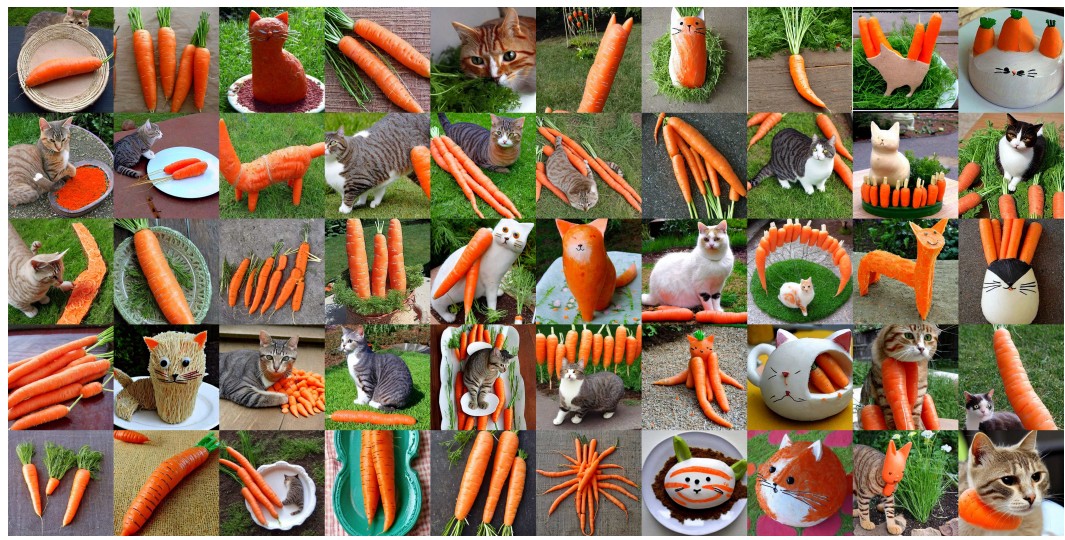

(a) Output from ReVersion

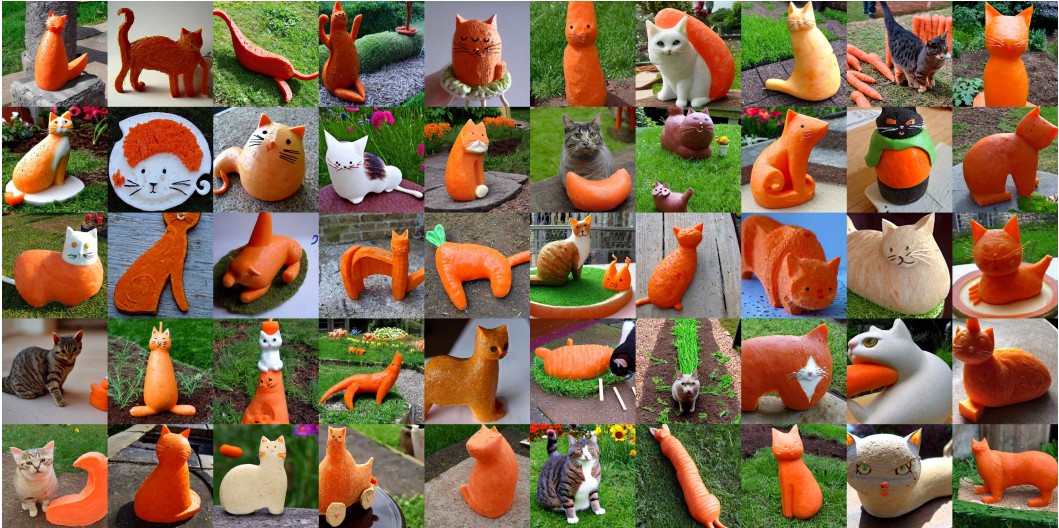

(b) Output from ReVersion with TEA

Figure 29: Comparison of the output from ReVersion pipeline with and without TEA with the same prompt: 'cat <R> carrot in the garden' and same random seed. ReVersion with TEA significantly improves the prompt fidelity, mitigating the failure cases of the original ReVersion pipeline, such as the cat is not carved by the carrot or only shows the carrot. More results showing the same improvement of ReVersion with TEA can be found in the repository at `https://github.com/tuananhbui89/Embedding-Adjustment`.

SKS barn surrounded by blooming sunflower field

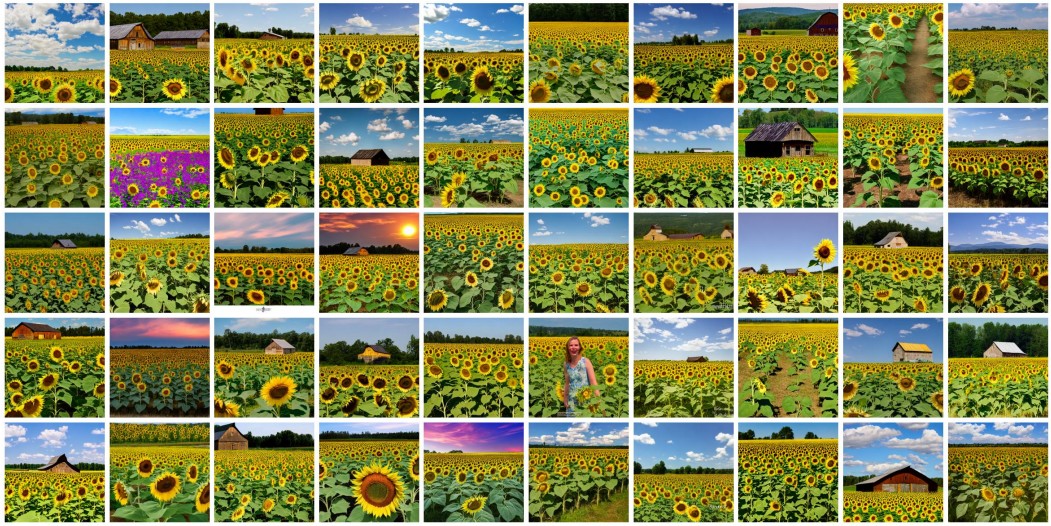

(a) Output from ClassDiffusion

SKS barn surrounded by blooming sunflower field

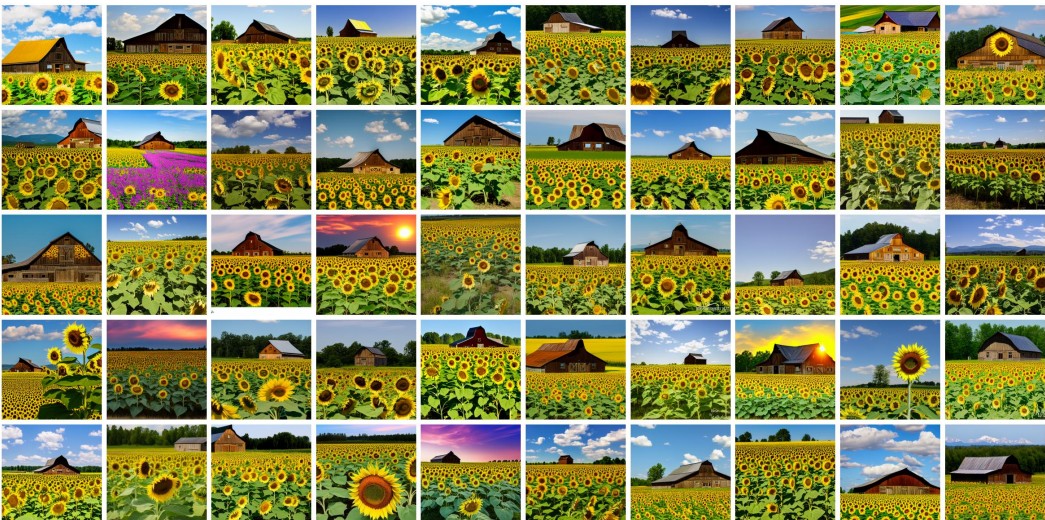

(b) Output from ClassDiffusion with TEA

Figure 30: Comparison of the output from ClassDiffusion pipeline with and without TEA with the same prompt: 'barn' and same random seed. ClassDiffusion with TEA significantly improves the prompt fidelity, mitigating the failure cases of the original ClassDiffusion pipeline, such as the image does not contain a barn but only a sunflower field. More results showing the same improvement of ClassDiffusion with TEA can be found in the repository at `https://github.com/tuananhbui89/Embedding-Adjustment`.

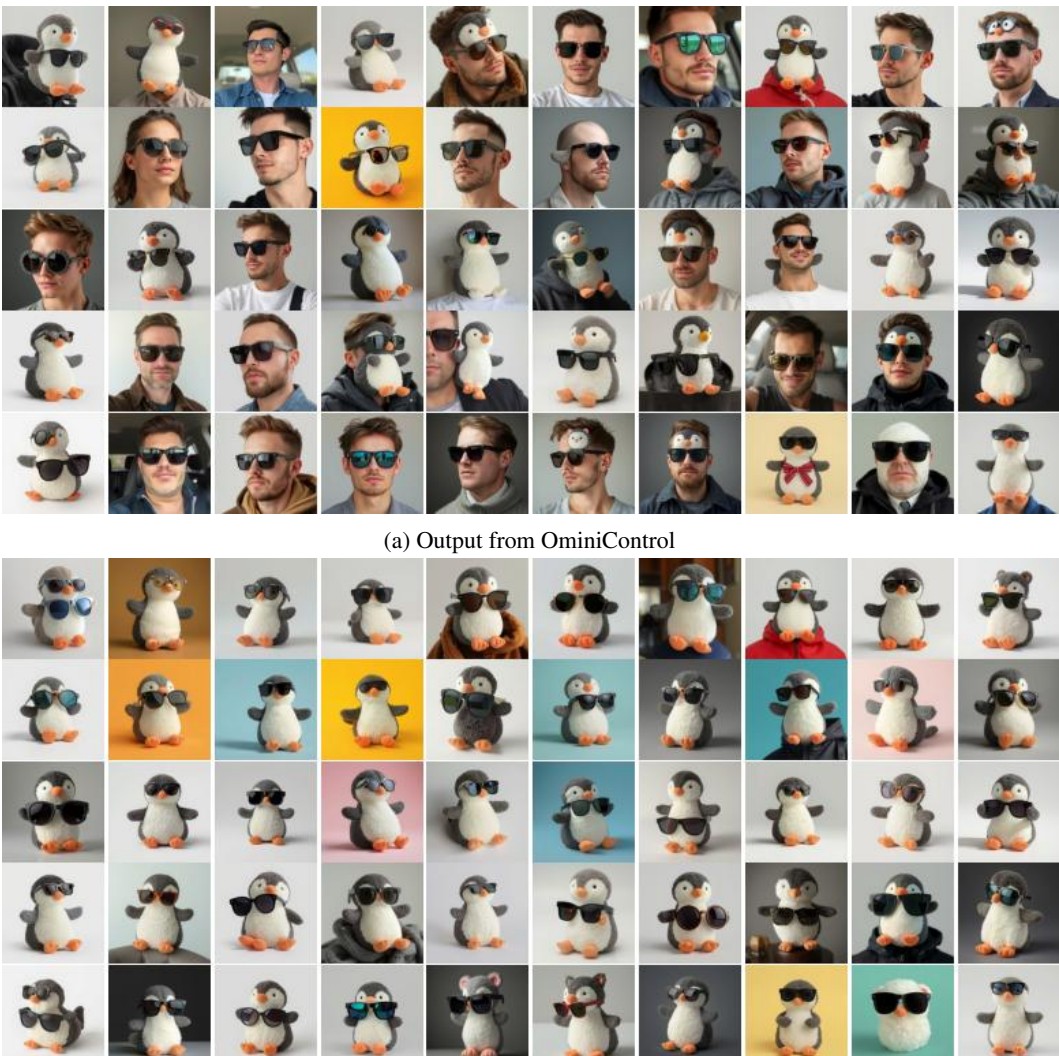

(a) Output from OminiControl

(b) Output from OminiControl with TEA

Figure 31: Comparison of the output from OminiControl pipeline with and without TEA with the same prompt: 'this item wearing glasses' and same random seed. OminiControl with TEA significantly improves the prompt fidelity, mitigating the failure cases of the original OminiControl pipeline, such as the subject (penguin) is not wearing glasses but a person does. More results showing the same improvement of OminiControl with TEA can be found in the repository at https://github.com/tuananhbui89/Embedding-Adjustment.

