# OpenReview forum: "Mitigating Semantic Collapse in Generative Personalization with Test-Time Embedding Adjustment"
_ICLR.cc/2026/Conference — ICLR 2026 Poster_

### Official Review · Reviewer_mTGU · 2025-10-28

**Soundness:** 3
**Presentation:** 3
**Contribution:** 3
**Rating:** 6
**Confidence:** 5

**Summary:**

The author define semantic collapse problem and propose a method TEA to mitigate it in test-time without re-training or altering embedding weights. Experiements show that TEA perform well on top of multiple personalizated models.

**Strengths:**

1. This paper provide a in-deep analysis about the root cause of SCP, claiming it's beacuse unconstrained optimisation, which make senses to this field.
2. The method is simple yet effective, though the idea that using the concept token to regularize the V* token is not novel, simply using SLERP at test-time is somehow interesting.
3. Comprehensive qualitative experiments shows the effectiveness of proposed TEA.

**Weaknesses:**

1. Novelty of SCP's definition: SCP is not an problem. that has not been discussed  What's the difference between SCP and semantic drift phenomenon discussed in ClassDiffusion[1]? According to the author's claim, they share a same phenomenon: can only generate "a photo of V*" when using the prompt "a photo of V* with other concepts". CoRe[2] also discuss similar problem. Also, some negative impace are somehow discussed in their paper.
2. Insufficient evaluation: as CLIP-T and CLIP-I often fail to provide fair evaluation about fine-grained results, could author provie other quantitative results about TEA's performance? Like using user study or VLMs as judge.


[1] Huang, Jiannan, et al. "ClassDiffusion: More Aligned Personalization Tuning with Explicit Class Guidance." The Thirteenth International Conference on Learning Representations.
[2] Wu, Feize, et al. "Core: Context-regularized text embedding learning for text-to-image personalization." Proceedings of the AAAI Conference on Artificial Intelligence. Vol. 39. No. 8. 2025.

**Questions:**

- What's CLIP$^{p}_T$ stands for? This metric is mentioned in Table 1, but not described in the experiment section.
- Do TEA's result align with author's discuss about SCP? Could author provide similar visulization/chart on models w/ TEA?

---

> ### Author Response · Authors · 2025-11-27
> **Response to Reviewer mTGU (1/n)**
>
> We thank the reviewer for the effort to review our paper. We appreciate your positive feedback on our work on the SCP problem and the TEA method. We would like to address the remaining concerns as follows:
>
> ### **What makes SCP different from semantic drift or previous works' discussion of similar issues?**
>
> We thank the reviewer for raising this important point. We fully agree that misalignment between personalized prompts and generated images is a known challenge, and we explicitly acknowledge this in the paper (Section 1 lines 70–73, Section 2.2 lines 146–147, Appendix A.3 lines 762–767).
> However, our contribution is not to rediscover this surface phenomenon, but to provide a fundamentally different problem definition, root-cause analysis, and solution. We clarify the differences below.
>
> **1. SCP defines a different phenomenon: collapse of the entire prompt embedding**
>
> ClassDiffusion and CoRe both describe the symptom that prompts like "a photo of
> V* with X" fail to incorporate X.
> However, their explanations focus on the drift of the learned token V* in isolation.
> More specifically, ClassDiffusion hypothesizes that V* drifts away from its superclass and therefore becomes hard to compose with other concepts.
> CoRe observes that token embeddings in a prompt shift more when the placeholder is replaced with V* compared to standard tokens like "man", "dog", etc.
>
> Crucially, neither work directly implies that the embedding of the entire prompt collapses to something equivalent to "a photo of V*".
> In contrast, our work not only explicitly defines this prompt-level semantic collapse, but also provides a quantitative measure of this collapse in the textual embedding and image spaces, which neither ClassDiffusion nor CoRe defines or quantifies.
>
> **2. A more comprehensive context-aware measurement to identify the problem**
>
> In ClassDiffusion, the authors verify their hypothesis using a t-SNE visualization to show how the learned keyword embedding V* far from the superclass concept c,
> while we use a more sophisticated approach, which measures the distance between two sets of prompts, with different distance metrics (Euclidean, Hausdorff, Mahalanobis, and KL divergence),
> to identify the problem of the semantic collapse more comprehensively and generalizable.
> Our measurement also allows to take the context of the prompt into account, which is not considered in ClassDiffusion.
> Modern LLMs or Text-to-Image Diffusion Models use contextualized text embeddings, in which the context of the prompt significantly influences the final representation of the prompt,
> therefore, our context-aware measurement is more comprehensive and generalizable.
>
> **3. SCP leads to different class of solutions**
>
> In ClassDiffusion, the authors propose a training-time cosine regularization loss to regularize the cosine similarity between the learned keyword embedding V* and the anchor concept c.
> In contrast, we propose a non-trivial inference-time embedding strategy to realign the learned embedding with its original semantic meaning,
> without altering the model weights or requiring any additional training and generalizes to all existing personalization frameworks.
> We already show in our paper that our method can effectively improve over ClassDiffusion itself in both image-alignment (DINO-I) and text-alignment (CLIP-f) metrics.
>
> **4. SCP provides new insights to Anti-DreamBooth**
>
> With the understanding of SCP, we provide an intriguing explanation for why Anti-Personalization frameworks like Anti-DreamBooth work and why TEA can mitigate them.
> To the best of our knowledge, this is the first work to uncover such a counter-intuitive vulnerability in Anti-DreamBooth,
> and provide a practical solution to mitigate it.

---

> ### Author Response · Authors · 2025-11-27
> **Response to Reviewer mTGU (2/n)**
>
> ### **Other quantitative evaluation metrics**
>
> We thank the reviewer for the suggestion. Following the recommendation, we incorporate two new VLM-based evaluation metrics, **VLM-P** and **VLM-I** using VLM (GPT-4o-mini in our experiments)
> as the judge. These metrics assess the two central aspects of generative personalization:
>
> - Prompt/Semantic Fidelity (with VLM-P): Given a pair of prompt P and generated image I, the VLM model evaluates how well I follows the prompt P in terms of scene composition, context, spatial relations, and actions.
>
> - Visual Fidelity (with VLM-I): Given a pair of reference image R and generated image I, the VLM model evaluates how well the subject in I visually matches the subject in R.
>
> Both metrics output a score between 0 and 4, where 0 means there are no correspondence between the generated image and the reference image (VLM-I) or input prompt (VLM-P),
> while 4 means the perfectly matches. The final score for each metric is obtained by averaging all inference prompts and samples (16 prompts times 50 random samples per concept/setting) and normalizing to the range [0, 100%].
> We include the full system prompts and evaluation scripts in the anonymous github repository.
>
> We evaluate these new metrics on EasyControl (the SOTA Flux-based personalization baseline already included in our paper).
> To further demonstrate the generality of TEA, we additionally conduct experiments on OminiControl [1], a recent state-of-the-art diffusion transformer–based control framework. The quantitative results are reported in the table below.
>
> The results show that TEA consistently improves the performance of both EasyControl and OminiControl across datasets, settings, and evaluation perspectives.
> These improvements are evident in both CLIP-based metrics and VLM-based metrics. For instance, in the Pet Dog setting with EasyControl, TEA improves CLIP-I by 3.23 points and DINO-I by 4.61 points, indicating **substantial gains in image-reference fidelity**.
> When evaluated with VLM-based metrics, TEA improves EasyControl by 2.25 points in VLM-P and 3.25 points in VLM-I, demonstrating consistent enhancement in both semantic and visual alignment.
> Similarly, in the Penguin setting with OminiControl, TEA improves VLM-P by 4.25 points and VLM-I by 3.50 points, as well as by 0.41 points in CLIP_T^f and 2.06 points in CLIP_I, demonstrating **the consistent improvement across different metrics**.
>
> In summary, in addition to the comprehensive quantitative results presented in Section 4.2 and Appendix C, these additional experiments on OminiControl and the newly incorporated VLM-based metrics further **reinforce the effectiveness, flexibility, and generalizability** of our method.
>
> [1] Tan, Z., Liu, S., Yang, X., Xue, Q., & Wang, X. (2025). Ominicontrol: Minimal and universal control for diffusion transformer. ICCV 2025.
>
> **Table 1: Performance improvement over EasyControl when integrating with our TEA.**
>
> | Method | CLIP_T^p ↑ | CLIP_T^f ↑ | CLIP-I ↑ | DINO-I ↑ | VLM-P ↑ | VLM-I ↑ |
> |--------|------------|------------|----------|----------|---------|---------|
> | **CC101 - Pet Dog** |
> | ES | 18.54 | 26.02 | 61.33 | 43.71 | 64.25 | 74.00 |
> | ES+TEA | 18.72 (+0.18) | 26.11 (+0.09) | 64.56 (+3.23) | 48.32 (+4.61) | 66.50 (+2.25) | 77.25 (+3.25) |
> | **CC101 - Plushie Teddybear** |
> | ES | 20.48 | 26.80 | 81.64 | 49.08 | 78.00 | 80.25 |
> | ES+TEA | 20.61 (+0.13) | 27.3 (+0.50) | 82.84 (+1.20) | 51.17 (+2.09) | 80.25 (+2.25) | 81.50 (+1.25) |
>
> ---
>
> **Table 2: Performance improvement over OminiControl when integrating with our TEA.**
>
> | Method | CLIP_T^p ↑ | CLIP_T^f ↑ | CLIP-I ↑ | DINO-I ↑ | VLM-P ↑ | VLM-I ↑ |
> |--------|------------|------------|----------|----------|---------|---------|
> | **Subject - Clock** |
> | Omini | 18.11 | 23.90 | 81.37 | 32.41 | 67.50 | 62.25 |
> | Omini+TEA | 18.78 (+0.67) | 23.98 (+0.08) | 83.10 (+1.73) | 34.48 (+2.07) | 71.75 (+4.25) | 64.50 (+2.25) |
> | **Subject - Oranges** |
> | Omini | 21.49 | 27.62 | 70.43 | 30.33 | 68.50 | 53.00 |
> | Omini+TEA | 21.60 (+0.11) | 27.70 (+0.08) | 71.90 (+1.47) | 31.64 (+1.31) | 70.00 (+1.50) | 55.50 (+2.50) |
> | **Subject - Penguin** |
> | Omini | 20.30 | 31.61 | 78.58 | 45.59 | 86.25 | 83.25 |
> | Omini+TEA | 20.33 (+0.03) | 32.02 (+0.41) | 80.64 (+2.06) | 49.37 (+3.78) | 90.50 (+4.25) | 86.75 (+3.50) |
>
> ---
>
> ### **What's CLIP_T^p stands for?**
>
> The metric CLIP_T^p has been defined in Section 2.3.2 (line 228 - 230) which measures the alignment between the generated image and the contextual part p of the prompt (e.g., "holding a cat"), to evaluate how well the generated image captures the contextual information.

---

> ### Author Response · Authors · 2025-11-29
> **Response to Reviewer mTGU (3/n, n=3)**
>
> ### **Do TEA's result align with author's discuss about SCP? Could author provide similar visualization/chart on models w/ TEA?**
>
> In the paper, we already provide comprehensive quantitative results (Section 4.2) and qualitative examples (Appendix D) that demonstrate TEA's effectiveness in mitigating the semantic collapse problem, thereby producing generated images that are more faithfully aligned with the input prompt. We believe these results offer strong evidence that TEA successfully addresses SCP in the image space.
>
> To further demonstrate the effectiveness of TEA on the text space, we measure how the adjusted embedding vector V̂* changes in the text encoder's latent space (Textual Inversion setting) after applying TEA. The updated analysis is now included in Figure 4 of the paper.
> We additionally conduct the same analysis in the DreamBooth setting, and the corresponding visualization has been added as Figure 21.
>
> It can be seen that, after applying TEA, the adjusted embedding vector V̂* consistently moves back toward the dense region occupied by normal tokens, which is precisely the expected behavior when mitigating the semantic collapse induced by unconstrained optimization.
>
> In summary, the combined evidence from both the image space (quantitative and qualitative) and the text space (embedding-space visualizations) strongly supports that TEA effectively mitigates the SCP problem.
>
> ### **Revised Paper with New Results**
>
> We have updated the paper with new experiments using OmniControl and newly integrated VLM-based evaluation metrics. All added discussions are clearly highlighted in orange for easy reference. We have also released the complete code and data in an anonymous GitHub repository. We hope these expanded results and transparent resources will make it easier for reviewers and the area chair to thoroughly assess—and reproduce—the strong empirical support for the effectiveness of TEA.

---

### Official Review · Reviewer_1iQw · 2025-10-30

**Soundness:** 2
**Presentation:** 2
**Contribution:** 2
**Rating:** 4
**Confidence:** 4

**Summary:**

This paper addresses the ​​Semantic Collapsing Problem in generative personalization, where personalized token embeddings drift significantly in both magnitude and direction during optimization, causing them to dominate multi-concept prompts and degrade compositional generation. The authors propose ​​Test-time Embedding Adjustment (TEA)​​, a training-free method that recalibrates the embedding’s norm and direction toward its original semantic concept during inference.

**Strengths:**

The identification and empirical analysis of SCP are interesting.

TEA is lightweight, requires no retraining, and is compatible with numerous existing frameworks.

The paper is well-structured and easy to follow.

**Weaknesses:**

While the Test-time Embedding Adjustment (TEA) method is practical and easy to deploy, its technical contribution is relatively modest. The core mechanism—adjusting the magnitude and direction of an embedding vector—is a straightforward application of existing vector space operations, lacking the novelty of a more transformative technique. The approach does not introduce new learning paradigms or architectural innovations, but rather applies a post-hoc correction to the outputs of existing models. This simplicity, though beneficial for accessibility, may limit the method's impact and long-term relevance in a fast-evolving field that often rewards more complex or foundational advancements.

The paper heavily relies on quantitative metrics like CLIP and DINO scores to demonstrate improvements, but provides limited qualitative evidence, especially for subject-driven customization tasks. For instance, while quantitative gains are reported, side-by-side visual comparisons showcasing TEA's enhancement over base methods (e.g., DreamBooth, Textual Inversion) across diverse concepts are sparse. More juxtaposed examples illustrating how TEA better preserves contextual details in complex prompts (e.g., "V* wearing glasses and writing in a red notebook") would strengthen the empirical claims. A broader set of qualitative results is necessary to help readers appreciate the practical improvements in output quality and alignment.

A thorough discussion of scenarios where TEA underperforms or introduces new artifacts is absent. The paper does not systematically analyze failure cases, such as conditions where the embedding adjustment might over-regularize and weaken the distinctiveness of the personalized concept. Furthermore, the limitations of only adjusting the embedding at inference time—potentially being a "shallow" fix rather than addressing optimization issues during training—are not critically examined. A dedicated section exploring these boundaries, possibly with examples of prompts where TEA fails to mitigate semantic collapse or even degrades output, would provide a more balanced and credible evaluation.

Although the effect of hyperparameters α (rotation factor) and β (scaling factor) is briefly studied, the analysis is conducted with a fixed base model and a limited set of prompts. The paper does not explore how these parameters interact with different model architectures (e.g., varying layers of U-Net or transformer-based text encoders) or diverse concept complexities. This narrow scope leaves open questions about the robustness and generalizability of the chosen default values across a wider range of real-world use cases.

**Questions:**

The core operations of TEA (normalization and SLERP) are geometrically intuitive but relatively simple. Were more complex adjustment strategies explored, such as a learnable linear transformation or a small neural network? What was the rationale for choosing this specific, non-adaptive formulation? The simplicity is a strength for deployment, but it raises questions about the exhaustiveness of the exploration.

The analysis in Appendix B on multi-concept personalization is intriguing but preliminary. When two personalized concepts, V_man and V_dog, compete in a prompt, TEA is applied to which concept? Both? Does applying it to both concepts restore a better balance, or does one still dominate? This is a critical scenario for assessing SCP and TEA's utility.

---

> ### Author Response · Authors · 2025-11-27
> **Response to Reviewer 1iQw (1/n)**
>
> We thank the reviewer for the effort to review our paper. We appreciate your positive feedback on our work on the SCP problem and the TEA method. We would like to address the remaining concerns as follows:
>
> ---
>
> ### **The approach does not introduce new learning paradigms or architecture innovation but rather applies a post-hoc correction to the outputs of existing models. This simplicity, through beneficial for accessibility, may limit the method's impact and long-term relevance in a fast-evolving field that often rewards more complex or foundational advancements. What was the rationale for choosing this specific, non-adaptive formulation?**
>
> We respectfully disagree with the reviewer's assessment. First, while it is correct that our method is training-free and does not introduce a new learning paradigm or architectural modification, we view this **simplicity as a core strength rather than a limitation**.
> In fact, TEA's training-free nature enables two key advantages that would be difficult for complex training-based or architecture-modification based methods to achieve:
>
> 1. **Broad applicability and seamless integration**.
>
> TEA can be immediately **integrated into almost any existing personalization framework** without requiring retraining or architectural changes.
> As demonstrated in the paper and extended in this rebuttal, **TEA consistently improves a diverse set of SOTA frameworks** including EasyControl, ClassDiffusion, ReVersion, and OminiControl, **despite their different training paradigms and architectures**.
> In contrast, approaches requiring retraining or architectural adjustments cannot be applied as universally, nor as effortlessly.
>
> 2. **Zero-cost solution for existing models**.
>
> A substantial amount of computational and human effort has already been invested into releasing thousands of pretrained personalization models. Many of these models might suffer from SCP, but re-training or redesigning them is infeasible in practice.
> Our inference-time embedding adjustment **offers a zero-cost, plug-in solution that improves their performance immediately**. More complex, training-heavy, or architecture-altering methods cannot serve this purpose.
>
> Second, we agree that the field evolves rapidly and that foundational innovations (e.g., ResNet, Transformer) have long-lasting impact. However, impactful contributions in generative modeling are not limited to architectural breakthroughs. **Empirical findings, new problem formulations, and improved understanding** of model behavior have historically shaped research directions and inspired subsequent foundational work. In this regard, **our paper contributes more than a post-hoc correction**:
>
> - We provide a **comprehensive analysis of the Semantic Collapse Problem (SCP)**, distinguishing it from prior loosely described phenomena.
>
> - We identify the root cause of SCP as the **unconstrained optimization characteristic** of personalization methods.
>
> - We connect SCP to **anti-personalization techniques** and clarify why such attacks succeed and how TEA naturally mitigates their effects.
>
> These insights are not merely observations; they **open clear avenues for future foundational advances**. For example, our diagnosis suggests that regularizing the deviation between learned embeddings and anchor embeddings could serve as a new training-time principle for personalization. Similarly, understanding the mechanism behind anti-DreamBooth enables the design of more robust adversarial defenses.
>
> In summary, while TEA is intentionally simple, its **generality, practicality, and the conceptual advances accompanying it** provide substantial value and relevance—both for real-world deployment and for guiding future research directions.
>
> ---
>
> ### **Limited qualitative evidence, especially for subject-driven customization tasks**
>
> We would like to respectfully remind the reviewer that we have provide a comprehensive qualitative results on the Appendix D of the paper.
> More specifically, we provide:
>
> - Correcting distorted generations (Figure 22) illustrates how SCP in DreamBooth leads to distorted generations, while TEA corrects the semantic embedding to produce more coherent and realistic images.
>
> - Cross-dataset comparisons (Figures 23, 24, 28, 29, 30), demonstrating the consistency of our method across diverse personalization frameworks and subject domains.
>
> - SCP under multi-concept prompts (Figures 25, 26, 27) demonstrate SCP when multiple concepts are combined.
>
> For Figures 23, 24, 28, 29, and 30, we have explicitly included the input prompts in the captions, along with references to the additional examples available in the anonymous repository.
>
> Taken together, these qualitative results offer substantial and diverse evidence of the effectiveness of our method. We therefore believe that the current presentation already provides sufficient qualitative support for the claims made in the paper.

---

> ### Author Response · Authors · 2025-11-27
> **Response to Reviewer 1iQw (2/n, n=2)**
>
> ---
>
> ### **Ablation Study on different model architectures or diverse concept complexities**
>
> We would like to clarify that our ablation study follows standard practice in analyzing the effect of individual hyperparameters by varying them while keeping other factors fixed. Exploring every possible combination of architectures, layers, and concept complexities is intractable and typically beyond the scope of ablation studies, whose goal is to isolate and understand the influence of specific parameters under representative settings.
>
> We agree that evaluating hyperparameter sensitivity across a broader landscape of architectures and concept complexities is an interesting direction. However, such exploration is orthogonal to the core objective of this paper and we believe it is better addressed in future work rather than within the scope of an ablation analysis.
>
> ---
>
> ### **TEA on Multiple Concept Prompts. TEA is applied to which concept? Both? Does applying it to both concepts restore a better balance, or does one still dominate?**
>
> We thank the reviewer for highlighting this intriguing aspect of our findings. In Appendix B, we specifically investigate the behavior of SCP in the context of multi-concept personalization. As an illustrative example, consider a prompt such as "A photo of a $V*_{man}$ touching a $V*_{dog}$ beside a park bench," where $V*_{man}$ and $V*_{dog}$ denote two independently personalized concepts.
>
> Our analysis reveals that SCP becomes even more pronounced in the multi-concept setting. When fixing $V*_{man}$ and varying $V*_{dog}$, we observe that at early personalization steps, $V*_{man}$ tends to dominate the generation, producing images centered almost entirely around the "man" concept.
> As personalization progresses and $V*_{dog}$ acquires more distinctive features, the dominance reverses: the "dog" concept begins to overshadow $V*_{man}$, yielding generations that primarily depict the dog while suppressing the other concept. This progression indicates that SCP not only persists but may intensify when multiple learned embeddings interact within a single prompt.
>
> Regarding the application of TEA, this experiment was designed to characterize the phenomenon rather than to propose a definitive multi-concept correction strategy. TEA can in principle be applied to either individual concept embeddings or both simultaneously; however, establishing a systematic procedure for balancing multiple personalized concepts—especially when they exhibit dynamic dominance—is non-trivial. Challenges include defining robust evaluation metrics for multi-concept fidelity, determining concept-interaction dynamics, and accommodating the diverse range of concepts that personalization systems may encounter.
> Given these complexities, we view multi-concept TEA as a promising and important direction for future work, but one that is beyond the scope of the current paper.

---

> ### Author Response · Authors · 2025-11-27
> **Additional experiments with OminiControl**
>
> We would like to further validate the generalization of our method by conducting additional experiments on OminiControl [1], a recent SOTA image control generation framework.
>
> We also incorporate two new VLM-based evaluation metrics, **VLM-P** and **VLM-I** using VLM (GPT-4o-mini in our experiments)
> as the judge. These metrics assess the two central aspects of generative personalization:
>
> - Prompt/Semantic Fidelity (with VLM-P): Given a pair of prompt P and generated image I, the VLM model evaluates how well I follows the prompt P in terms of scene composition, context, spatial relations, and actions.
>
> - Visual Fidelity (with VLM-I): Given a pair of reference image R and generated image I, the VLM model evaluates how well the subject in I visually matches the subject in R.
>
> Both metrics output a score between 0 and 4, where 0 means there are no correspondence between the generated image and the reference image (VLM-I) or input prompt (VLM-P),
> while 4 means the perfectly matches. The final score for each metric is obtained by averaging all inference prompts and samples (16 prompts times 50 random samples per concept/setting) and normalizing to the range [0, 100%].
> We include the full system prompts and evaluation scripts in the anonymous github repository.
>
> The results show that **TEA consistently improves the performance of both EasyControl and OminiControl** across datasets, settings, and evaluation perspectives.
> These improvements are evident in both CLIP-based metrics and VLM-based metrics. For instance, in the Pet Dog setting with EasyControl, TEA improves CLIP-I by 3.23 points and DINO-I by 4.61 points, indicating **substantial gains in image-reference fidelity**.
> When evaluated with VLM-based metrics, TEA improves EasyControl by 2.25 points in VLM-P and 3.25 points in VLM-I, demonstrating consistent enhancement in both semantic and visual alignment.
> Similarly, in the Penguin setting with OminiControl, TEA improves VLM-P by 4.25 points and VLM-I by 3.50 points, as well as by 0.41 points in CLIP_T^f and 2.06 points in CLIP_I, demonstrating the consistent improvement across different metrics.
>
> In summary, in addition to the comprehensive quantitative results presented in Section 4.2 and Appendix C, these additional experiments on OminiControl and the newly incorporated VLM-based metrics further reinforce the effectiveness, flexibility, and generalizability of our method.
>
> [1] Tan, Z., Liu, S., Yang, X., Xue, Q., & Wang, X. (2025). Ominicontrol: Minimal and universal control for diffusion transformer. ICCV 2025.
>
>
> **Table 1: Performance improvement over EasyControl when integrating with our TEA.**
>
> | Method | CLIP_T^p ↑ | CLIP_T^f ↑ | CLIP-I ↑ | DINO-I ↑ | VLM-P ↑ | VLM-I ↑ |
> |--------|------------|------------|----------|----------|---------|---------|
> | **CC101 - Pet Dog** |
> | ES | 18.54 | 26.02 | 61.33 | 43.71 | 64.25 | 74.00 |
> | ES+TEA | 18.72 (+0.18) | 26.11 (+0.09) | 64.56 (+3.23) | 48.32 (+4.61) | 66.50 (+2.25) | 77.25 (+3.25) |
> | **CC101 - Plushie Teddybear** |
> | ES | 20.48 | 26.80 | 81.64 | 49.08 | 78.00 | 80.25 |
> | ES+TEA | 20.61 (+0.13) | 27.3 (+0.50) | 82.84 (+1.20) | 51.17 (+2.09) | 80.25 (+2.25) | 81.50 (+1.25) |
>
> ---
>
> **Table 2: Performance improvement over OminiControl when integrating with our TEA.**
>
> | Method | CLIP_T^p ↑ | CLIP_T^f ↑ | CLIP-I ↑ | DINO-I ↑ | VLM-P ↑ | VLM-I ↑ |
> |--------|------------|------------|----------|----------|---------|---------|
> | **Subject - Clock** |
> | Omini | 18.11 | 23.90 | 81.37 | 32.41 | 67.50 | 62.25 |
> | Omini+TEA | 18.78 (+0.67) | 23.98 (+0.08) | 83.10 (+1.73) | 34.48 (+2.07) | 71.75 (+4.25) | 64.50 (+2.25) |
> | **Subject - Oranges** |
> | Omini | 21.49 | 27.62 | 70.43 | 30.33 | 68.50 | 53.00 |
> | Omini+TEA | 21.60 (+0.11) | 27.70 (+0.08) | 71.90 (+1.47) | 31.64 (+1.31) | 70.00 (+1.50) | 55.50 (+2.50) |
> | **Subject - Penguin** |
> | Omini | 20.30 | 31.61 | 78.58 | 45.59 | 86.25 | 83.25 |
> | Omini+TEA | 20.33 (+0.03) | 32.02 (+0.41) | 80.64 (+2.06) | 49.37 (+3.78) | 90.50 (+4.25) | 86.75 (+3.50) |
>
> ---

---

### Official Review · Reviewer_AfeA · 2025-11-01

**Soundness:** 2
**Presentation:** 3
**Contribution:** 2
**Rating:** 4
**Confidence:** 3

**Summary:**

This paper introduces the "Semantic Collapsing Problem" in generative personalization, where a model over-focuses on a learned subject (e.g., a specific person) and neglects other contextual details in a prompt. The authors propose a simple, training-free solution called Test-time Embedding Adjustment (TEA) that corrects the subject's text embedding at inference time. This method significantly improves text-image alignment across various personalization techniques and surprisingly reveals a vulnerability in current anti-personalization defenses.

**Strengths:**

1.	High quality of writing and presentation.
2.	The introduction of TEA, a method that is training-free and easily transferable.
3.	An insightful analysis of the "Semantic Collapsing Problem" within generative personalization and the mechanics of anti-dreambooth methods.

**Weaknesses:**

1.	The paper lacks comparisons to recent, mainstream works, particularly those in the in-context generation paradigm (e.g., OminiControl[1], FLUX.1 Kontext[2], and Diffusion Self-Distillation[3]).
2.	The evaluation is insufficient. It should be strengthened by including MLLM-based benchmarks, such as DreamBench++[4].
3.	Table 1 shows a performance decrease in Reference and Image alignment scores. This is presumably because TEA's objective focuses exclusively on fidelity between the learned and original subjects, potentially neglecting other alignment aspects.
4.	The choice of SLERP is not justified. The paper provides no evidence or ablation study to demonstrate that SLERP is superior to other interpolation methods.

[1] Tan, Z., Liu, S., Yang, X., Xue, Q., & Wang, X. (2025). Ominicontrol: Minimal and universal control for diffusion transformer. In Proceedings of the IEEE/CVF International Conference on Computer Vision (pp. 14940-14950).
[2] Batifol, S., Blattmann, A., Boesel, F., Consul, S., Diagne, C., Dockhorn, T., ... & Smith, L. (2025). FLUX. 1 Kontext: Flow Matching for In-Context Image Generation and Editing in Latent Space. arXiv e-prints, arXiv-2506.
[3] Cai, S., Chan, E. R., Zhang, Y., Guibas, L., Wu, J., & Wetzstein, G. (2025). Diffusion self-distillation for zero-shot customized image generation. In Proceedings of the Computer Vision and Pattern Recognition Conference (pp. 18434-18443).
[4] Peng, Y., Cui, Y., Tang, H., Qi, Z., Dong, R., Bai, J., ... & Xia, S. T. (2024). Dreambench++: A human-aligned benchmark for personalized image generation. arXiv preprint arXiv:2406.16855.

**Questions:**

1.	How does the paper demonstrate that the method genuinely improves performance, rather than simply acting as a trade-off between instruction alignment and reference image alignment?
2.	To better understand the model's behavior, could the authors provide results for prompts that include the subject class (e.g., "a photo of V* dog"), not just the identifier "V*" alone?
3.	What is the specific effect of the prior preservation loss on the semantic collapsing phenomenon?

---

> ### Author Response · Authors · 2025-11-27
> **Response to Reviewer AfeA (1/n)**
>
> We thank the reviewer for the effort to review our paper. We appreciate your positive feedback on the practicality and effectiveness of TEA and the intriguing findings on the SCP problem.
> We would like to address the remaining concerns as follows:
>
> ---
>
> ### **The paper lacks comparisons to recent, mainstream works, particularly those in the in-context generation paradigm ...**
>
> We respectfully remind the reviewer that in our paper, we already included experiments with EasyControl, a recent SOTA image control generation framework, similar to the suggested works.
> However, we follow the reviewer's suggestion to include additional experiments with OminiControl, a recent SOTA image control generation framework, to further validate the generalization of our method.
>
> We also incorporate two new VLM-based evaluation metrics, **VLM-P** and **VLM-I** using VLM (GPT-4o-mini in our experiments)
> as the judge. These metrics assess the two central aspects of generative personalization:
>
> - Prompt/Semantic Fidelity (with VLM-P): Given a pair of prompt P and generated image I, the VLM model evaluates how well I follows the prompt P in terms of scene composition, context, spatial relations, and actions.
>
> - Visual Fidelity (with VLM-I): Given a pair of reference image R and generated image I, the VLM model evaluates how well the subject in I visually matches the subject in R.
>
> Both metrics output a score between 0 and 4, where 0 means there are no correspondence between the generated image and the reference image (VLM-I) or input prompt (VLM-P),
> while 4 means the perfectly matches. The final score for each metric is obtained by averaging all inference prompts and samples (16 prompts times 50 random samples per concept/setting) and normalizing to the range [0, 100%].
> We include the full system prompts and evaluation scripts in the anonymous github repository.
>
> The results show that **TEA consistently improves the performance of both EasyControl and OminiControl across datasets**, settings, and evaluation perspectives.
> These improvements are evident in both CLIP-based metrics and VLM-based metrics. For instance, in the Pet Dog setting with EasyControl, TEA improves CLIP-I by 3.23 points and DINO-I by 4.61 points, indicating **substantial gains in image-reference fidelity**.
> When evaluated with VLM-based metrics, TEA improves EasyControl by 2.25 points in VLM-P and 3.25 points in VLM-I, demonstrating consistent enhancement in both semantic and visual alignment.
> Similarly, in the Penguin setting with OminiControl, TEA improves VLM-P by 4.25 points and VLM-I by 3.50 points, as well as by 0.41 points in CLIP_T^f and 2.06 points in CLIP_I, demonstrating the consistent improvement across different metrics.
>
> In summary, in addition to the comprehensive quantitative results presented in Section 4.2 and Appendix C, these additional experiments on OminiControl and the newly incorporated VLM-based metrics further **reinforce the effectiveness, flexibility, and generalizability of our method**.
>
> [1] Tan, Z., Liu, S., Yang, X., Xue, Q., & Wang, X. (2025). Ominicontrol: Minimal and universal control for diffusion transformer. ICCV 2025.
>
>
> **Table 1: Performance improvement over EasyControl when integrating with our TEA.**
>
> | Method | CLIP_T^p ↑ | CLIP_T^f ↑ | CLIP-I ↑ | DINO-I ↑ | VLM-P ↑ | VLM-I ↑ |
> |--------|------------|------------|----------|----------|---------|---------|
> | **CC101 - Pet Dog** |
> | ES | 18.54 | 26.02 | 61.33 | 43.71 | 64.25 | 74.00 |
> | ES+TEA | 18.72 (+0.18) | 26.11 (+0.09) | 64.56 (+3.23) | 48.32 (+4.61) | 66.50 (+2.25) | 77.25 (+3.25) |
> | **CC101 - Plushie Teddybear** |
> | ES | 20.48 | 26.80 | 81.64 | 49.08 | 78.00 | 80.25 |
> | ES+TEA | 20.61 (+0.13) | 27.3 (+0.50) | 82.84 (+1.20) | 51.17 (+2.09) | 80.25 (+2.25) | 81.50 (+1.25) |
>
> ---
>
> **Table 2: Performance improvement over OminiControl when integrating with our TEA.**
>
> | Method | CLIP_T^p ↑ | CLIP_T^f ↑ | CLIP-I ↑ | DINO-I ↑ | VLM-P ↑ | VLM-I ↑ |
> |--------|------------|------------|----------|----------|---------|---------|
> | **Subject - Clock** |
> | Omini | 18.11 | 23.90 | 81.37 | 32.41 | 67.50 | 62.25 |
> | Omini+TEA | 18.78 (+0.67) | 23.98 (+0.08) | 83.10 (+1.73) | 34.48 (+2.07) | 71.75 (+4.25) | 64.50 (+2.25) |
> | **Subject - Oranges** |
> | Omini | 21.49 | 27.62 | 70.43 | 30.33 | 68.50 | 53.00 |
> | Omini+TEA | 21.60 (+0.11) | 27.70 (+0.08) | 71.90 (+1.47) | 31.64 (+1.31) | 70.00 (+1.50) | 55.50 (+2.50) |
> | **Subject - Penguin** |
> | Omini | 20.30 | 31.61 | 78.58 | 45.59 | 86.25 | 83.25 |
> | Omini+TEA | 20.33 (+0.03) | 32.02 (+0.41) | 80.64 (+2.06) | 49.37 (+3.78) | 90.50 (+4.25) | 86.75 (+3.50) |
>
> ---

---

> ### Author Response · Authors · 2025-11-27
> **Response to Reviewer AfeA (2/n)**
>
> ### **The evaluation is insufficient**
>
> We respectfully disagree with the reviewer's concern regarding the sufficiency of our evaluation. Our paper conducts experiments on a broad and representative suite of datasets widely used in the generative personalization literature. Specifically, we evaluate on nine datasets from the CustomConcept101 (CC101) benchmark and ten identities from the CelebA-HQ dataset, along with **three relational datasets** introduced in the ReVersion paper—yielding a **total of 22 distinct datasets**.
>
> To further strengthen the evidence, our rebuttal includes additional experiments on **three datasets from OminiControl** (Clock, Oranges, and Penguin), demonstrating that TEA generalizes effectively even across frameworks with fundamentally different personalization mechanisms.
>
> We therefore believe that the breadth and diversity of the evaluated datasets, together with the supplemental results provided in the rebuttal, offer a sufficiently comprehensive assessment of the effectiveness and generalization capability of our method.
>
> ---
>
> ### **Table 1 shows a performance decrease in Reference and Image alignment scores. How does the paper demonstrate that the method genuinely improves performance?**
>
> We would like to clarify and contextualize the results presented in Table 1. On the CC101 dataset, with respect to reference-image fidelity, our method outperforms three out of four baselines, with Textual Inversion (TI) being the only exception.
> On the CelebA-HQ dataset, our method outperforms ClassDiffusion (CD), achieves comparable performance to Custom Latent (CL), and shows lower fidelity compared to TI and DreamBooth (DB). In Table 2, **TEA consistently outperforms EasyControl across all three evaluation settings**.
>
> Beyond the results reported in the paper, our rebuttal further includes experiments on OminiControl (Clock, Oranges, and Penguin), where TEA again surpasses the original method across all evaluation settings.
> Overall, across all datasets and baselines considered, **TEA outperforms 6 out of 10 methods, ties with one,** and is outperformed by three—demonstrating competitive and robust performance across diverse personalization frameworks.
>
> **Regarding text–image alignment**, TEA delivers consistently strong improvements: **it outperforms all baselines (10/10 methods)** across all datasets.
> Enhancing text alignment is the central goal of TEA, and these improvements directly support the core contribution of our work.
>
> While certain baselines such as TI and DB show higher reference fidelity in specific cases, we note that SCP often introduces severe distortions or unintended artifacts in the generated images—akin to the effects of Anti-Personalization methods but emerging implicitly due to unconstrained optimization and dataset bias.
> As illustrated in Figure 22 of the paper, TEA effectively mitigates these distortions, restoring the semantic integrity of the output while preserving the core characteristics of the personalized concept.
>
> Taken together, these quantitative and qualitative results show that **TEA provides consistent improvements in text–image alignment, alleviates SCP-induced distortions, and offers competitive reference fidelity**—thereby demonstrating clear and meaningful performance gains.

---

> ### Author Response · Authors · 2025-11-27
> **Response to Reviewer AfeA (3/n, n=3)**
>
> ### **The choice of SLERP is not justified**
>
> We would like to clarify that our choice of SLERP is principled rather than arbitrary.
> As discussed in Section 2.3.3, SCP arises from the unconstrained optimization in personalization, which causes the learned embedding to drift in the latent space—both in magnitude and direction. Our formulation directly addresses these two failure modes.
>
> Motivated by this geometric insight, we propose using SLERP at inference time for embedding adjustment. **SLERP consists of two components—normalization and rotation—which correspond precisely to the issues identified above**:
> (1) the normalization step corrects the uncontrolled change in magnitude;
> (2) the rotation step moves the embedding toward the direction of the reference concept, counteracting direction drift.
> Thus, **SLERP intentionally operationalizes the geometric structure uncovered in our SCP analysis**.
>
> Secondly, as discussed in Section 3 and supported by Shoemake's work [1], **SLERP is known to produce smoother and more stable transitions than linear interpolation in high-dimensional vector spaces**. This property is particularly relevant in our setting because standard tokens in the text encoder lie approximately on a normalized hypersphere, whereas linear interpolation cuts through the interior of this hypersphere. This leads to undesirable non-smooth trajectories and inconsistent semantic behavior—precisely what we aim to avoid in mitigating SCP.
>
> For these reasons, SLERP is not only justified but naturally aligned with both the theoretical analysis and empirical characteristics of the embedding space.
>
> [1] Shoemake, Ken. "Animating rotation with quaternion curves." ACM SIGGRAPH Computer Graphics. Vol. 19. No. 3. 1985.
>
> ---
>
> ### **What is the specific effect of the prior preservation loss on the semantic collapsing phenomenon?**
>
> We thank the reviewer for this very insightful question. In our paper, we provide a detailed analysis of the semantic collapsing phenomenon, where a prompt of the form ⌊p, V*⌋ collapses into a simplified representation ⌊V*⌋, with the learned embedding V* dominating and overriding the contextual prompt p.
>
> The prior preservation loss was originally introduced to mitigate overfitting during personalization. Its goal is to ensure that the model retains the ability to generate the original concept c.
> While this loss does succeed to some extent in preserving the original concept, we hypothesize that it may inadvertently contribute to semantic collapse, τ_i(⌊p, c⌋) → τ_i(⌊c⌋).
>
> where τ_i denotes the text encoder output at personalization step i, encourages the model to gradually disregard the contextual component p. This dynamic parallels the root cause of SCP for the original token c: unconstrained optimization combined with highly repetitive and structurally simple prompt templates (e.g., "a photo of a man") used during personalization.
>
> Unlike the rich and diverse captions that the base text-to-image model is trained on, personalization relies on extremely homogeneous text inputs repeated across thousands of iterations, which can promote collapse into overly simplified semantic regions.
>
> However, because personalization aims to generate images of the new concept V* rather than the base concept c, the collapse of c often goes unnoticed—and is therefore rarely examined.

---

### Official Review · Reviewer_WZt3 · 2025-11-01

**Soundness:** 2
**Presentation:** 2
**Contribution:** 3
**Rating:** 4
**Confidence:** 4

**Summary:**

This paper pioneers the definition of the Semantic Collapsing Problem (SCP)—a long-overlooked issue where personalized tokens lose their original textual semantics and dominate complex prompts, with unconstrained optimization identified as its root cause. It then proposes Test-time Embedding Adjustment (TEA), a lightweight, training-free solution that aligns the magnitude and direction of personalized embeddings with their original semantic concepts at inference, effectively mitigating SCP. TEA demonstrates strong generality, enhancing text-image alignment across diverse personalization frameworks (e.g., Textual Inversion, DreamBooth).

**Strengths:**

The work proposes the Semantic Collapsing Problem (SCP) in generative personalization—an under-explored issue—and rigorously identifying unconstrained optimization as its root cause, with solid empirical evidence across textual and image spaces.
The proposed TEA method is lightweight and practical: it requires no additional training, avoids modifying model weights, and generalizes well across diverse frameworks (e.g., Textual Inversion, DreamBooth) and architectures (Stable Diffusion, Flux), making it easy to integrate into existing pipelines.
The unexpected finding that TEA mitigates Anti-DreamBooth’s adversarial corruption adds novel insights to privacy-utility tradeoffs in personalization, uncovering vulnerabilities in current anti-personalization defenses.

**Weaknesses:**

TEA relies on fixed hyperparameters (α=0.2, β=1.5) across all prompts, which may not be optimal for diverse scenario.

**Questions:**

How to determine the  rotation factor for each customization process?
Why TI+TEA and DB+TEA will degrade the customized image's alignment with reference？
why the author choose SLERP interpolation to adjust the embedding?

---

> ### Author Response · Authors · 2025-11-27
> **Response to Reviewer WZt3 (1/n)**
>
> We thank the reviewer for the effort to review our paper. We appreciate your positive feedback on the practicality and effectiveness of TEA and the intriguing findings on the SCP problem. We would like to address the remaining concerns as follows:
>
> ---
>
> ### **How to determine the rotation factor for each customization process?**
>
> We thank the reviewer for the question. In our experiments, we do not tune the rotation factor for each individual setting. Instead, we adopt a single fixed value α = 0.2 across all models, concepts, and datasets.
>
> Our goal in this work is not to optimize α for every case, but to demonstrate that the SCP problem exists and that a simple test-time embedding adjustment can effectively mitigate it. Designing a setting-specific or concept-specific strategy for selecting α is indeed an interesting research direction, but it is orthogonal to our core contribution.
>
> Importantly, despite using the same fixed rotation factor everywhere, our method already yields consistent and robust improvements across major settings even with SOTA personalization frameworks, as shown in the quantitative evaluations in Section 4.2 and the extended results in Appendix C. This suggests that the method is not overly sensitive to hyperparameter tuning and that a single default value can generalize well in practice.
>
> We believe that exploring adaptive or learned rotation factors could further enhance performance, and we highlight this as promising future work.
>
> ---
>
> ### **Why TI+TEA and DB+TEA will degrade the customized image's alignment with reference?**
>
> We would like to clarify and contextualize the results presented in Table 1. On the CC101 dataset, with respect to reference-image fidelity, our method outperforms three out of four baselines, with Textual Inversion (TI) being the only exception.
> On the CelebA-HQ dataset, our method outperforms ClassDiffusion (CD), achieves comparable performance to Custom Latent (CL), and shows lower fidelity compared to TI and DreamBooth (DB). In Table 2, TEA consistently outperforms EasyControl across all three evaluation settings.
>
> Beyond the results reported in the paper, our rebuttal further includes experiments on OminiControl (Clock, Oranges, and Penguin), where TEA again surpasses the original method across all evaluation settings.
>
> Overall, across all datasets and baselines considered, **TEA outperforms 6 out of 10 methods, ties with one,** and is outperformed by three—demonstrating competitive and robust performance across diverse personalization frameworks.
>
> **Regarding text–image alignment**, TEA delivers consistently strong improvements: **it outperforms all baselines (10/10 methods)** across all datasets.
> Enhancing text alignment is the central goal of TEA, and these improvements directly support the core contribution of our work.
>
> While certain baselines such as TI and DB show higher reference fidelity in specific cases, we note that SCP often introduces severe distortions or unintended artifacts in the generated images—akin to the effects of Anti-Personalization methods but emerging implicitly due to unconstrained optimization and dataset bias.
> As illustrated in Figure 22 of the paper, TEA effectively mitigates these distortions, restoring the semantic integrity of the output while preserving the core characteristics of the personalized concept.
>
> Taken together, these quantitative and qualitative results show that TEA provides consistent improvements in text–image alignment, alleviates SCP-induced distortions, and offers competitive reference fidelity—thereby demonstrating clear and meaningful performance gains.

---

> ### Author Response · Authors · 2025-11-27
> **Response to Reviewer WZt3 (2/n, n=2)**
>
> ### **Why the author choose SLERP interpolation to adjust the embedding?**
>
> We would like to clarify that our choice of SLERP is principled rather than arbitrary.
> As discussed in Section 2.3.3, SCP arises from the unconstrained optimization in personalization, which causes the learned embedding to drift in the latent space—both in magnitude and direction. Our formulation directly addresses these two failure modes.
>
> Motivated by this geometric insight, we propose using SLERP at inference time for embedding adjustment. SLERP consists of two components—normalization and rotation—which correspond precisely to the issues identified above:
> (1) the normalization step corrects the uncontrolled change in magnitude;
> (2) the rotation step moves the embedding toward the direction of the reference concept, counteracting direction drift.
> Thus, SLERP intentionally operationalizes the geometric structure uncovered in our SCP analysis.
>
> Secondly, as discussed in Section 3 and supported by Shoemake's work [1], SLERP is known to produce smoother and more stable transitions than linear interpolation in high-dimensional vector spaces. This property is particularly relevant in our setting because standard tokens in the text encoder lie approximately on a normalized hypersphere, whereas linear interpolation cuts through the interior of this hypersphere. This leads to undesirable non-smooth trajectories and inconsistent semantic behavior—precisely what we aim to avoid in mitigating SCP.
>
> For these reasons, SLERP is not only justified but naturally aligned with both the theoretical analysis and empirical characteristics of the embedding space.
>
> [1] Shoemake, Ken. "Animating rotation with quaternion curves." ACM SIGGRAPH Computer Graphics. Vol. 19. No. 3. 1985.

---

> ### Author Response · Authors · 2025-11-27
> **Additional experiments with OminiControl**
>
> We would like to further validate the generalization of our method by conducting additional experiments on OminiControl [1], a recent SOTA image control generation framework.
>
> We also incorporate two new VLM-based evaluation metrics, **VLM-P** and **VLM-I** using VLM (GPT-4o-mini in our experiments)
> as the judge. These metrics assess the two central aspects of generative personalization:
>
> - Prompt/Semantic Fidelity (with VLM-P): Given a pair of prompt P and generated image I, the VLM model evaluates how well I follows the prompt P in terms of scene composition, context, spatial relations, and actions.
>
> - Visual Fidelity (with VLM-I): Given a pair of reference image R and generated image I, the VLM model evaluates how well the subject in I visually matches the subject in R.
>
> Both metrics output a score between 0 and 4, where 0 means there are no correspondence between the generated image and the reference image (VLM-I) or input prompt (VLM-P),
> while 4 means the perfectly matches. The final score for each metric is obtained by averaging all inference prompts and samples (16 prompts times 50 random samples per concept/setting) and normalizing to the range [0, 100%].
> We include the full system prompts and evaluation scripts in the anonymous github repository.
>
> The results show that **TEA consistently improves the performance of both EasyControl and OminiControl** across datasets, settings, and evaluation perspectives.
> These improvements are evident in both CLIP-based metrics and VLM-based metrics. For instance, in the Pet Dog setting with EasyControl, TEA improves CLIP-I by 3.23 points and DINO-I by 4.61 points, indicating **substantial gains in image-reference fidelity**.
> When evaluated with VLM-based metrics, TEA improves EasyControl by 2.25 points in VLM-P and 3.25 points in VLM-I, demonstrating consistent enhancement in both semantic and visual alignment.
> Similarly, in the Penguin setting with OminiControl, TEA improves VLM-P by 4.25 points and VLM-I by 3.50 points, as well as by 0.41 points in CLIP_T^f and 2.06 points in CLIP_I, demonstrating the consistent improvement across different metrics.
>
> In summary, in addition to the comprehensive quantitative results presented in Section 4.2 and Appendix C, these additional experiments on OminiControl and the newly incorporated VLM-based metrics further reinforce the effectiveness, flexibility, and generalizability of our method.
>
> [1] Tan, Z., Liu, S., Yang, X., Xue, Q., & Wang, X. (2025). Ominicontrol: Minimal and universal control for diffusion transformer. ICCV 2025.
>
>
> **Table 1: Performance improvement over EasyControl when integrating with our TEA.**
>
> | Method | CLIP_T^p ↑ | CLIP_T^f ↑ | CLIP-I ↑ | DINO-I ↑ | VLM-P ↑ | VLM-I ↑ |
> |--------|------------|------------|----------|----------|---------|---------|
> | **CC101 - Pet Dog** |
> | ES | 18.54 | 26.02 | 61.33 | 43.71 | 64.25 | 74.00 |
> | ES+TEA | 18.72 (+0.18) | 26.11 (+0.09) | 64.56 (+3.23) | 48.32 (+4.61) | 66.50 (+2.25) | 77.25 (+3.25) |
> | **CC101 - Plushie Teddybear** |
> | ES | 20.48 | 26.80 | 81.64 | 49.08 | 78.00 | 80.25 |
> | ES+TEA | 20.61 (+0.13) | 27.3 (+0.50) | 82.84 (+1.20) | 51.17 (+2.09) | 80.25 (+2.25) | 81.50 (+1.25) |
>
> ---
>
> **Table 2: Performance improvement over OminiControl when integrating with our TEA.**
>
> | Method | CLIP_T^p ↑ | CLIP_T^f ↑ | CLIP-I ↑ | DINO-I ↑ | VLM-P ↑ | VLM-I ↑ |
> |--------|------------|------------|----------|----------|---------|---------|
> | **Subject - Clock** |
> | Omini | 18.11 | 23.90 | 81.37 | 32.41 | 67.50 | 62.25 |
> | Omini+TEA | 18.78 (+0.67) | 23.98 (+0.08) | 83.10 (+1.73) | 34.48 (+2.07) | 71.75 (+4.25) | 64.50 (+2.25) |
> | **Subject - Oranges** |
> | Omini | 21.49 | 27.62 | 70.43 | 30.33 | 68.50 | 53.00 |
> | Omini+TEA | 21.60 (+0.11) | 27.70 (+0.08) | 71.90 (+1.47) | 31.64 (+1.31) | 70.00 (+1.50) | 55.50 (+2.50) |
> | **Subject - Penguin** |
> | Omini | 20.30 | 31.61 | 78.58 | 45.59 | 86.25 | 83.25 |
> | Omini+TEA | 20.33 (+0.03) | 32.02 (+0.41) | 80.64 (+2.06) | 49.37 (+3.78) | 90.50 (+4.25) | 86.75 (+3.50) |
>
> ---

---

### Author Response · Authors · 2025-12-02
**Global Response**

We thank the Area Chair and the reviewers for the effort to review our paper. We appreciate your positive feedback on our work and your acknowledgements on the strengths and contributions made in our paper, which we humbly reiterate below for reference.

We propose a **Test-Time Embedding Adjustment** (TEA) method to address the **Semantic Collapsing Problem** (SCP) in generative personalization.

One of the core strengths of our method lies in its **simplicity**—being entirely **training-free**—and its **generality**, allowing **seamless integration with almost all existing personalization frameworks**. Our approach is compatible with both token-based methods (e.g., Textual Inversion, ReVersion) and weight-based methods (e.g., DreamBooth, Custom Diffusion, EasyControl, OminiControl), **requiring no changes to their fine-tuning/training pipelines but just a pretrained model**.

Despite the simplicity, our method has been shown to be **consistently effective** in improving the performance of personalization frameworks even with the recent SOTA ones as demonstrated in our paper and the additional experiments in this rebuttal, such as ClassDiffusion (ICLR 2025), EasyControl (ICCV 2025), ReVersion (SIGGRAPH Asia 2024), and OminiControl (ICCV 2025).

Our method is **well-grounded** in a comprehensive analysis of the Semantic Collapsing Problem (SCP). By identifying and diagnosing the root cause of SCP, which is the unconstrainted optimization in personalization, we provide not only a clear explanation of the failure mode but also an intuitive understanding of why our solution is both effective and broadly applicable.

Moreover, we bring a **new perspective to explain why Anti-Personalization frameworks** like Anti-DreamBooth work from the perspective of SCP. We find that when applying TEA to models poisoned by Anti-DreamBooth, we can partially reverse adversarial corruption and restore more faithful generations of the protected concept. This finding suggests that current defenses may offer a false sense of security, **opening an intriguing direction** for future work at the intersection of personalization and privacy protection.

The quantitative results show consistent improvements across both frameworks when combined with our method, confirming its effectiveness and robustness. These results are complemented by qualitative examples, which are available in our **anonymous GitHub repository** (linked in the abstract of the paper). **We encourage reviewers to explore these results for a more complete understanding of our method's impact**.

While it is unfortunate that we do not have the opportunity to further discuss these points with the reviewers, we believe that our comprehensive responses have addressed their remaining concerns and have further strengthened the contributions and validity of our method.

Once again, we sincerely thank you for your constructive feedback and thoughtful evaluation.

---

### Meta-Review · Area_Chair_dwui · 2026-01-05

**Summary:**

The paper identifies semantic collapsing in generative personalization as a drift of learned concept embeddings and proposes a simple, training‑free inference‑time adjustment of embedding magnitude and direction. Most reviewers appreciate that the proposed TEA method is lightweight, requires no retraining, and generalizes well across diverse frameworks. The AC finds that most major concerns raised by the reviewers have been well addressed during the rebuttal, including the lack of comparisons to recent mainstream work and the unjustified choice of SLERP, etc.

**Reviewer Concerns:**

Several major concerns were raised before discussion: the paper lacks comparisons to recent mainstream works (e.g., OmniControl), the evaluation is insufficient, the choice of SLERP is not justified, and TEA relies on fixed hyperparameters across all prompts. During the rebuttal, the authors provided additional comparisons to methods including OmniControl, offered a stronger justification for the choice of SLERP, and explained their decision to use fixed hyperparameters. The AC finds the rebuttal sound and adequate.

**Reviewer Scores:**

This paper receives the following ratings: Marginally Below, Marginally Below, Marginally Below, and Marginally Above. With the additional results and justifications provided in the authors’ rebuttal, if the reviewers had been able to participate fully in the discussion, the AC would expect those “Marginally Below” ratings to be highly likely raised to “Marginally Above.” Thus, the AC recommends accepting this paper.

---

### Decision · Program_Chairs · 2026-01-26

Accept (Poster)